# Iterative Value-Aware Model Learning

**Amir-massoud Farahmand**[*]
Vector Institute, Toronto, Canada
farahmand@vectorinstitute.ai

## Abstract

This paper introduces a model-based reinforcement learning (MBRL) framework that incorporates the underlying decision problem in learning the transition model of the environment. This is in contrast with conventional approaches to MBRL that learn the model of the environment, for example by finding the maximum likelihood estimate, without taking into account the decision problem. Value-Aware Model Learning (VAML) framework argues that this might not be a good idea, especially if the true model of the environment does not belong to the model class from which we are estimating the model.

The original VAML framework, however, may result in an optimization problem that is difficult to solve. This paper introduces a new MBRL class of algorithms, called Iterative VAML, that benefits from the structure of how the planning is performed (i.e., through approximate value iteration) to devise a simpler optimization problem. The paper theoretically analyzes Iterative VAML and provides finite sample error upper bound guarantee for it.

## 1 Introduction

Value-Aware Model Learning (VAML) is a novel framework for learning the model of the environment in Model-Based Reinforcement Learning (MBRL) [Farahmand et al., 2017a, 2016a]. The conventional approach to model learning in MBRL is based on minimizing some kind of probabilistic loss. A common choice is to minimize the KL-divergence between the empirical data and the model, which leads to the Maximum Likelihood Estimator (MLE). Farahmand et al. [2017a, 2016a] argue that minimizing a probabilistic loss function might not be a good idea because it does not take into account the underlying decision problem. Any knowledge about the reward, value function, or policy is ignored in the conventional model learning approaches in MBRL (some recent exceptions are Joseph et al. [2013], Silver et al. [2017], Oh et al. [2017], Farquhar et al. [2018]; refer to the supplementary material for a detailed literature review of MBRL). The main thesis behind *decision-aware model learning*, including VAML, is that the knowledge about the underlying decision problem, which is often available, should be considered in the model learning itself. VAML, as its name suggests, uses the information about the value function. In particular, the formulation by Farahmand et al. [2017a] incorporates the knowledge about the *value function space* in learning the model. In this work, we suggest an alternative, and possibly simpler, approach called Iterative VAML (IterVAML, for short).

VAML defines a robust loss function and has a $\min_{\mathcal{P} \in \mathcal{M}} \max_{V \in \mathcal{F}}$ structure, where $\mathcal{M}$ is the transition probability model of the environment to be learned and $\mathcal{F}$ is the function space to which the value function belongs (we discuss this in more detail in Section 2). Solving this $\min \max$ optimization can be difficult in general, unless we impose some structure on $\mathcal{F}$, e.g., linear function space. IterVAML mitigates this issue by benefiting from special structure of how value functions are generated within the approximate value iteration (AVI) framework (Section 3).

---

[*]Homepage: http://academic.sologen.net. Part of this work has been done when the author was affiliated with Mitsubishi Electric Research Laboratories (MERL), Cambridge, USA.

We theoretically analyze IterVAML (Section 4). We provide a finite-sample error upper bound guarantee for the model learning that shows the effect of the number of samples and complexity of the model on the error bound (Section 4.1). We also analyze how the errors in the learned model affect the quality of the outcome policy. This is in the form of an error propagation result (Section 4.2).

## 2 Background on Value-Aware Model Learning

To formalize the framework, let us consider a discounted Markov Decision Process (MDP) $(\mathcal{X}, \mathcal{A}, \mathcal{R}^*, \mathcal{P}^*, \gamma)$ [Szepesvári, 2010]. Here $\mathcal{X}$ is the state space, $\mathcal{A}$ is the action space, $\mathcal{R}^*$ is the reward distribution, $\mathcal{P}^*$ is the transition probability kernel, and $0 \leq \gamma < 1$ is the discount factor. In the RL setting, $\mathcal{P}^*$ and $\mathcal{R}^*$ are not known to the agent. Instead, the agent can interact with the environment to collect samples from these distributions. The collected data is in the form of

$$\mathcal{D}_n = \{(X_i, A_i, R_i, X_i')\}_{i=1}^n, \tag{1}$$

with the current state-action being distributed according to $Z_i = (X_i, A_i) \sim \nu(\mathcal{X} \times \mathcal{A}) \in \bar{\mathcal{M}}(\mathcal{X} \times \mathcal{A})$, the reward $R_i \sim \mathcal{R}^*(\cdot|X_i, A_i)$, and the next-state $X_i' \sim \mathcal{P}^*(\cdot|X_i, A_i)$. We denote the expected reward by $r(x, a) = \mathbb{E}[\mathcal{R}^*(\cdot|x, a)]$.[2]

The goal of model learning is to find a $\hat{\mathcal{P}}$ that is *close* to $\mathcal{P}^*$.[3] The learned model $\hat{\mathcal{P}}$ is then used by an MBRL algorithm to find a policy. To formalize this, let us denote Planner as an algorithm that receives a model $\hat{\mathcal{P}}$ and returns a policy, i.e., $\pi \leftarrow \mathsf{Planner}(\hat{\mathcal{P}})$. We assume that the reward function is already known to Planner, so we do not explicitly pass it as an argument. There are many variations on how Planner may use the learned model to obtain a new policy. For example, Planner might be a value function-based approach that computes an estimate of the optimal value function based on $\hat{\mathcal{P}}$, and then returns the greedy policy of the estimated value function. Or it might be a policy gradient method that computes the gradient of the performance with respect to (w.r.t.) the policy using the learned model.

A fundamental question is how we should measure the closeness of $\hat{\mathcal{P}}$ to $\mathcal{P}^*$. The answer to this question depends on how Planner is going to use the model. It is possible that some aspects of the dynamics is irrelevant to Planner. The usual approaches based on the probabilistic losses, such as the KL-divergence that leads to MLE, ignore this dependency. Therefore, they might be less efficient than an approach that considers how Planner is going to use the learned model.

VAML, introduced by Farahmand et al. [2016a, 2017a], is a value-based approach and assumes that Planner uses the Bellman optimality operator defined based on $\hat{\mathcal{P}}$ to find a $\hat{Q}^*$, that is

$$T_{\hat{\mathcal{P}}}^* : Q \mapsto r + \gamma \hat{\mathcal{P}} \max_a Q, \tag{2}$$

and then outputs $\pi = \hat{\pi}(\cdot; \hat{Q}^*)$, the greedy policy w.r.t. $\hat{Q}^*$ defined as $\hat{\pi}(x; Q) = \mathrm{argmax}_{a \in \mathcal{A}} Q(x, a)$. For brevity, we sometimes use $\hat{T}^*$ instead of $T_{\hat{\mathcal{P}}}^*$. The use of the Bellman [optimality] operator is central to value-based approaches such as the family of (Approximate) Value Iteration [Gordon, 1995, Szepesvári and Smart, 2004, Ernst et al., 2005, Munos and Szepesvári, 2008, Farahmand et al., 2009, Farahmand and Precup, 2012, Mnih et al., 2015, Tosatto et al., 2017, Farahmand et al., 2017b] or (Approximate) Policy Iteration (API) algorithms [Lagoudakis and Parr, 2003, Antos et al., 2008, Bertsekas, 2011, Lazaric et al., 2012, Scherrer et al., 2012, Farahmand et al., 2016b].

VAML focuses on finding $\hat{\mathcal{P}}$ such that the difference between $T^*Q$ and $\hat{T}^*Q$ is small. It starts from assuming that $V$ is known and defines the pointwise loss (or cost) between $\hat{\mathcal{P}}$ and $\mathcal{P}^*$ as

$$c(\hat{\mathcal{P}}, \mathcal{P}^*; V)(x, a) = \left| \left\langle \mathcal{P}^*(\cdot|x, a) - \hat{\mathcal{P}}(\cdot|x, a), V \right\rangle \right|$$

$$= \left| \int \left[ \mathcal{P}^*(\mathrm{d}x'|x, a) - \hat{\mathcal{P}}(\mathrm{d}x'|x, a) \right] V(x') \right|, \tag{3}$$

in which we substituted $\max_a Q(\cdot, a)$ in (2) with $V$ to simplify the presentation. In the rest of the paper, we sometimes use $\mathcal{P}_z(\cdot)$ with $z = (x, a) \in \mathcal{Z} = \mathcal{X} \times \mathcal{A}$ to refer to the probability distribution $\mathcal{P}(\cdot|x, a)$, so $\mathcal{P}_z V = \int \mathcal{P}(\mathrm{d}x'|x, a)V(x')$.

Given a probability distribution $\nu \in \bar{\mathcal{M}}(\mathcal{X} \times \mathcal{A})$, which can be the same distribution as the data generating one, VAML defines the expected loss function

$$c_{2,\nu}^2(\hat{\mathcal{P}}, \mathcal{P}^*; V) = \int \mathrm{d}\nu(x, a) \left| \int \left[ \mathcal{P}^*(\mathrm{d}x'|x, a) - \hat{\mathcal{P}}(\mathrm{d}x'|x, a) \right] V(x') \right|^2 . \tag{4}$$

Notice that the value function $V$ is unknown, so we cannot readily minimize this loss function, or its empirical version. What differentiates this work from the original VAML formulation is how this unknown $V$ is handled. VAML takes a robust approach: It considers the worst-case choice of $V$ in the value function space $\mathcal{F}$ that is used by Planner. Therefore, it minimizes

$$c_{2,\nu}^2(\hat{\mathcal{P}}, \mathcal{P}^*) = \int \mathrm{d}\nu(x, a) \sup_{V \in \mathcal{F}} \left| \int \left[ \mathcal{P}^*(\mathrm{d}x'|x, a) - \hat{\mathcal{P}}(\mathrm{d}x'|x, a) \right] V(x') \right|^2 . \tag{5}$$

As argued by Farahmand et al. [2017a], this is still a tighter objective to minimize than the KL-divergence. To see this, consider a fix $z = (x, a)$. We have $\sup_{V \in \mathcal{F}} |\langle \mathcal{P}^*(\cdot|x, a) - \hat{\mathcal{P}}(\cdot|x, a), V \rangle| \leq \|\mathcal{P}_z^* - \hat{\mathcal{P}}_z\|_1 \sup_{V \in \mathcal{F}} \|V\|_\infty \leq \sqrt{2\mathsf{KL}(\mathcal{P}_z^* || \hat{\mathcal{P}}_z)} \sup_{V \in \mathcal{F}} \|V\|_\infty$, where we used Pinsker's inequality. As MLE is the minimizer of the KL-divergence based on data, these upper bounds suggest that if we find a good MLE (with small KL-divergence), we also have an accurate Bellman operator. These sequences of upper bounding, however, might be quite loose. For an extreme, but instructive, example, suppose that the value function space consists of bounded constant functions ($\mathcal{F} = \{ x \mapsto c : |c| < \infty \}$). In that case, $\sup_{V \in \mathcal{F}} |\langle \mathcal{P}^*(\cdot|x, a) - \hat{\mathcal{P}}(\cdot|x, a), V \rangle|$ is always zero, no matter how large the total variation and the KL-divergence of two distributions are. MLE does not explicitly benefit from these interaction of the value function and the model. Asadi et al. [2018] show that the VAML objective $\sup_{V \in \mathcal{F}} |\langle \mathcal{P}^*(\cdot|x, a) - \hat{\mathcal{P}}(\cdot|x, a), V \rangle|$ with the choice of 1-Lipschitz functions for $\mathcal{F}$ is equivalent to the Wasserstein metric between $\mathcal{P}^*(\cdot|x, a)$ and $\hat{\mathcal{P}}(\cdot|x, a)$. Refer to Farahmand et al. [2017a] for more detail and discussion about VAML and its properties.

The loss function (5) defines the population version loss of VAML. The empirical version, which is minimized in practice, replaces $\mathcal{P}^*$ and $\nu$ by their empirical distributions. The result is

$$c_{2,n}^2(\hat{\mathcal{P}}) = \frac{1}{n} \sum_{(X_i, A_i, X_i') \in \mathcal{D}_n} \sup_{V \in \mathcal{F}} \left| V(X_i') - \int \hat{\mathcal{P}}(\mathrm{d}x'|X_i, A_i)V(x') \right|^2 . \tag{6}$$

The estimated probability distribution $\hat{\mathcal{P}}$ is obtain by solving the following optimization problem:

$$\hat{\mathcal{P}} \leftarrow \operatorname*{argmin}_{\mathcal{P} \in \mathcal{M}} c_{2,n}^2(\mathcal{P}). \tag{7}$$

Farahmand et al. [2017a] provide an expression for the gradient of this objective function when $\mathcal{F}$ is the space of linear function approximators, commonly used in the RL literature, and $\mathcal{M}$ is an exponential family. They also provide finite sample error upper bound guarantee showing that the minimizer of (6) converges to the minimizer of (5).

## 3 Iterative Value-Aware Model Learning

In this section we describe an alternative approach to formulating a value-aware model learning method. As opposed to the original formulation (5), and its empirical version (6), it is not based on a worst-case formulation. Instead, it defines the loss function based on the actual sequence of value functions generated by an (Approximate) Value Iteration (AVI) type of Planner.

Consider the Value Iteration (VI) procedure: At iteration $k = 0, 1, \ldots,$

$$Q_{k+1}(x, a) \leftarrow r(x, a) + \gamma \int \mathcal{P}^*(\mathrm{d}x'|x, a) \max_{a'} Q_k(x', a'), \tag{8}$$

or more succinctly,

$$Q_{k+1} \leftarrow T_{\mathcal{P}*}^* Q_k \triangleq r + \gamma \mathcal{P}^* V_k,$$

with $V_k(x) \triangleq \max_a Q_k(x, a)$. Here $T_{\mathcal{P}*}^*$ is the Bellman optimality operator defined based on the true transition probability kernel $\mathcal{P}^*$ (we similarly define $T_{\mathcal{P}}^*$ for any generic transition probability kernel $\mathcal{P}$). Because of the contraction property of the Bellman optimality operator for discounted MDPs, we have $Q_k \to Q^*$ as $k \to \infty$. This is the basis of the VI procedure.

The intuition behind IterVAML can be developed by studying the sequence $Q_0, Q_1, \ldots$ generated by VI. Starting from $Q_0 \leftarrow r$, VI generates the following sequence

$$\begin{aligned} Q_0 &\leftarrow r \\ Q_1 &\leftarrow T_{\mathcal{P}*}^* V_0 = r + \gamma \mathcal{P}^* r \\ Q_2 &\leftarrow T_{\mathcal{P}*}^* V_1 = r + \gamma \mathcal{P}^* V_1 \\ &\vdots \end{aligned}$$

To obtain the value of $Q_0$, we do not need the knowledge of $\mathcal{P}^*$. To obtain the value of $Q_1$, we only need to compute $\mathcal{P}^* V_0 = \mathcal{P}^* r$. If we find a $\hat{\mathcal{P}}$ such that

$$\hat{\mathcal{P}} r = \mathcal{P}^* r,$$

we may replace it with $\mathcal{P}^*$ to obtain $Q_1$ without any error, i.e., $Q_1 = r + \gamma \mathcal{P}^* r = r + \gamma \hat{\mathcal{P}} r$. Likewise, for any $k \geq 1$ and given $Q_k$, in order to compute $Q_{k+1}$ exactly we only need to find a $\hat{\mathcal{P}}$ such that

$$\hat{\mathcal{P}} V_k = \mathcal{P}^* V_k.$$

We have two sources of errors though. The first is that we may only guarantee that

$$\hat{\mathcal{P}} V_k \approx \mathcal{P}^* V_k.$$

As a result, $T_{\hat{\mathcal{P}}}^* V_k$ is not exactly the same as $T_{\mathcal{P}*}^* V_k$. IterVAML's goal is to make the error $\hat{\mathcal{P}} V_k - \mathcal{P}^* V_k$ as small as possible. Based on this, we define the following "idealized" optimization problem:

Given a model space $\mathcal{M}$ and the current approximation of the value function $\hat{Q}_k$ (and therefore $\hat{V}_k$), solve

$$\hat{\mathcal{P}}^{(k)} \leftarrow \underset{\mathcal{P} \in \mathcal{M}}{\arg\min} \left\| (\mathcal{P} - \mathcal{P}^*) \hat{V}_k \right\|_2^2 = \int \left| (\mathcal{P} - \mathcal{P}^*)(\mathrm{d}x'|z) \max_{a'} \hat{Q}_k(x', a') \right|^2 \mathrm{d}\nu(z), \qquad (9)$$

where $\nu \in \bar{\mathcal{M}}(\mathcal{X} \times \mathcal{A})$ is a user-defined distribution over the state-action space. Oftentimes, this distribution is the same as the empirical distribution generating $\mathcal{D}_n$ (1), hence our use of the same notation. Afterwards, we use $\hat{\mathcal{P}}^{(k)}$ to find $\hat{V}_{k+1}$ by using the usual VI approach, that is,

$$\hat{Q}_{k+1} \leftarrow T_{\hat{\mathcal{P}}^{(k)}}^* \hat{Q}_k. \qquad (10)$$

This procedure is repeated. This is using the exact VI based on $\hat{\mathcal{P}}^{(k)}$.

This formulation is idealized as we do not have access to $\mathcal{P}^*$ and $\nu$, but only samples from them. We use the empirical distribution instead:

$$\hat{\mathcal{P}}^{(k+1)} \leftarrow \underset{\mathcal{P} \in \mathcal{M}}{\arg\min} \frac{1}{n} \sum_{(X_i, A_i, X_i') \in \mathcal{D}_n} \left| \hat{V}_k(X_i') - \int \mathcal{P}(\mathrm{d}x'|X_i, A_i) \hat{V}_k(x') \right|^2. \qquad (11)$$

The optimization problem minimizes the distance between the next-state expectation of $\hat{V}_k$ according to $\mathcal{P}$ and the samples $\hat{V}_k(X')$ with $X'$ being drawn from the true next-state distribution. In case the integral is difficult to compute, we may replace it by samples from $\mathcal{P}$, i.e.,

$$\int \mathcal{P}(\mathrm{d}x'|X_i, A_i) \hat{V}_k(x') \approx \frac{1}{m} \sum_{j=1}^m \hat{V}_k(X_{i,j}'),$$

with $X'_{i,j} \sim \mathcal{P}(\cdot|X_i, A_i)$ for $j = 1, \ldots, m$. These are "virtual" samples generated from the model.

The second source of error is that the VI cannot be performed exactly (for example because the state space is very large), and can only be performed approximately. This leads to the Approximate Value Iteration (AVI) procedure (also known as Fitted Value or Q-Iteration) with a function space $\mathcal{F}^{|\mathcal{A}|}$ (the space of action-value functions), see e.g., Ernst et al. [2005], Munos and Szepesvári [2008], Farahmand et al. [2009], Farahmand and Precup [2012], Mnih et al. [2015], Tosatto et al. [2017]. Instead of setting $\hat{Q}_{k+1} \leftarrow T^*_{\hat{\mathcal{P}}^{(k)}} \hat{Q}_k$, in AVI we have

$$\hat{Q}_{k+1} \leftarrow \operatorname*{argmin}_{Q \in \mathcal{F}^{|\mathcal{A}|}} \frac{1}{n} \sum_{(X_i, A_i, R_i) \in \mathcal{D}_n} \left| Q(X_i, A_i) - \left( R_i + \gamma \int \hat{\mathcal{P}}^{(k+1)}(\mathrm{d}x'|X_i, A_i)\hat{V}_k(x') \right) \right|^2. \tag{12}$$

Notice that finding $\hat{Q}_{k+1}$ is a regression problem, for which many methods to solve are available, including the regularized variants of this empirical risk minimization problem [Farahmand et al., 2009]. As before, we may replace the integral with virtual samples, i.e.,

$$\int \hat{\mathcal{P}}^{(k+1)}(\mathrm{d}x'|X_i, A_i)\hat{V}_k(x') \approx \frac{1}{m} \sum_{j=1}^m \hat{V}_k(X'_{i,j}),$$

with $X'_{i,j} \sim \hat{\mathcal{P}}^{(k+1)}(\cdot|X_i, A_i)$ for $j = 1, \ldots, m$, for each $i = 1, \ldots, n$. These "virtual" samples from the model play the same role as in the hypothetical experience in the Dyna architecture [Sutton, 1990] or imagination in imagination-augmented agents by Racanière et al. [2017].

Algorithm 1 summarizes a generic IterVAML procedure. The algorithm receives a model space $\mathcal{M}$, the action-value function space $\mathcal{F}^{|\mathcal{A}|}$, the space of reward functions $\mathcal{G}$, and the number of iterations $K$. At each iteration $k = 0, 1, \ldots$, it generates a fresh training dataset $\mathcal{D}_n^{(k)} = \{(X_i, A_i, R_i, X'_i)\}_{i=1}^n$ by interacting with the environment. It learns the transition model $\hat{\mathcal{P}}^{(k+1)}$ by solving (11). It also learns the reward function $\hat{r}$, by minimizing $\mathrm{Loss}_{\mathcal{R}}$, which can be the squared loss (or a robust variant). We do not analyze learning the reward function in this work. Afterwards, it performs one step of AVI by solving (12). These steps are repeated for $K$ iterations.

Many variations of this algorithm are possible. We briefly remark on some of them. Here the AVI step only uses the model $\hat{\mathcal{P}}$. In practice, however, one may use both the learned model $\hat{\mathcal{P}}$ and the data $\mathcal{D}_n$ in solving the optimization problem (12) in order to obtain better solutions, as in the Dyna architecture [Sutton, 1990]. Moreover, the summation in (12) is $\mathcal{D}_n$ (or $\cup_{i=0}^k \mathcal{D}_n^{(i)}$ as stated in the algorithm), which is the dataset of true samples. If we also learn a distribution model of $\nu$ by $\hat{\nu}$, we can sample from it too. In that case, we can increase the number of samples used in solving the regression step of IterVAML. We have more discussion about the algorithm in the supplementary material.

We can also similarly define a *policy evaluation* version of IterVAML.

## 4 Theoretical Analysis of Iterative VAML

We analyze the statistical properties of the IterVAML procedure. The analysis is divided into two parts. First we analyze one iteration of model learning (cf. (11)) and provide an upper bound on the error in learning the model (Theorem 1 in Section 4.1). Afterwards, we consider how errors at each iteration propagate throughout the iterations of IterVAML and affect the quality of the learned policy (Theorem 2 in Section 4.2). Theorem 3 combines these two results and shows how the model learning errors affect the quality of the outcome policy. The proofs and more extensive discussions are all referred to the extended version of the paper, which is provided as a supplementary material.

### 4.1 Error Analysis for a Single Iteration

We analyze the $k$-th iteration of IterVAML (11) and provide an error bound on $\|(\hat{\mathcal{P}}^{(k+1)} - \mathcal{P}^*)V_k\|_2$. To reduce clutter, we do not specify the iteration index $k$, e.g., the analyzed loss would be denoted by $\|(\hat{\mathcal{P}} - \mathcal{P}^*)V\|_2$ for a fixed $V$.

**Algorithm 1** Model-based Reinforcement Learning Algorithm with Iterative VAML

---

// MDP $(\mathcal{X}, \mathcal{A}, \mathcal{R}^*, \mathcal{P}^*, \gamma)$
// $K$: Number of iterations
// $\mathcal{M}$: Space of transition probability kernels
// $\mathcal{F}^{|\mathcal{A}|}$: Space of action-value functions
// $\mathcal{G}$: Space of reward functions
Initialize a policy $\pi_0$ and a value function $\hat{V}_0$.
**for** $k = 0$ to $K - 1$ **do**
    Generate training set $\mathcal{D}_n^{(k)} = \{(X_i, A_i, R_i, X_i')\}_{i=1}^n$ by interacting with the true environment (potentially using $\pi_k$), i.e., $(X_i, A_i) \sim \nu_k$ with $X_i' \sim \mathcal{P}^*(\cdot|X_i, A_i)$ and $R_i \sim \mathcal{R}^*(\cdot|X_i, A_i)$.
    $\hat{\mathcal{P}}^{(k+1)} \leftarrow \operatorname{argmin}_{\mathcal{P} \in \mathcal{M}} \left\| \hat{V}_k(X_i') - \int \mathcal{P}(\mathrm{d}x'|X_i, A_i)\hat{V}_k(x') \right\|_{\cup_{i=0}^k \mathcal{D}_n^{(i)}}^2$.
    $\hat{r} \leftarrow \operatorname{argmin}_{r \in \mathcal{G}} \operatorname{Loss}_{\mathcal{R}}(r; \cup_{i=0}^k \mathcal{D}_n^{(i)})$
    $\hat{Q}_{k+1} \leftarrow \operatorname{argmin}_{Q \in \mathcal{F}^{|\mathcal{A}|}} \left\| Q(X_i, A_i) - \left( \hat{r}(X_i, A_i) + \gamma \int \hat{\mathcal{P}}^{(k+1)}(\mathrm{d}x'|X_i, A_i)\hat{V}_k(x') \right) \right\|_{\cup_{i=0}^k \mathcal{D}_n^{(i)}}^2$.
    $\pi_{k+1} \leftarrow \hat{\pi}(\cdot; \hat{Q}_{k+1})$.
**end for**
**return** $\pi_K$

---

Consider a fixed value function $V : \mathcal{X} \to \mathbb{R}$. We are given a dataset $\mathcal{D}_n = \{(X_i, A_i, X_i')\}_{i=1}^n$ with $Z_i = (X_i, A_i) \sim \nu(\mathcal{X} \times \mathcal{A}) \in \bar{\mathcal{M}}(\mathcal{X} \times \mathcal{A})$, and the next-state $X_i' \sim \mathcal{P}^*(\cdot|X_i, A_i)$, as specified in (1).

We now enlist our set of assumptions. Some of them are technical assumptions to simplify the analysis, and some are characterizing crucial aspects of the model learning. We shall remark on these as we introduce them.

**Assumption A1 (Samples)** At the $k$-th iteration we are given a dataset $\mathcal{D}_n (= \mathcal{D}_n^{(k)})$

$$\mathcal{D}_n = \{(X_i, A_i, X_i')\}_{i=1}^n, \tag{13}$$

with $Z_i = (X_i, A_i)$ being independent and identically distributed (i.i.d.) samples drawn from $\nu(\mathcal{X} \times \mathcal{A}) \in \bar{\mathcal{M}}(\mathcal{X} \times \mathcal{A})$ and the next-state $X_i' \sim \mathcal{P}^*(\cdot|X_i, A_i)$. Furthermore, we assume that $\mathcal{D}_n^{(k)}$ and $\mathcal{D}_n^{(k')}$ for $k \neq k'$ are independent.

The i.i.d. assumption is to simplify the analysis, and with extra effort one can provide similar results for dependent processes that gradually "forget" their past. The forgetting behaviour can be characterized by the mixing behaviour of the stochastic process [Doukhan, 1994]. One can then provide statistical guarantees for learning algorithms under various mixing conditions [Yu, 1994, Meir, 2000, Steinwart and Christmann, 2009, Mohri and Rostamizadeh, 2009, 2010, Farahmand and Szepesvári, 2012].

In this assumption we also require that the datasets of two different iterations are independent. This is again to simplify the analysis. In practice, we might reuse the same dataset in all iterations. Theoretical results by Munos and Szepesvári [2008] suggest that the dependence between iterations may not lead to significant performance degradation.

We need to make some assumptions about the model space $\mathcal{M}$ and its complexity (i.e., capacity). We use covering number (and its logarithm, i.e., metric entropy) of a function space (here being the model space $\mathcal{M}$) as the characterizer of its complexity. The covering number at resolution $\varepsilon$ is the minimum number of balls with radius $\varepsilon$ required to cover the model space $\mathcal{M}$ according to a particular metric, and is denoted by $\mathcal{N}(\varepsilon, \mathcal{M})$ (see the supplementary material for definitions). As $\varepsilon$ decreases, the covering number increases (or more accurately, the covering number is non-decreasing). For example, the covering number for a $p$-dimensional linear function approximator with constraint on the magnitude of its functions behaves like $O(\frac{1}{\varepsilon^p})$. A similar result holds when the subgraphs of a function space has a VC-dimension $p$. Model spaces whose covering number grows faster are more complex, and estimating a function within them is more difficult. This leads to larger estimation error, as we shall see. On the other hand, those model spaces often (but not always) have better model approximation properties too.

In order to show the finer behaviour of the error bound, we define $\mathcal{M}$ as a subset of a larger family of probability distributions $\mathcal{M}_0$. Let $J : \mathcal{M}_0 \to [0, \infty)$ be a pseudo-norm defined on functions in $\mathcal{M}_0$.

We then define $\mathcal{M} = \{ \mathcal{P} \in \mathcal{M}_0 \, : \, J(\mathcal{P}) \leq R \}$ for some $R > 0$. One may think of $J$ as a measure of complexity of functions in $\mathcal{M}_0$, so $\mathcal{M}$ would be a ball with a fixed radius $R$ w.r.t. $J$. If $\mathcal{M}_0$ is defined based on a reproducing kernel Hilbert space (RKHS), we can think of $J$ as the inner product norm of the RKHS.

**Assumption A2 (Model Space)** For $R > 0$, let $\mathcal{M} = \mathcal{M}_R = \{ \mathcal{P} \in \mathcal{M}_0 \, : \, J(\mathcal{P}) \leq R \}$. There exist constants $c > 0$ and $0 < \alpha < 1$ such that for any $\varepsilon, R > 0$ and all sequences $z_{1:n} \triangleq z_1, \ldots, z_n \subset \mathcal{X} \times \mathcal{A}$, the following metric entropy condition is satisfied:

$$\log \mathcal{N}\left(\varepsilon, \mathcal{M}, L_2(z_{1:n})\right) \leq c \left(\frac{R}{\varepsilon}\right)^{2\alpha}.$$

Furthermore, the model space $\mathcal{M}$ is convex, and compact w.r.t. $d_{\infty,\mathrm{TV}}(\mathcal{P}_1, \mathcal{P}_2) = \sup_{z \in \mathcal{Z}} \int |\mathcal{P}_1(\mathrm{d}y|z) - \mathcal{P}_2(\mathrm{d}y|z)|$.

This form of the metric entropy of $\mathcal{M} = \mathcal{M}_R$ is suitable to capture the complexity of large function spaces such as some RKHS and Sobolev spaces. For example, for $\mathbb{W}^k(\mathbb{R}^d) = \mathbb{W}^{k,2}(\mathbb{R}^d)$, the Sobolev space defined w.r.t. the $L_2$-norm of the weak derivatives up to order $k$, we can set $\alpha = \frac{d}{2k}$, see e.g., Lemma 20.6 of Györfi et al. [2002]. For smaller function spaces, such as the $p$-dimensional linear function approximator mentioned above, the behaviour of the metric entropy is $p \log(\frac{1}{\varepsilon})$, which can be seen as having $\alpha \to 0$ with a certain rate, For many examples of the covering number and metric entropy results, refer to van de Geer [2000], Györfi et al. [2002], Zhou [2003], Steinwart and Christmann [2008], Giné and Nickl [2015]. Also note that here we require the convexity and compactness of $\mathcal{M}$. The convexity is a crucial assumption for obtaining fast convergence rate, as was shown and discussed by Lee et al. [1998, 2008], Mendelson [2008]. The compactness w.r.t. this particular metric is a technical assumption and it may be possible to be relaxed.

**Assumption A3 (Value Function)** The value function $V$ is fixed (i.e., not dependent on $\mathcal{D}_n$) and is $V_{\max}$-bounded with $V_{\max} \geq 1$.

This assumption is for the simplicity of the analysis. We use it in large deviation results that require the boundedness of the involved random variables, e.g., Theorem 2.1 of Bartlett et al. [2005] or Theorem 19.1 of Györfi et al. [2002], which we use in our proofs.

We are now ready to state the main result of this section.

**Theorem 1.** *Suppose that Assumptions A1, A2, and A3 hold. Consider $\hat{\mathcal{P}}$ obtained by solving* (11). *There exists a finite $c(\alpha) > 0$, depending only on $\alpha$, such that for any $\delta > 0$, with probability at least $1 - \delta$, we have*

$$\left\| (\hat{\mathcal{P}}_z - \mathcal{P}_z^*)V \right\|_{2,\nu}^2 \leq \inf_{\mathcal{P} \in \mathcal{M}} \| (\mathcal{P}_z - \mathcal{P}_z^*)V \|_{2,\nu}^2 + \frac{c(\alpha) V_{max}^2 R^{\frac{2\alpha}{1+\alpha}} \sqrt{\log(1/\delta)}}{n^{\frac{1}{1+\alpha}}}.$$

This result upper bounds the error of $\hat{\mathcal{P}}$ in approximating the next-state expectation of the value function. The upper bound has the model (or function) approximation error (the first term) and the estimation error (the second term). It is notable that the constant in front of the model approximation error is one, so the best we can hope from this algorithm, in the limit of infinite data, is as good as the best model in the model class $\mathcal{M}$.

The estimation error behaves as $n^{-1/(1+\alpha)}$ ($0 < \alpha < 1$). This is a fast rate and can reach $n^{-1}$ whenever $\alpha \to 0$. We do not know whether this is an optimal rate for this particular problem, but results from regression with least-squares loss suggest that this might indeed be optimal: For a regression function belonging to a function space that has a packing entropy in the same form as in the upper bound of Assumption A2, the rate $\Omega(n^{-1/(1+\alpha)})$ is its minimax lower bound [Yang and Barron, 1999].

We use local Rademacher complexity and analyze the modulus of continuity of empirical processes to obtain rates faster than what could be achieved by more conventional techniques of analyzing the supremum of the empirical processes. Farahmand et al. [2017a] used the supremum of the empirical process to analyze VAML and obtained $n^{-1/2}$ rate, which is slower than $n^{-1/(1+\alpha)}$. Notice that the loss functions of VAML and IterVAML are different, so this is only an approximate comparison. The

rate $n^{-1/2}$, however, is common in the supremum of empirical process-based analysis, so we would expect it to hold if we used those techniques to analyze IterVAML. Finally notice that the error rate of VAML is not necessarily slower than IterVAML's; the present difference is at least somehow due to the shortcoming of their simpler proof technique.

## 4.2 Error Propagation

We analyze how the errors incurred at each step of IterVAML propagate throughout the iterations and affect the quality of the outcome. For policy evaluation, the quality is defined as the difference between $V^\pi$ and $\hat{V}_K$, weighted according to a user-defined probability distribution $\rho_\mathcal{X} \in \bar{\mathcal{M}}(\mathcal{X})$, i.e., $\|V^\pi - \hat{V}_K\|_{1,\rho_\mathcal{X}}$. For the control case we consider the performance loss, which is defined as the difference between the value of following the greedy policy w.r.t. $\hat{Q}_K$ compared to the value of the optimal policy $Q^*$, weighted according to a user-defined probability distribution $\rho \in \bar{\mathcal{M}}(\mathcal{X} \times \mathcal{A})$, i.e., $\rho(Q^* - Q^{\pi_K})$ (cf. Algorithm 1). This type of error propagation analysis has been performed before by Munos [2007], Antos et al. [2008], Farahmand et al. [2010], Scherrer et al. [2012], Huang et al. [2015], Mann et al. [2015], Farahmand et al. [2016c].

Recall that there are two sources of errors in the IterVAML procedure. The first is the error in model learning, which is caused because the model $\hat{\mathcal{P}}^{(k+1)}$ learned by solving the minimization problem (11) only satisfies $\hat{\mathcal{P}}^{(k+1)}\hat{V}_k \approx \mathcal{P}^*\hat{V}_k$ instead of being exact. This error is studied in Section 4.1, and Theorem 1 provides an upper bound on it.

The second source of error is that the AVI performs the Bellman update only approximately. So instead of having $\hat{Q}_{k+1} = T^*_{\hat{\mathcal{P}}^{(k)}}\hat{Q}_k$ (or its policy evaluation equivalent), the function $\hat{Q}_{k+1}$ obtained by solving (12) is only approximately equal to $T^*_{\hat{\mathcal{P}}^{(k)}}\hat{Q}_k$. As already mentioned, this step is essentially solving a regression problem. Hence, many of the standard error guarantees for regression can be used here too, with possibly some minor changes.

Consider a sequence of $\hat{Q}_0, \hat{Q}_1, \ldots, \hat{Q}_K$ with $\hat{Q}_{k+1} \approx T^*_{\hat{\mathcal{P}}^{(k+1)}}\hat{Q}_k$, with $\hat{\mathcal{P}}^{(k+1)}$ being an approximation of the true $\mathcal{P} \triangleq \mathcal{P}^*$. IterVAML, which consists of repeated solving of (11) and (12), is an example of a procedure that generates these $\hat{Q}_k$. The result, however, is more general and does not depend on the particular way $\hat{\mathcal{P}}^{(k+1)}$ and $\hat{Q}_{k+1}$ are produced.

We define the following concentrability coefficients, similar to the coefficient introduced by Farahmand et al. [2010], which itself is a relaxation of the coefficient introduced by Munos [2007]. These coefficients are the Radon-Nikydom (R-N) derivative of the multi-step ahead state-action distribution w.r.t. the distribution $\nu$. The R-N. derivative can be thought of as the ratio of two probability density functions.

**Definition 1** (Expected Concentrability of the Future State-Action Distribution)**.** *Given $\rho, \nu \in \bar{\mathcal{M}}(\mathcal{X} \times \mathcal{A})$, an integer number $k \geq 0$, and an arbitrary sequence of policies $(\pi_i)_{i=1}^m$, the distribution $\rho\mathcal{P}^{\pi_1} \cdots \mathcal{P}^{\pi_k}$ denotes the future state-action distribution obtained when the first state-action is distributed according $\rho$ and the agent follows the sequence of policies $\pi_1, \pi_2, \ldots, \pi_k$. Define*

$$\bar{c}_{VI,\rho,\nu}(k) = \sup_{\pi_1,\ldots,\pi_k} \left\| \frac{\mathrm{d}\rho\mathcal{P}^{\pi_1}\cdots\mathcal{P}^{\pi_k}}{\mathrm{d}\nu} \right\|_{2,\nu}.$$

*If the future state-action distribution $\rho\mathcal{P}^{\pi_1}\cdots\mathcal{P}^{\pi_k}$ is not absolutely continuous w.r.t. $\nu$, we take $\bar{c}_{VI,\rho,\nu}(k) = \infty$. Moreover, for a discount factor $0 \leq \gamma < 1$, define the discounted weighted average concentrability coefficient as*

$$\bar{C}(\rho, \nu) = (1-\gamma)^2 \sum_{k\geq 1} k\gamma^{k-1}\bar{c}_{VI,\rho,\nu}(k).$$

The definition of $\bar{C}(\rho, \nu)$ is similar to the second order discounted future state distribution concentration coefficient of Munos [2007], with the main difference being that it is defined for the expectation of the R-N derivative instead of its supremum.

The following theorem is our main error propagation result. It can be seen as the generalization of the results of Farahmand et al. [2010], Munos [2007] to the case when we use a model that has an error, whereas the aforementioned papers are for model-free case (or when the model is exact). Because of this similarity, several steps of the proof are similar to theirs.

**Theorem 2.** *Consider a sequence of action-value function $(\hat{Q}_k)_{k=0}^K$, and their corresponding $(\hat{V}_k)_{k=0}^K$, each of which is defined as $\hat{V}_k(x) = \max_a \hat{Q}_k(x,a)$. Suppose that the MDP is such that the expected rewards are $R_{max}$-bounded, and $\hat{Q}_0$ is initialized such that it is $V_{max} \leq \frac{R_{max}}{1-\gamma}$-bounded. Let $\varepsilon_k = T^*_{\hat{\mathcal{P}}^{(k+1)}} \hat{Q}_k - \hat{Q}_{k+1}$ (regression error) and $e_k = (\mathcal{P}^* - \hat{\mathcal{P}}^{(k+1)})\hat{V}_k$ (modelling error) for $k = 0, 1, \ldots, K-1$. Let $\pi_K$ be the greedy policy w.r.t. $\hat{Q}_K$, i.e., $\pi_K(x) = argmax_{a \in \mathcal{A}} \hat{Q}(x,a)$ for all $x \in \mathcal{X}$. Consider probability distributions $\rho, \nu \in \bar{\mathcal{M}}(\mathcal{X} \times \mathcal{A})$. We have*

$$\|Q^* - Q^{\pi_K}\|_{1,\rho} \leq \frac{2\gamma}{(1-\gamma)^2}\left[\bar{C}(\rho,\nu)\max_{0 \leq k \leq K-1}\left(\|\varepsilon_k\|_{2,\nu} + \gamma\|e_k\|_{2,\nu}\right) + 2\gamma^K R_{max}\right]$$

We compare this result with the results of Munos [2007], Farahmand et al. [2010], Farahmand [2011] in the same section of the supplementary material. Before stating Theorem 3, which is the direct implication of Theorems 1 and 2, we state another assumption.

**Assumption A4 (Value Function Space)** The value function space $\mathcal{F}^{|\mathcal{A}|}$ is $V_{\max}$-bounded with $V_{\max} \leq \frac{R_{\max}}{1-\gamma}$, and $V_{\max} \geq 1$.

This assumption requires that all the value functions $\hat{Q}_k$ and $\hat{V}_k$ generated by performing a step of AVI (12) and used in the model learning steps (11) are $V_{\max}$-bounded. This ensures that Assumption A3, which is required by Theorem 1, is satisfied in all iterations. This assumption is easy to satisfy in practice by clipping the output of the value function estimator at the level of $\pm V_{\max}$. Theoretical analysis of such a clipped value function estimator, however, is more complicated. As we do not analyze the value function estimation steps of IterVAML, which depends on the choice of $\mathcal{F}^{|\mathcal{A}|}$, we ignore this issue.

**Theorem 3.** *Consider the IterVAML procedure in which at the $k$-th iteration the model $\hat{\mathcal{P}}^{(k+1)}$ is obtained by solving (11) and $\hat{Q}_{k+1}$ is obtained by solving (12). Let $\varepsilon_k = T^*_{\hat{\mathcal{P}}^{(k+1)}} \hat{Q}_k - \hat{Q}_{k+1}$ be the regression error. Suppose that Assumptions A1, A2, and A4 hold. Consider the greedy policy $\pi_K$ w.r.t. $\hat{Q}_K$. For any $\rho \in \bar{\mathcal{M}}(\mathcal{X} \times \mathcal{A})$, there exists a finite $c(\alpha) > 0$, depending only on $\alpha$, such that for any $\delta > 0$, with probability at least $1 - \delta$, we have*

$$\|Q^* - Q^{\pi_K}\|_{1,\rho} \leq \frac{2\gamma}{(1-\gamma)^2}\left[\bar{C}(\rho,\nu)\left(\max_{0 \leq k \leq K-1}\|\varepsilon_k\|_{2,\nu} + \gamma e_{model}(n)\right) + 2\gamma^K R_{max}\right]$$

*where*

$$e_{model}(n) = \sup_{V \in \mathcal{F}^+}\inf_{\mathcal{P} \in \mathcal{M}}\|(\mathcal{P}_z - \mathcal{P}_z^*)V\|_{2,\nu} + \frac{c(\alpha)V_{max}R^{\frac{\alpha}{1+\alpha}}\sqrt[4]{\log(K/\delta)}}{n^{\frac{1}{2(1+\alpha)}}},$$

*and $\mathcal{F}^+ = \left\{\max_a Q(\cdot, a) : Q \in \mathcal{F}^{|\mathcal{A}|}\right\}$.*

This result provides an upper bound on the quality of the learned policy $\pi_K$, as a function of the number of samples and the properties of the model space $\mathcal{M}$ and the MDP. The estimation error due to the model learning is $n^{\frac{1}{2(1+\alpha)}}$, which is discussed in some detail after Theorem 1. The model approximation error term $\sup_{V \in \mathcal{F}^+}\inf_{\mathcal{P} \in \mathcal{M}}\|(\mathcal{P}_z - \mathcal{P}_z^*)V\|_{2,\nu}$ shows the interaction between the model $\mathcal{M}$ and the value function space $\mathcal{F}^{|\mathcal{A}|}$. This quantity is likely to be conservative and can be improved. We also note that an upper bound on $\|\varepsilon_k\|_{2,\nu}$ depends on the regression method, the choice of $\mathcal{F}^{|\mathcal{A}|}$, and the number of samples generated from $\hat{\mathcal{P}}^{(k+1)}$.

## 5 Conclusion

We have introduced IterVAML, a decision-aware model-based RL algorithm. We proved finite sample error upper bound for the model learning procedure (Theorem 1) and a generic error propagation result for an approximate value iteration algorithm that uses an inaccurate model (Theorem 2). The consequence of these two results was Theorem 3, which provides an error upper bound guarantee on the quality of the outcome policy of IterVAML.

There are several possible future research directions. One is empirical studies of IterVAML and comparing it with non-decision-aware methods. Another direction is to investigate other approaches to decision-aware model-based RL algorithms.

**Acknowledgments**

I would like to thank the anonymous reviewers for their helpful feedback, and Mehdi Ghasemi and Murat A. Erdogdu for discussions.

## Footnotes

[2]Given a set $\Omega$ and its $\sigma$-algebra $\sigma_\Omega$, $\bar{\mathcal{M}}(\Omega)$ refers to the set of all probability distributions defined over $\sigma_\Omega$. As we do not get involved in the measure theoretic issues in this paper, we do not explicitly define the $\sigma$-algebra, and simply use a well-defined and "standard" one, e.g., Borel sets defined for metric spaces.

[3]Learning the expected reward $r$ is also a part of model learning, which can be formulated as a regression problem. For simplicity of presentation, we assume that $r$ is known.

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
