[Supplementary Material · IterVAML(NeurIPS2018)(extended).pdf]

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

The algorithm as presented collects a fresh dataset at each iteration, and the optimization problems are solved w.r.t. all collected datasets $\cup_{i=0}^{k} \mathcal{D}_n^{(i)}$. This is not a requirement. We may reuse a single dataset throughout iterations (i.e., a batch setting) or keep the data only from the past few iterations, if the memory is a concern. One can also consider the online variant of IterVAML where instead of solving the optimization problems up to their minimums after collecting the dataset $\mathcal{D}_n$, we keep updating the model and the value function as data arrives. In Appendix D we argue that there is no significant difference in the convergence rate between the case of adding new datasets at each iteration and reusing all of them in the optimization problem compared to generating a fresh dataset at each iteration and throwing out the previous ones. Nevertheless, to simplify the theoretical analysis, we assume that a new fresh dataset is used at each iteration, and the data collected in previous iterations are not used, see Assumption A1 in Section 4.

We can also similarly define a *policy evaluation* version of IterVAML. The goal of a policy evaluation is to find an (approximate) value of a given policy $\pi$. The difference of IterVAML for policy evaluation with what we have already described is that instead of repeated application of the Bellman optimality operator w.r.t. the learned model (i.e., $T^*_{\hat{\mathcal{P}}}$, cf. (8)), we use the Bellman operator

$$(T^\pi_{\hat{\mathcal{P}}} V)(x) = r(x, \pi(a)) + \gamma \int \hat{\mathcal{P}}(\mathrm{d}x'|x, \pi(x)) V(x').$$

---

**Algorithm 1** Model-based Reinforcement Learning Algorithm with Iterative VAML

---
// MDP $(\mathcal{X}, \mathcal{A}, \mathcal{R}^*, \mathcal{P}^*, \gamma)$
// $K$: Number of iterations
// $\mathcal{M}$: Space of transition probability kernels
// $\mathcal{F}^{|\mathcal{A}|}$: Space of action-value functions
// $\mathcal{G}$: Space of reward functions
Initialize a policy $\pi_0$ and a value function $\hat{V}_0$.
**for** $k = 0$ to $K - 1$ **do**
    Generate training set $\mathcal{D}_n^{(k)} = \{(X_i, A_i, R_i, X_i')\}_{i=1}^n$ by interacting with the true environment (potentially using $\pi_k$), i.e., $(X_i, A_i) \sim \nu_k$ with $X_i' \sim \mathcal{P}^*(\cdot|X_i, A_i)$ and $R_i \sim \mathcal{R}^*(\cdot|X_i, A_i)$.
    $\hat{\mathcal{P}}^{(k+1)} \leftarrow \operatorname{argmin}_{\mathcal{P} \in \mathcal{M}} \left\| \hat{V}_k(X_i') - \int \mathcal{P}(\mathrm{d}x'|X_i, A_i)\hat{V}_k(x') \right\|_{\cup_{i=0}^k \mathcal{D}_n^{(i)}}^2$.
    $\hat{r} \leftarrow \operatorname{argmin}_{r \in \mathcal{G}} \operatorname{Loss}_{\mathcal{R}}(r; \cup_{i=0}^k \mathcal{D}_n^{(i)})$
    $\hat{Q}_{k+1} \leftarrow \operatorname{argmin}_{Q \in \mathcal{F}^{|\mathcal{A}|}} \left\| Q(X_i, A_i) - \left( \hat{r}(X_i, A_i) + \gamma \int \hat{\mathcal{P}}^{(k+1)}(\mathrm{d}x'|X_i, A_i)\hat{V}_k(x') \right) \right\|_{\cup_{i=0}^k \mathcal{D}_n^{(i)}}^2$.
    $\pi_{k+1} \leftarrow \hat{\pi}(\cdot; \hat{Q}_{k+1})$.
**end for**
**return** $\pi_K$

---

Here we use the state-value function $V$ instead of $Q$; the definition of $(T_{\hat{\mathcal{P}}}^\pi Q)$ and the corresponding procedure and results would be similar with slight modifications. Given an integer $K$, the IterVAML procedure for policy evaluation would be

$$\hat{V}_0 \approx r$$
$$\hat{\mathcal{P}}^{(1)}\hat{V}_0 \approx \mathcal{P}^*\hat{V}_0$$
$$\hat{V}_1 \approx T_{\hat{\mathcal{P}}^{(1)}}^\pi \hat{V}_0$$
$$\vdots$$
$$\hat{\mathcal{P}}^{(K)}\hat{V}_{K-1} \approx \mathcal{P}^*\hat{V}_{K-1}$$
$$\hat{V}_K \approx T_{\hat{\mathcal{P}}^{(K)}}^\pi \hat{V}_{K-1},$$

where the approximation symbol in $\hat{V}_{k+1} \approx T_{\hat{\mathcal{P}}^{(k+1)}}^\pi \hat{V}_k$ refers to finding $\hat{V}_{k+1}$ that is close to $T_{\hat{\mathcal{P}}^{(k+1)}}^\pi \hat{V}_k$, for example by solving (12); and the approximation symbol in $\hat{\mathcal{P}}^{(k+1)}\hat{V}_k \approx \mathcal{P}^*\hat{V}_k$ refers to solving the empirical version of

$$\hat{\mathcal{P}}^{(k)} \leftarrow \operatorname*{argmin}_{\mathcal{P} \in \mathcal{M}} \left\| (\mathcal{P} - \mathcal{P}^*)\hat{V}_k \right\|_2^2 = \int \left| (\mathcal{P} - \mathcal{P}^*)(\mathrm{d}x'|x, \pi(x))\hat{V}_k(x') \right|^2 \mathrm{d}\nu_{\mathcal{X}}(x).$$

Note that as we use state-value function instead of state-action value function, the distribution $\nu_{\mathcal{X}}$ is defined over $\bar{\mathcal{M}}(\mathcal{X})$ instead of $\nu \in \bar{\mathcal{M}}(\mathcal{X} \times \mathcal{A})$ of (9).

## 4 Theoretical Analysis of Iterative VAML

We analyze the statistical properties of the IterVAML procedure. The analysis is divided into two parts. First we analyze one iteration of model learning (cf. (11)) and provide an upper bound on the error in learning the model (Theorem 1 in Section 4.1). Afterwards, we consider how errors at each iteration propagate throughout the iterations of IterVAML and affect the quality of the learned policy (Theorem 3 in Section 4.2). Theorem 4 combines these two results and shows how the model learning errors affect the quality of the outcome policy.

### 4.1 Error Analysis for a Single Iteration

We analyze the $k$-th iteration of IterVAML (11) and provide an error bound on $\|(\hat{\mathcal{P}}^{(k+1)} - \mathcal{P}^*)V_k\|_2$. To reduce clutter, we do not specify the iteration index $k$, e.g., the analyzed loss would be denoted by $\|(\hat{\mathcal{P}} - \mathcal{P}^*)V\|_2$ for a fixed $V$.

Consider a fixed value function $V : \mathcal{X} \to \mathbb{R}$. We are given a dataset $\mathcal{D}_n = \{(X_i, A_i, X_i')\}_{i=1}^n$ with $Z_i = (X_i, A_i) \sim \nu(\mathcal{X} \times \mathcal{A}) \in \bar{\mathcal{M}}(\mathcal{X} \times \mathcal{A})$, and the next-state $X_i' \sim \mathcal{P}^*(\cdot|X_i, A_i)$, as specified in (1).

For a transition probability kernel $\mathcal{P} : \mathcal{X} \times \mathcal{A} \to \bar{\mathcal{M}}(\mathcal{X})$, and for $z \in \mathcal{X} \times \mathcal{A}$ and $x' \in \mathcal{X}$, we define the following pointwise loss functions

$$l_{\mathcal{P}}(z) = l(z; \mathcal{P}) \triangleq |(\mathcal{P}_z - \mathcal{P}_z^*)V|^2 = \left| \int (\mathcal{P}(\mathrm{d}x'|z) - \mathcal{P}^*(\mathrm{d}x'|z)) V(x') \right|^2,$$

$$\hat{l}(z, x'; \mathcal{P}) = |\mathcal{P}_z V - V(x')|^2.$$

Given a distribution $\nu \in \bar{\mathcal{M}}(\mathcal{X} \times \mathcal{A})$ and the empirical data $\mathcal{D}_n$, we define the following expected/empirical loss functions:

$$L(\mathcal{P}) = \mathbb{E}\left[l(Z; \mathcal{P})\right] = \|(\mathcal{P}_z - \mathcal{P}_z^*)V\|_{2,\nu}^2 \qquad Z \sim \nu,$$

$$L_n(\mathcal{P}) = \frac{1}{n} \sum_{i=1}^n l(Z_i; \mathcal{P}),$$

$$\hat{L}_n(\mathcal{P}) = \frac{1}{n} \sum_{i=1}^n \hat{l}(Z_i, X_i'; \mathcal{P}).$$

The pointwise $l_{\mathcal{P}}$ and it expected value $L(\mathcal{P})$ measure how well $\mathcal{P}$ approximates $\mathcal{P}^*$ through the lens of value function $V$. These are the losses that we are interested in, but as they depend on $\mathcal{P}^*$, which is not available, they cannot be directly optimized. On the other hand, the pointwise loss $\hat{l}$ and its empirical expectation $\hat{L}_n(\mathcal{P})$ are defined only based on data and can be optimized by the algorithm, cf. (9) and (11).

At each iteration of IterVAML, given a fixed $V \in \mathcal{F}$ and a model space $\mathcal{M}$, we solve

$$\hat{\mathcal{P}} \leftarrow \operatorname*{argmin}_{\mathcal{P} \in \mathcal{M}} \hat{L}_n(\mathcal{P}). \tag{13}$$

This is the same as (11). We also define

$$\tilde{\mathcal{P}} \leftarrow \operatorname*{argmin}_{\mathcal{P} \in \mathcal{M}} L(\mathcal{P}). \tag{14}$$

This is (one of) the best models within the model space $\mathcal{M}$. Note that $\tilde{\mathcal{P}}$ is not necessarily unique.

We now enlist our set of assumptions. Some of them are technical assumptions to simplify the analysis, and some are characterizing crucial aspects of the model learning. We shall remark on these as we introduce them.

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

This assumption is for the simplicity of the analysis. We use it in large deviation results that require the boundedness of the involved random variables, e.g., Theorem 2.1 of Bartlett et al. [2005] or Theorem 19.1 of Györfi et al. [2002], which we use in our proofs. There exist similar large deviation results that hold under more relaxed assumptions, for example random variables being sub-Gaussian or sub-exponential [van de Geer, 2000]. An example of an analysis in the RL context that relaxes the usual boundedness assumptions is the work of Tu and Recht [2018]. The condition $V_{\max} \geq 1$ is used to simplify some terms in the bounds, and the results hold with smaller $V_{\max}$ too.

We are now ready to state the main result of this section.

**Theorem 1.** *Suppose that Assumptions A1, A2, and A3 hold. Consider $\hat{\mathcal{P}}$ obtained by solving* (13). *There exists a finite $c(\alpha) > 0$, depending only on $\alpha$, such that for any $\delta > 0$, with probability at least $1 - \delta$, we have*

$$\left\| (\hat{\mathcal{P}}_z - \mathcal{P}_z^*)V \right\|_{2,\nu}^2 \leq \inf_{\mathcal{P} \in \mathcal{M}} \| (\mathcal{P}_z - \mathcal{P}_z^*)V \|_{2,\nu}^2 + \frac{c(\alpha)V_{max}^2 R^{\frac{2\alpha}{1+\alpha}} \sqrt{\log(1/\delta)}}{n^{\frac{1}{1+\alpha}}}.$$

*Proof.* We start from the observation that for any $\mathcal{P}$, the pointwise loss $\hat{l}(z, x'; \mathcal{P})$ can be decomposed as

$$\hat{l}(z, x'; \mathcal{P}) = |\mathcal{P}_z V - V(x')|^2 = |\mathcal{P}_z V - \mathcal{P}_z^* V + \mathcal{P}_z^* V - V(x')|^2$$

$$= \underbrace{|(\mathcal{P}_z - \mathcal{P}_z^*)|^2}_{=l_\mathcal{P}(z)} + |\mathcal{P}_z^* V - V(x')|^2 + 2\left[(\mathcal{P}_z - \mathcal{P}_z^*)V\right]\left[\mathcal{P}_z^* V - V(x')\right].$$

So the empirical loss function $L_n(\mathcal{P})$ can be written as

$$L_n(\mathcal{P}) = \hat{L}_n(\mathcal{P}) - \underbrace{\frac{1}{n}\sum_{i=1}^n |\mathcal{P}_{Z_i}^* V - V(X_i')|^2}_{\triangleq e_\sigma} + 2 \times \underbrace{\frac{1}{n}\sum_{i=1}^n \left[(\mathcal{P}_{Z_i}^* - \mathcal{P}_{Z_i})V\right]\left[\mathcal{P}_{Z_i}^* V - V(X_i')\right]}_{\triangleq e_I(\mathcal{P})}.$$

$$(16)$$

To provide a fast rate, we control the excess error $L(\hat{\mathcal{P}}) - L(\tilde{\mathcal{P}})$, with $\tilde{\mathcal{P}}$ being the best model in the class $\mathcal{M}$, as defined in (14). We define the following function space:

$$\mathcal{G} = \{ z \mapsto l_\mathcal{P}(z) - l_{\tilde{\mathcal{P}}}(z) : \mathcal{P} \in \mathcal{M} \}. \qquad (17)$$

As $l_\mathcal{P} = |(\mathcal{P}_z - \mathcal{P}_z^*)V|^2 \leq 4V_{\max}^2$ and $l_\mathcal{P} \geq 0$, any $g \in \mathcal{G}$ satisfies $-4V_{\max}^2 \leq g(z) \leq 4V_{\max}^2$.

We use Theorem 5 in. Appendix A.3 to upper bound $L(\mathcal{P}) - L(\tilde{\mathcal{P}})$ by a constant multiply of the empirical average $L_n(\mathcal{P}) - L_n(\tilde{\mathcal{P}})$, and some terms depending on the complexity of the function space $\mathcal{G}$. Let us fix $\delta_1 > 0$. With the choice of $K = 2$ in the aforementioned theorem, under the conditions of the theorem, which shall be verified, there exist constants $c_1, c_2, B > 0$ such that for any $\mathcal{P} \in \mathcal{M}$ we have that

$$L(\mathcal{P}) - L(\tilde{\mathcal{P}}) \leq 2\left(L_n(\mathcal{P}) - L_n(\tilde{\mathcal{P}})\right) + \frac{2c_1}{B}r^*(\mathcal{G}) + \frac{(11 \times 8V_{\max}^2 + 2c_2 B)\ln(\frac{1}{\delta_1})}{n}, \qquad (18)$$

with probability at least $1 - \delta_1$. The term $r^*(\mathcal{G})$ corresponds to the local Rademacher complexity of $\mathcal{G}$, defined in Appendix A.3, and is upper bounded by Proposition 10 in Appendix B.3. One can choose $c_1 = 704$ and $c_2 = 26$. We shall specify the value of $B$ later. As the inequality holds for any choice of $\mathcal{P} \in \mathcal{M}$, we substitute $\mathcal{P}$ with $\hat{\mathcal{P}}$.

We upper bound the empirical term $L_n(\hat{\mathcal{P}}) - L_n(\tilde{\mathcal{P}})$ by using the error decomposition (16) along the fact that $\hat{\mathcal{P}}$ is the minimizer of $\hat{L}_n$ within $\mathcal{M}$ as

$$L_n(\hat{\mathcal{P}}) - L_n(\tilde{\mathcal{P}}) = \left[\hat{L}_n(\hat{\mathcal{P}}) - e_\sigma + 2e_I(\hat{\mathcal{P}})\right] - \left[\hat{L}_n(\tilde{\mathcal{P}}) - e_\sigma + 2e_I(\tilde{\mathcal{P}})\right]$$

$$= \underbrace{\hat{L}_n(\hat{\mathcal{P}}) - \hat{L}_n(\tilde{\mathcal{P}})}_{\leq 0} + 2\left[e_I(\hat{\mathcal{P}}) - e_I(\tilde{\mathcal{P}})\right]$$

$$\leq 2\left[e_I(\hat{\mathcal{P}}) - e_I(\tilde{\mathcal{P}})\right]. \qquad (19)$$

We denote

$$J(\hat{\mathcal{P}}, \tilde{\mathcal{P}}) = e_I(\hat{\mathcal{P}}) - e_I(\tilde{\mathcal{P}}) = \left\langle \mathcal{P}_{Z_i}^* V - V(X_i'), (\tilde{\mathcal{P}}_{Z_i} - \hat{\mathcal{P}}_{Z_i})V \right\rangle_n,$$

$$\triangleq \frac{1}{n}\sum_{i=1}^n \left[\mathcal{P}_{Z_i}^* V - V(X_i')\right] \cdot \left[(\tilde{\mathcal{P}}_{Z_i} - \hat{\mathcal{P}}_{Z_i})V\right]. \qquad (20)$$

Therefore by choosing $\mathcal{P} = \hat{\mathcal{P}}$ in (18) and using (19) as the upper bound for the empirical excess error, we get that

$$L(\hat{\mathcal{P}}) - L(\tilde{\mathcal{P}}) \le 4J(\hat{\mathcal{P}}, \tilde{\mathcal{P}}) + \frac{2c_1}{B} r^*(\mathcal{G}) + \frac{(88V_{\max}^2 + 2c_2 B)\ln(\frac{1}{\delta_1})}{n},$$

with probability at least $1 - \delta_1$.

We now turn to verifying the conditions of Theorem 5, required for the validity of (18). Condition (40) requires us to find a functional $T$ that satisfies the variance condition $\operatorname{Var}[f] \le T(f) \le B\mathbb{E}[f]$. Note that for any $g \in \mathcal{G}$ (17), we have

$$
\begin{aligned}
|g| = |l_\mathcal{P} - l_{\tilde{\mathcal{P}}}| &= \left| |(\mathcal{P}_z - \mathcal{P}_z^*)V|^2 - |(\tilde{\mathcal{P}}_z - \mathcal{P}_z^*)V|^2 \right| \\
&= \left| (\mathcal{P}_z - \tilde{\mathcal{P}}_z)V \right| \cdot \left| (\mathcal{P}_z + \tilde{\mathcal{P}}_z - 2\mathcal{P}_z^*)V \right| \le 4V_{\max} \left| (\mathcal{P}_z - \tilde{\mathcal{P}}_z)V \right|.
\end{aligned}
$$

We upper bound the variance of $g$ using the previous inequality and Proposition 7 in Appenix A.3 as follows

$$\operatorname{Var}[g] \le \mathbb{E}\left[g^2\right] \le 16V_{\max}^2 \mathbb{E}\left[ \left| (\mathcal{P}_Z - \tilde{\mathcal{P}}_Z)V \right|^2 \right] \le 16V_{\max}^2 (L(\mathcal{P}) - L(\tilde{\mathcal{P}})) = 16V_{\max}^2 \mathbb{E}[g].$$

So we may choose

$$
\begin{aligned}
T(g) &= 16V_{\max}^2 \mathbb{E}\left[ |(\mathcal{P}_z - \tilde{\mathcal{P}}_z)V|^2 \right], \\
B &= 16V_{\max}^2.
\end{aligned}
\tag{21}
$$

Therefore, with probability at least $1 - \delta_1$, it holds that

$$L(\hat{\mathcal{P}}) - L(\tilde{\mathcal{P}}) \le 4J(\hat{\mathcal{P}}, \tilde{\mathcal{P}}) + \frac{2c_1}{16V_{\max}^2} r^*(\mathcal{G}) + \frac{(88V_{\max}^2 + 32c_2 V_{\max}^2)\ln(\frac{1}{\delta_1})}{n}. \tag{22}$$

It remains to provide some upper bounds for $J(\hat{\mathcal{P}}, \tilde{\mathcal{P}})$ and $r^*(\mathcal{G})$. Proposition 10 provides an upper bound for the local Rademacher complexity, which we shall evoke shortly. So next we focus on $J(\hat{\mathcal{P}}, \tilde{\mathcal{P}})$.

Consider the inner product term $J(\hat{\mathcal{P}}, \tilde{\mathcal{P}})$ (20). Let $W_i = \mathcal{P}_{Z_i}^* V - V(X_i')$ and $h(z) = (\tilde{\mathcal{P}}_z - \mathcal{P}_z)V$ and define the function space

$$\mathcal{H} = \left\{ z \mapsto (\tilde{\mathcal{P}}_z - \mathcal{P}_z)V \; : \; \mathcal{P} \in \mathcal{M} \right\}. \tag{23}$$

With this notation, we have $J(\hat{\mathcal{P}}, \tilde{\mathcal{P}}) = \left\langle W_i, \hat{h}(Z_i) \right\rangle_n$ with $\hat{h}$ being the function in $\mathcal{H}$ corresponding to the choice of $\mathcal{P} = \hat{\mathcal{P}}$. Lemma 9 in Appendix B.2, which is a modulus of continuity result for weighted sums, upper bounds this inner product. We first verify its conditions.

By construction $W_i$ are zero-mean random variables, and they are bounded too:

$$|W_i| \le 2V_{\max}.$$

Also the assumed metric entropy condition on $\mathcal{M}$ implies a similar metric entropy on $\mathcal{H}$: Proposition 11 in Appendix B.4 shows that

$$\log \mathcal{N}\left(\varepsilon, \mathcal{M}, L_2(z_{1:n})\right) \le c \left( \frac{R}{\varepsilon} \right)^{2\alpha} \Rightarrow \log \mathcal{N}\left(\varepsilon, \mathcal{H}, L_2(z_{1:n})\right) \le c \left( \frac{V_{\max} R}{\varepsilon} \right)^{2\alpha}$$

Therefore, Lemma 9 shows that for any fixed $\delta_2 > 0$, we have

$$\frac{J(\hat{\mathcal{P}}, \tilde{\mathcal{P}})}{\left\| (\tilde{\mathcal{P}}_{Z_i} - \hat{\mathcal{P}}_{Z_i})V \right\|_n^{1-\alpha}} \le \sup_{h \in \mathcal{H}} \frac{\langle W_i, h(Z_i) \rangle_n}{\|h\|_n^{1-\alpha}} \le c(\alpha) \left[ V_{\max}^{1+\alpha} + (V_{\max} R)^\alpha \right] \sqrt{\frac{\ln(1/\delta_2)}{n}},$$

with probability at least $1 - \delta_2$. After re-arranging, we get that, with the same probability,

$$J(\hat{\mathcal{P}}, \tilde{\mathcal{P}}) \leq c(\alpha) \left[ V_{\max}^{1+\alpha} + (V_{\max} R)^\alpha \right] \left\| (\hat{\mathcal{P}}_{Z_i} - \tilde{\mathcal{P}}_{Z_i}) V \right\|_n^{1-\alpha} \sqrt{\frac{\ln(1/\delta_2)}{n}}. \tag{24}$$

In the RHS, we have the empirical norm $\left\| (\hat{\mathcal{P}}_{Z_i} - \tilde{\mathcal{P}}_{Z_i}) V \right\|_n$, which is not desirable as it has $\hat{\mathcal{P}}$ in it. To handle that term, we first relate it to the true norm, and then upper bound it by the excess error $L(\hat{\mathcal{P}}) - L(\tilde{\mathcal{P}})$. Afterwards we can solve the resulting inequality for the excess error.

We use Theorem 5 again to relate the empirical norm to the true norm. To prepare for the application of that theorem, define the function space

$$\mathcal{F} = \left\{ z \mapsto \left| (\mathcal{P}_z - \tilde{\mathcal{P}}_z) V \right|^2 : \mathcal{P} \in \mathcal{M} \right\}. \tag{25}$$

It is easy to see that for any $f \in \mathcal{F}$,

$$\operatorname{Var}[f] \leq \mathbb{E}\left[ f^2 \right] = \mathbb{E}\left[ \left| (\mathcal{P}_z - \tilde{\mathcal{P}}) V \right|^4 \right] \leq 4 V_{\max}^2 \mathbb{E}[f].$$

So we can choose

$$B = 4 V_{\max}^2,$$
$$T(f) = 4 V_{\max}^2 \mathbb{E}[f], \tag{26}$$

in the statement of that theorem.

Assuming that we can find a sub-root function $\psi$ that satisfies the condition (41) of the theorem and has the fixed point of $r^*(\mathcal{F})$, which we prove and provide an explicit value of in Proposition 10 in Appendix B.3, we have that for any fixed $\delta_3 > 0$,

$$\left\| (\hat{\mathcal{P}}_{Z_i} - \tilde{\mathcal{P}}_{Z_i}) V \right\|_n^2 \leq 2 \mathbb{E}\left[ \left| (\hat{\mathcal{P}} - \tilde{\mathcal{P}}) V \right|^2 \right] + \frac{c_1}{4 V_{\max}^2} r^*(\mathcal{F}) + \frac{(11 \times 4 V_{\max}^2 + c_2 \times 4 V_{\max}^2) \ln(1/\delta_3)}{n},$$

with probability at least $1 - \delta_3$.

We substitute this upper bound in (24) and apply Proposition 7 to upper bound $\mathbb{E}\left[ \left| (\hat{\mathcal{P}} - \tilde{\mathcal{P}}) V \right|^2 \right]$ by $L(\hat{\mathcal{P}}) - L(\tilde{\mathcal{P}})$. To shorten the expressions, we denote $c(\alpha) \left[ V_{\max}^{1+\alpha} + (V_{\max} R)^\alpha \right]$ by $c(\alpha, V_{\max}, R)$ and $\sqrt{\frac{\ln(1/\delta_2)}{n}}$ by $t_n$. We have

$$J(\hat{\mathcal{P}}, \tilde{\mathcal{P}}) \leq c(\alpha, V_{\max}, R) \left\| (\hat{\mathcal{P}}_{Z_i} - \tilde{\mathcal{P}}_{Z_i}) V \right\|_n^{1-\alpha} t_n$$

$$\leq c(\alpha, V_{\max}, R) t_n \times$$

$$\left[ 2 \mathbb{E}\left[ \left| (\hat{\mathcal{P}} - \tilde{\mathcal{P}}) V \right|^2 \right] + \frac{c_1}{4 V_{\max}^2} r^*(\mathcal{F}) + \frac{(44 V_{\max}^2 + c_2 \times 4 V_{\max}^2) \ln(1/\delta_3)}{n} \right]^{\frac{1-\alpha}{2}}$$

$$\leq c(\alpha, V_{\max}, R) t_n \times$$

$$\left[ 2 \left( L(\hat{\mathcal{P}}) - L(\tilde{\mathcal{P}}) \right) + \frac{c_1}{4 V_{\max}^2} r^*(\mathcal{F}) + \frac{(88 V_{\max}^2 + c_2 \times 4 V_{\max}^2) \ln(1/\delta_3)}{n} \right]^{\frac{1-\alpha}{2}},$$

with probability at least $1 - (\delta_2 + \delta_3)$.

Substituting this inequality in the upper bound on the excess risk $L(\hat{\mathcal{P}}) - L(\tilde{\mathcal{P}})$ (22), we obtain

$$L(\hat{\mathcal{P}}) - L(\tilde{\mathcal{P}}) \leq 4 c(\alpha, V_{\max}, R) t_n \left[ 2 \left( L(\hat{\mathcal{P}}) - L(\tilde{\mathcal{P}}) \right) + \right.$$

$$\left. \frac{c_1}{4 V_{\max}^2} r^*(\mathcal{F}) + \frac{(44 V_{\max}^2 + c_2 \times 4 V_{\max}^2) \ln(1/\delta_3)}{n} \right]^{\frac{1-\alpha}{2}} +$$

$$\frac{2 c_1}{16 V_{\max}^2} r^*(\mathcal{G}) + \frac{(88 V_{\max}^2 + 32 c_2 V_{\max}^2) \ln(\frac{1}{\delta_1})}{n}, \tag{27}$$

with probability at least $1 - (\delta_1 + \delta_2 + \delta_3)$.

Notice that the excess loss $E \triangleq L(\hat{\mathcal{P}}) - L(\tilde{\mathcal{P}})$ appears on both sides of the inequality, so we have to solve it in order to obtain an explicit upper bound on $E$. This can be done by considering several cases in which each term dominates the others, and solving the inequality for that case, and then summing up the upper bounds to obtain an overall upper bound for $E$. The detailed calculations are delegated to Proposition 12 in Appendix B.5. If we fix $\delta > 0$ and let $\delta_1 = \delta_2 = \delta_3 = \delta/3$, the proposition shows that

$$
\begin{aligned}
E &\leq c_3(\alpha) V_{\max}^2 R^{\frac{2\alpha}{1+\alpha}} \left[ \frac{1}{n^{\frac{1}{1+\alpha}}} + \frac{\log n}{n^{1-\frac{\alpha}{2}}} \right] \sqrt{\ln(1/\delta)} \\
&\leq c_4(\alpha) V_{\max}^2 R^{\frac{2\alpha}{1+\alpha}} \frac{\sqrt{\ln(1/\delta)}}{n^{\frac{1}{1+\alpha}}},
\end{aligned}
$$

with probability at least $1 - \delta$. The last inequality is because $n^{\frac{-1}{1+\alpha}}$ dominates $n^{-(1-\frac{\alpha}{2})} \log n$ as $n$ grows. But notice that when $\alpha$ is very small, this happens for large $n$, so as a result $c_4(\alpha)$ would be large. To finish the proof, recall that $E = L(\hat{\mathcal{P}}) - L(\tilde{\mathcal{P}}) = L(\hat{\mathcal{P}}) - \inf_{\mathcal{P} \in \mathcal{M}} L(\tilde{\mathcal{P}})$. $\qquad \square$

This result upper bounds the error of $\hat{\mathcal{P}}$ in approximating the next-state expectation of the value function. The upper bound has the model (or function) approximation error (the first term) and the estimation error (the second term). It is notable that the constant in front of the model approximation error is one, so the best we can hope from this algorithm, in the limit of infinite data, is as good as the best model in the model class $\mathcal{M}$.

The estimation error behaves as $n^{-1/(1+\alpha)}$ ($0 < \alpha < 1$). This is a fast rate and can reach $n^{-1}$ whenever $\alpha \to 0$. We do not know whether this is an optimal rate for this particular problem, but results from regression with least-squares loss suggest that this might indeed be optimal: For a regression function belonging to a function space that has a packing entropy in the same form as in the upper bound of Assumption A2, the rate $\Omega(n^{-1/(1+\alpha)})$ is its minimax lower bound [Yang and Barron, 1999].

We use local Rademacher complexity and analyze the modulus of continuity of empirical processes to obtain rates faster than what could be achieved by more conventional techniques of analyzing the supremum of the empirical processes. Farahmand et al. [2017a] used the supremum of the empirical process to analyze VAML and obtained $n^{-1/2}$ rate, which is slower than $n^{-1/(1+\alpha)}$. Notice that the loss functions of VAML and IterVAML are different, so this is only an approximate comparison. The rate $n^{-1/2}$, however, is common in the supremum of empirical process-based analysis, so we would expect it to hold if we used those techniques to analyze IterVAML. Finally notice that the error rate of VAML is not necessarily slower than IterVAML's; the present difference is at least somehow due to the shortcoming of their simpler proof technique.

## 4.2 Error Propagation

We analyze how the errors incurred at each step of IterVAML propagate throughout the iterations and affect the quality of the outcome. For policy evaluation, the quality is defined as the difference between $V^\pi$ and $\hat{V}_K$, weighted according to a user-defined probability distribution $\rho_{\mathcal{X}} \in \bar{\mathcal{M}}(\mathcal{X})$, i.e., $\|V^\pi - \hat{V}_K\|_{1,\rho_{\mathcal{X}}}$. For the control case we consider the performance loss, which is defined as the difference between the value of following the greedy policy w.r.t. $\hat{Q}_K$ compared to the value of the optimal policy $Q^*$, weighted according to a user-defined probability distribution $\rho \in \bar{\mathcal{M}}(\mathcal{X} \times \mathcal{A})$, i.e., $\rho(Q^* - Q^{\pi_K})$ (cf. Algorithm 1). This type of error propagation analysis has been performed before by Munos [2007], Antos et al. [2008], Farahmand et al. [2010], Scherrer et al. [2012], Huang et al. [2015], Mann et al. [2015], Farahmand et al. [2016c]. We compare some of them with our results later.

Recall that there are two sources of errors in the IterVAML procedure. The first is the error in model learning, which is caused because the model $\hat{\mathcal{P}}^{(k+1)}$ learned by solving the minimization problem (11) only satisfies $\hat{\mathcal{P}}^{(k+1)} \hat{V}_k \approx \mathcal{P}^* \hat{V}_k$ instead of being exact. This error is studied in Section 4.1, and Theorem 1 provides an upper bound on it.

The second source of error is that the AVI performs the Bellman update only approximately. So instead of having $\hat{Q}_{k+1} = T^*_{\hat{\mathcal{P}}(k)} \hat{Q}_k$ (or its policy evaluation equivalent), the function $\hat{Q}_{k+1}$ obtained by solving (12) is only approximately equal to $T^*_{\hat{\mathcal{P}}(k)} \hat{Q}_k$. As already mentioned, this step is essentially solving a regression problem. Hence, many of the standard error guarantees for regression can be used here too, with possibly some minor changes.[4]

We first study the *policy evaluation* scenario, which is simpler to analyze and the results are more accesible. Afterwards, we turn to the control problem. In order to reduce the clutter, we may drop the dependence of the Bellman (optimality) operator on $\mathcal{P}^*$, and use $T^*$ and $T^\pi$ (and alike for other policies) instead of $T^*_{\mathcal{P}^*}$ and $T^\pi_{\mathcal{P}^*}$, etc. Likewise, we may use $\mathcal{P}$ (and its variants) instead of $\mathcal{P}^*$.

Define

$$\varepsilon_k = T^\pi_{\hat{\mathcal{P}}(k+1)} \hat{V}_k - \hat{V}_{k+1}, \qquad k = 0, 1, \ldots, K-1$$
$$e_k = \left( \mathcal{P}^* - \hat{\mathcal{P}}^{(k+1)} \right) \hat{V}_k. \qquad k = 0, 1, \ldots, K-1 \tag{28}$$

The function $\varepsilon_k$ is the error due to the inexact application of the Bellman operator; this is the regression error. If $\hat{V}_{k+1} \leftarrow T^\pi_{\hat{\mathcal{P}}(k+1)} \hat{V}_k$, this error would be zero. The function $e_k$ is the error caused by the model error, when applied to $\hat{V}_k$. Theorem 1 provides an upper bound on the $L_2(\nu_{\mathcal{X}})$-norm of this error, where $\nu_{\mathcal{X}} \in \bar{\mathcal{M}}(\mathcal{X})$ is the distribution used to generate data.[5]

Let us also define $\Delta_k = \varepsilon_k + \gamma e_k$. The key observation is that for any $k = 0, 1, \ldots, K-1$, we have

$$
\begin{aligned}
\hat{V}_{k+1} &= T^\pi_{\hat{\mathcal{P}}(k+1)} \hat{V}_k - \varepsilon_k \\
&= T^\pi_{\hat{\mathcal{P}}(k+1)} \hat{V}_k - T^\pi_{\mathcal{P}^*} \hat{V}_k + T^\pi_{\mathcal{P}^*} \hat{V}_k - \varepsilon_k \\
&= \gamma \left( \hat{\mathcal{P}}^{(k+1)} - \mathcal{P}^* \right) \hat{V}_k + T^\pi_{\mathcal{P}^*} \hat{V}_k - \varepsilon_k \\
&= T^\pi_{\mathcal{P}^*} \hat{V}_k - (\varepsilon_k + \gamma e_k) = T^\pi_{\mathcal{P}^*} \hat{V}_k - \Delta_k.
\end{aligned}
$$

This equality relates $\hat{V}_{k+1}$, generated by the algorithm, to the ideal value that could be obtained if we applied the true Bellman operator $T^\pi_{\mathcal{P}^*}$ to $\hat{V}_k$.

This error decomposition and the fact that $V^\pi = T^\pi V^\pi$ allow us to write

$$
\begin{aligned}
V^\pi - \hat{V}_{k+1} &= T^\pi V^\pi - \hat{V}_{k+1} \\
&= T^\pi V^\pi - T^\pi \hat{V}_k + T^\pi \hat{V}_k - \hat{V}_{k+1} \\
&= \gamma \mathcal{P}^\pi \left( V^\pi - \hat{V}_k \right) + T^\pi \hat{V}_k - \left( T^\pi \hat{V}_k - \Delta_k \right) \\
&= \gamma \mathcal{P}^\pi \left( V^\pi - \hat{V}_k \right) + \Delta_k.
\end{aligned}
$$

By induction, we get[6]

$$V^\pi - \hat{V}_K = \sum_{k=0}^{K-1} (\gamma \mathcal{P}^\pi)^{K-1-k} \Delta_k + (\gamma \mathcal{P}^\pi)^K \left( V^\pi - \hat{V}_0 \right).$$

This equality shows how the error sequence $(\Delta_k)_k$ affects the final outcome. We observe that the error $\Delta_k$ goes through $K - 1 - k$ steps of the dynamics $\mathcal{P}^\pi$, discounted by $\gamma^{K-1-k}$. This means that the contribution of the error $\Delta_k$ at a particular state $x \in \mathcal{X}$ comes from the value of $\Delta_k$ at the $K - 1 - k$ steps ahead. Moreover, we observe that the effect of the errors in earlier iterations of IterVAML are discounted more.

We are often interested in an average error over the state space that is weighted according to a user-defined probability distribution $\rho_\mathcal{X} \in \bar{\mathcal{M}}(\mathcal{X})$. We would like to relate it to the norm of the errors $\Delta_k$ and some properties of the MDP. The norm of $\Delta_k$ may be measured according to another probability distribution $\nu_\mathcal{X} \in \bar{\mathcal{M}}(\mathcal{X})$, which in general is different from $\rho_\mathcal{X}$. For instance, $\nu_\mathcal{X}$ might be the distribution of the data $\mathcal{D}_n$, which can be different from the evaluation distribution $\rho_\mathcal{X}$ (this discrepancy would be typical under the off-policy sampling).

To obtain such a result, we first take the absolute value of both sides and then apply $\rho_\mathcal{X}$ to get

$$\left\| V^\pi - \hat{V}_K \right\|_{1,\rho_\mathcal{X}} \leq \sum_{k=0}^{K-1} \gamma^{K-1-k} \rho_\mathcal{X} (\mathcal{P}^\pi)^{K-1-k} |\Delta_k| + \gamma^K \rho_\mathcal{X} (\mathcal{P}^\pi)^K \left| V^\pi - \hat{V}_0 \right|. \qquad (29)$$

We want to relate $\rho_\mathcal{X} (\mathcal{P}^\pi)^k |\Delta_k|$ to a norm of $\Delta_k$ (and therefore, norms of $\varepsilon_k$ and $e_k$) and some properties of the MDP. Before providing a more general result, we consider the illustrating case when $\rho_\mathcal{X}^\pi$ is the stationary distribution of $\mathcal{P}^\pi$, i.e., $\rho_\mathcal{X}^\pi \mathcal{P}^\pi = \rho_\mathcal{X}^\pi$. For the stationary distribution, we have $\rho_\mathcal{X}^\pi (\mathcal{P}^\pi)^k = \rho_\mathcal{X}^\pi$ for any $k \geq 1$, so we obtain

$$\begin{aligned}
\left\| V^\pi - \hat{V}_K \right\|_{1,\rho_\mathcal{X}^\pi} &\leq \sum_{k=0}^{K-1} \gamma^{K-1-k} \rho_\mathcal{X}^\pi |\Delta_k| + \gamma^K \rho_\mathcal{X}^\pi \left| V^\pi - \hat{V}_0 \right| \\
&\leq \sum_{k=0}^{K-1} \gamma^{K-1-k} \left( \|\varepsilon_k\|_{1,\rho_\mathcal{X}^\pi} + \gamma \|e_k\|_{1,\rho_\mathcal{X}^\pi} \right) + \gamma^K \left\| V^\pi - \hat{V}_0 \right\|_{1,\rho_\mathcal{X}^\pi} \\
&\leq \sum_{k=0}^{K-1} \gamma^{K-1-k} \left( \|\varepsilon_k\|_{2,\rho_\mathcal{X}^\pi} + \gamma \|e_k\|_{2,\rho_\mathcal{X}^\pi} \right) + \gamma^K \left\| V^\pi - \hat{V}_0 \right\|_{2,\rho_\mathcal{X}^\pi}. \qquad (30)
\end{aligned}$$

This result shows that the effect of the regression error $\|\varepsilon_k\|_{2,\rho_\mathcal{X}^\pi}$ and the modelling error $\|e_k\|_{2,\rho_\mathcal{X}^\pi}$ at each iteration is comparable, with the difference of having an extra $\gamma$ factor. Moreover, the errors at the $k$-th iteration influence the final error by being discounted proportional to $\gamma^{K-k}$. Therefore, the errors in latter iterations are contributing more to the value function error compared to the earlier ones. This suggests spending more resources (e.g., number of samples, the expressiveness of function approximation architecture, computation, etc.) on latter iterations. This observation aligns the analysis for AVI and API by Farahmand et al. [2010], which is not surprising as IterVAML is a particular model-based version of AVI.

The inequality (30) measures the errors $\varepsilon_k$ and $e_k$ w.r.t. the stationary distribution $\rho_\mathcal{X}^\pi$. In order to obtain a more general result in which the norms of $\varepsilon_k$ and $e_k$ are w.r.t. another distribution $\nu_\mathcal{X}$, we can use a change of measure argument. Recall that for a measurable function $f : \mathcal{X} \to \mathbb{R}$ and two probability measures $\mu_1, \mu_2 \in \bar{\mathcal{M}}(\mathcal{X})$, if $\mu_1$ is absolutely continuous w.r.t. $\mu_2$ ($\mu_1 \ll \mu_2$), the Radon-Nikydom (R-N) derivative $\frac{d\mu_1}{d\mu_2}$ exists, and we have

$$\int f(x) d\mu_1 = \int f(x) \frac{d\mu_1}{d\mu_2} d\mu_2 \leq \left\| \frac{d\mu_1}{d\mu_2} \right\|_\infty \int |f(x)| d\mu_2,$$

where $\|\frac{d\mu_1}{d\mu_2}\|_\infty = \sup_x |\frac{d\mu_1}{d\mu_2}(x)|$. This change of measure argument has been used before to analyze RL algorithms, e.g., by Kakade and Langford [2002], Munos [2007]. The supremum of the R-N derivative, however, is a crude measure of the difference between two distributions. One may provide a more refined analysis by considering the average of R-N derivative:

$$\begin{aligned}
\int f(x) d\mu_1 &= \int f(x) \frac{d\mu_1}{d\mu_2} d\mu_2 = \int f(x) d\mu_1 = \int f(x) \frac{d\mu_1}{d\mu_2} \sqrt{d\mu_2} \sqrt{d\mu_2} \\
&\leq \sqrt{\int \left| \frac{d\mu_1}{d\mu_2}(x) \right|^2 d\mu_2} \sqrt{\int f^2(x) d\mu_2} = \left\| \frac{d\mu_1}{d\mu_2} \right\|_{2,\mu_2} \|f\|_{2,\mu_2}, \qquad (31)
\end{aligned}$$

where we used the Cauchy-Schwarz inequality. Here the expected value of the R-N derivative w.r.t. the probability distribution $\mu_2$ appears, which can be significantly smaller than its supremum. Refer to Farahmand et al. [2010] for more discussions. Note than this argument can be generalized to any $(p,q)$-Hölder pairs of $\|\frac{\mathrm{d}\mu_1}{\mathrm{d}\mu_2}\|_{p,\mu_2}$ and $\|f\|_{q,\mu_2}$ by using Hölder's inequality instead of the Cauchy-Schwarz inequality. This has been done by Scherrer et al. [2012] to analyze Approximate Modified Policy Iteration. We define the following concentrability coefficient.

**Definition 1** (Expected Concentrability of the Future State Distribution—Policy Evaluation). *Given $\rho_{\mathcal{X}}, \nu_{\mathcal{X}} \in \bar{\mathcal{M}}(\mathcal{X})$, policy $\pi$, and an integer number $k \geq 0$, let $\rho(\mathcal{P}^\pi)^k$ denote the future state distribution obtained when the first state is distributed according $\rho$ and the agent follows $\mathcal{P}^\pi (= \mathcal{P}^{*\pi})$ for $k$ steps. Define*

$$c_{VI,\rho_{\mathcal{X}},\nu_{\mathcal{X}}}^\pi(k) = \left\| \frac{\mathrm{d}\rho_{\mathcal{X}}(\mathcal{P}^\pi)^k}{\mathrm{d}\nu_{\mathcal{X}}} \right\|_{2,\nu_{\mathcal{X}}} \triangleq \sqrt{\int \left| \frac{\mathrm{d}\rho_{\mathcal{X}}(\mathcal{P}^\pi)^k}{\mathrm{d}\nu_{\mathcal{X}}}(x) \right|^2 \mathrm{d}\nu_{\mathcal{X}}} \ .$$

*If the future state distribution is not absolutely continuous w.r.t. $\nu_{\mathcal{X}}$, we take $c_{VI,\rho_{\mathcal{X}},\nu_{\mathcal{X}}}^\pi(k) = \infty$.*

The application of (31) to inequality (29) and using the just defined concentrability coefficient lead to the following proposition.

**Proposition 2** (Error Propagation for Model-based AVI for Policy Evaluation). *Consider a sequence of $(\hat{V}_k)_{k=0}^K$ generated as described. Let $\varepsilon_k = T_{\hat{\mathcal{P}}^{(k+1)}}^\pi \hat{V}_k - \hat{V}_{k+1}$ (regression error) and $e_k = (\mathcal{P}^* - \hat{\mathcal{P}}^{(k+1)})\hat{V}_k$ (modelling error) for $k = 0, 1, \ldots, K-1$. Consider probability distributions $\rho_{\mathcal{X}}, \nu_{\mathcal{X}} \in \bar{\mathcal{M}}(\mathcal{X})$. We have*

$$\left\| V^\pi - \hat{V}_K \right\|_{1,\rho_{\mathcal{X}}} \leq \sum_{k=0}^{K-1} \gamma^{K-1-k} c_{VI,\rho_{\mathcal{X}},\nu_{\mathcal{X}}}^\pi(K-1-k) \left( \|\varepsilon_k\|_{2,\nu_{\mathcal{X}}} + \gamma \|e_k\|_{2,\nu_{\mathcal{X}}} \right) + \gamma^K \left\| V^\pi - \hat{V}_0 \right\|_{1,\nu_{\mathcal{X}}} .$$

We now provide a result for the model-based AVI for the control case, i.e., the Bellman *optimality* operator is used. The is the extension of the result of Farahmand et al. [2010], Farahmand [2011] for AVI to the case when a model is used and the model has some errors.

Consider a sequence of $\hat{Q}_0, \hat{Q}_1, \ldots, \hat{Q}_K$ with $\hat{Q}_{k+1} \approx T_{\hat{\mathcal{P}}^{(k+1)}}^* \hat{Q}_k$, with $\hat{\mathcal{P}}^{(k+1)}$ being an approximation of the true $\mathcal{P} \triangleq \mathcal{P}^*$. IterVAML, which consists of repeated solving of (11) and (12), is an example of a procedure that generates these $\hat{Q}_k$. The result, however, is more general and does not depend on the particular way $\hat{\mathcal{P}}^{(k+1)}$ and $\hat{Q}_{k+1}$ are produced.

We define the error functions

$$\varepsilon_k = T_{\hat{\mathcal{P}}^{(k+1)}}^* \hat{Q}_k - \hat{Q}_{k+1}, \qquad\qquad k = 0, 1, \ldots, K-1$$
$$e_k = \left( \mathcal{P}^* - \hat{\mathcal{P}}^{(k+1)} \right) \max_{a'} \hat{Q}_k(\cdot, a'), \qquad k = 0, 1, \ldots, K-1 \qquad (32)$$

which are similar to (28) with the difference that they are defined based on the Bellman optimality operator and the action-value function. Here $\max_{a'} \hat{Q}_k(\cdot, a')$ is the function $\hat{V}_k(x) = \max_{a'} \hat{Q}_k(x, a')$.

We define the following concentrability coefficients, similar to the coefficient introduced by Farahmand et al. [2010], which itself is a relaxation of the coefficient introduced by Munos [2007].

**Definition 2** (Expected Concentrability of the Future State-Action Distribution). *Given $\rho, \nu \in \bar{\mathcal{M}}(\mathcal{X} \times \mathcal{A})$, an integer number $k \geq 0$, and an arbitrary sequence of policies $(\pi_i)_{i=1}^m$, the distribution $\rho\mathcal{P}^{\pi_1} \cdots \mathcal{P}^{\pi_k}$ denotes the future state-action distribution obtained when the first state-action is distributed according $\rho$ and the agent follows the sequence of policies $\pi_1, \pi_2, \ldots, \pi_k$. Define*

$$\bar{c}_{VI,\rho,\nu}(k) = \sup_{\pi_1,\ldots,\pi_k} \left\| \frac{\mathrm{d}\rho\mathcal{P}^{\pi_1} \cdots \mathcal{P}^{\pi_k}}{\mathrm{d}\nu} \right\|_{2,\nu} .$$

*If the future state-action distribution $\rho\mathcal{P}^{\pi_1} \cdots \mathcal{P}^{\pi_k}$ is not absolutely continuous w.r.t. $\nu$, we take $\bar{c}_{VI,\rho,\nu}(k) = \infty$. Moreover, for a discount factor $0 \leq \gamma < 1$, define the discounted weighted average concentrability coefficient as*

$$\bar{C}(\rho, \nu) = (1-\gamma)^2 \sum_{k \geq 1} k\gamma^{k-1} \bar{c}_{VI,\rho,\nu}(k).$$

The definition of $\bar{c}_{\text{VI},\rho,\nu}(k)$ is a generalization of $c^{\pi}_{\text{VI},\rho_{\mathcal{X}},\nu_{\mathcal{X}}}$ in Definition 1 from a fixed policy $\pi$ to a worst case over a sequence of policies. The definition of $\bar{C}(\rho, \nu)$ is similar to the second order discounted future state distribution concentration coefficient of Munos [2007], with the main difference being that it is defined for the expectation of the R-N derivative instead of its supremum.

The following theorem is our main error propagation result. It can be seen as the generalization of the results of Farahmand et al. [2010], Munos [2007] to the case when we use a model that has an error, whereas the aforementioned papers are for model-free case (or when the model is exact). Because of this similarity, several steps of the proof are similar to theirs. We discuss the similarities and differences in more detail after the theorem.

**Theorem 3.** *Consider a sequence of action-value function $(\hat{Q}_k)_{k=0}^{K}$, and their corresponding $(\hat{V}_k)_{k=0}^{K}$, each of which is defined as $\hat{V}_k(x) = \max_a \hat{Q}_k(x, a)$. Suppose that the MDP is such that the expected rewards are $R_{max}$-bounded, and $\hat{Q}_0$ is initialized such that it is $V_{max} \leq \frac{R_{max}}{1-\gamma}$-bounded. Let $\varepsilon_k = T^*_{\hat{\mathcal{P}}^{(k+1)}} \hat{Q}_k - \hat{Q}_{k+1}$ (regression error) and $e_k = (\mathcal{P}^* - \hat{\mathcal{P}}^{(k+1)})\hat{V}_k$ (modelling error) for $k = 0, 1, \ldots, K - 1$. Let $\pi_K$ be the greedy policy w.r.t. $\hat{Q}_K$, i.e., $\pi_K(x) = argmax_{a \in \mathcal{A}} \hat{Q}(x, a)$ for all $x \in \mathcal{X}$. Consider probability distributions $\rho, \nu \in \bar{\mathcal{M}}(\mathcal{X} \times \mathcal{A})$. We have*

$$\|Q^* - Q^{\pi_K}\|_{1,\rho} \leq \frac{2\gamma}{(1-\gamma)^2} \left[ \bar{C}(\rho, \nu) \max_{0 \leq k \leq K-1} \left( \|\varepsilon_k\|_{2,\nu} + \gamma \|e_k\|_{2,\nu} \right) + 2\gamma^K R_{max} \right]$$

*Proof.* The key observation is that for any $k = 0, 1, \ldots, K - 1$, we have

$$
\begin{aligned}
\hat{Q}_{k+1} &= T^*_{\hat{\mathcal{P}}^{(k+1)}} \hat{Q}_k - \varepsilon_k \\
&= T^*_{\hat{\mathcal{P}}^{(k+1)}} \hat{Q}_k - T^*_{\mathcal{P}^*} \hat{Q}_k + T^*_{\mathcal{P}^*} \hat{Q}_k - \varepsilon_k \\
&= \gamma \left( \hat{\mathcal{P}}^{(k+1)} - \mathcal{P}^* \right) \hat{V}_k + T^*_{\mathcal{P}^*} \hat{Q}_k - \varepsilon_k \\
&= T^*_{\mathcal{P}^*} \hat{Q}_k - (\varepsilon_k + \gamma e_k) = T^*_{\mathcal{P}^*} \hat{Q}_k - \Delta_k,
\end{aligned}
\tag{33}
$$

where we denote $\varepsilon_k + \gamma e_k$ by $\Delta_k$, and used

$$
\begin{aligned}
&(T^*_{\hat{\mathcal{P}}^{(k+1)}} \hat{Q}_k)(x, a) - (T^*_{\mathcal{P}^*} \hat{Q}_k)(x, a) = \\
&\left[ r(x, a) + \gamma \int \hat{\mathcal{P}}^{(k+1)}(\mathrm{d}x'|x, a) \max_{a'} \hat{Q}_k(x', a') \right] - \left[ r(x, a) + \gamma \int \mathcal{P}^*(\mathrm{d}x'|x, a) \max_{a'} \hat{Q}_k(x', a') \right] \\
&= \gamma \int \left( \hat{\mathcal{P}}^{(k+1)}(\mathrm{d}x'|x, a) - \mathcal{P}^*(\mathrm{d}x'|x, a) \right) \hat{V}_k(x').
\end{aligned}
$$

For the rest of the proof, we may use $T^*$ instead of $T^*_{\mathcal{P}^*}$, and $\mathcal{P}$ instead of $\mathcal{P}^*$ to reduce clutter.

Consider the optimal policy $\pi^*$. Note that it satisfies $T^*Q^* = T^{\pi^*}Q^* = Q^*$. Also we have $T^*\hat{Q}_k \geq T^{\pi^*}\hat{Q}_k$ (and in fact for any policy, including $\pi^*$). These facts in addition to the error decomposition (33) show that

$$
\begin{aligned}
Q^* - \hat{Q}_{k+1} &= T^{\pi^*}Q^* - \hat{Q}_{k+1} \\
&= T^{\pi^*}Q^* - T^{\pi^*}\hat{Q}_k + \underbrace{T^{\pi^*}\hat{Q}_k - T^*\hat{Q}_k}_{\leq 0} + T^*\hat{Q}_k - \hat{Q}_{k+1} \\
&\leq \gamma\mathcal{P}^{\pi^*}(Q^* - \hat{Q}_k) + \Delta_k.
\end{aligned}
\tag{34}
$$

Likewise, notice that the greedy policy $\pi_k$ w.r.t. $\hat{Q}_k$ satisfies $T^*\hat{Q}_k = T^{\pi_k}\hat{Q}_k$, and $T^*Q^* \geq T^{\pi_k}Q^*$. These facts along with the previous error decomposition show that

$$
\begin{aligned}
Q^* - \hat{Q}_{k+1} &= T^*Q^* - \hat{Q}_{k+1} \\
&= \underbrace{T^*Q^* - T^{\pi_k}Q^*}_{\geq 0} + T^{\pi_k}Q^* - \hat{Q}_{k+1} \\
&\geq T^{\pi_k}Q^* - \hat{Q}_{k+1} = T^{\pi_k}Q^* - \left(T^*\hat{Q}_k - \Delta_k\right) \\
&= T^{\pi_k}Q^* - T^{\pi_k}\hat{Q}_k + \Delta_k \\
&= \gamma \mathcal{P}^{\pi_k}\left(Q^* - \hat{Q}_k\right) + \Delta_k.
\end{aligned}
\tag{35}
$$

By induction from (34),

$$
Q^* - \hat{Q}_K \leq \sum_{k=0}^{K-1} \gamma^{K-k-1}(\mathcal{P}^{\pi^*})^{K-k-1}\Delta_k + \gamma^K(\mathcal{P}^{\pi^*})^K(Q^* - \hat{Q}_0).
\tag{36}
$$

Likewise, from (35), we obtain

$$
Q^* - \hat{Q}_K \geq \sum_{k=0}^{K-1} \gamma^{K-1-k}\left(\mathcal{P}^{\pi_{K-1}}\cdots\mathcal{P}^{\pi_{k+1}}\right)\Delta_k + \gamma^K\left(\mathcal{P}^{\pi_{K-1}}\cdots\mathcal{P}^{\pi_0}\right)(Q^* - \hat{Q}_0).
\tag{37}
$$

We need to relate $Q^* - Q^{\pi_K}$ (the performance loss of following policy $\pi_K$) to the value function estimation error $Q^* - \hat{Q}_K$. We have

$$
\begin{aligned}
Q^* - Q^{\pi_K} &= T^{\pi^*}Q^* - T^{\pi_K}Q^{\pi_K} \\
&= T^{\pi^*}Q^* - T^{\pi^*}\hat{Q}_K + \underbrace{T^{\pi^*}\hat{Q}_K - T^*\hat{Q}_K}_{\leq 0} + \underbrace{T^*\hat{Q}_K}_{=T^{\pi_K}\hat{Q}_K} - T^{\pi_K}Q^{\pi_K} \\
&\leq \gamma\mathcal{P}^{\pi^*}\left(Q^* - \hat{Q}_K\right) + \gamma\mathcal{P}^{\pi_K}\left(\hat{Q}_K - Q^{\pi_K}\right) \\
&= \gamma\mathcal{P}^{\pi^*}\left(Q^* - \hat{Q}_K\right) + \gamma\mathcal{P}^{\pi_K}\left(\hat{Q}_K - Q^* + Q^* - Q^{\pi_K}\right) \\
&= \gamma\left(\mathcal{P}^{\pi^*} - \mathcal{P}^{\pi_K}\right)\left(Q^* - \hat{Q}_K\right) + \gamma\mathcal{P}^{\pi_K}\left(Q^* - Q^{\pi_K}\right).
\end{aligned}
$$

By rearranging, we obtain

$$
\left(\mathbf{I} - \gamma\mathcal{P}^{\pi_K}\right)\left(Q^* - Q^{\pi_K}\right) \leq \gamma\left(\mathcal{P}^{\pi^*} - \mathcal{P}^{\pi_K}\right)\left(Q^* - \hat{Q}_K\right).
$$

Here $\mathbf{I}$ is the identify operator. Therefore, we have

$$
\left(Q^* - Q^{\pi_K}\right) \leq \gamma\left(\mathbf{I} - \gamma\mathcal{P}^{\pi_K}\right)^{-1}\left(\mathcal{P}^{\pi^*} - \mathcal{P}^{\pi_K}\right)\left(Q^* - \hat{Q}_K\right).
\tag{38}
$$

We combine just obtained (38) with previous pointwise inequalities (36) and (37) and take the absolute value to get

$$
\begin{aligned}
Q^* - Q^{\pi_K} \leq \gamma\left(\mathbf{I} - \gamma\mathcal{P}^{\pi_K}\right)^{-1}\Bigg[ &\sum_{k=0}^{K-1} \gamma^{K-k-1}\left((\mathcal{P}^{\pi^*})^{K-k} + (\mathcal{P}^{\pi_K}\cdots\mathcal{P}^{\pi_{k+1}})\right)|\Delta_k| + \\
&\gamma^K\left((\mathcal{P}^{\pi^*})^{K+1} + (\mathcal{P}^{\pi_K}\cdots\mathcal{P}^{\pi_0})\right)\left|Q^* - \hat{Q}_0\right|\Bigg]
\end{aligned}
$$

To simplify the notation, we introduce

$$
\alpha_k = \begin{cases} \dfrac{(1-\gamma)\gamma^{K-k-1}}{1-\gamma^{K+1}}, & 0 \leq k < K \\ \dfrac{(1-\gamma)\gamma^K}{1-\gamma^{K+1}}. & k = K \end{cases}
$$

One may verify that $\sum_{k=0}^{K} \alpha_k = 1$. We also define stochastic kernels

$$A_k = \begin{cases} \frac{1-\gamma}{2} \left(\mathbf{I} - \gamma \mathcal{P}^{\pi_K}\right)^{-1} \left[(\mathcal{P}^{\pi^*})^{K-k} + (\mathcal{P}^{\pi_K} \cdots \mathcal{P}^{\pi_{k+1}})\right], & 0 \le k < K \\ \frac{1-\gamma}{2} \left(\mathbf{I} - \gamma \mathcal{P}^{\pi_K}\right)^{-1} \left[(\mathcal{P}^{\pi^*})^{K+1} + (\mathcal{P}^{\pi_K} \cdots \mathcal{P}^{\pi_0})\right]. & 0 \le k < K \end{cases}$$

Therefore, we have

$$Q^* - Q^{\pi_K} \le \frac{2(1-\gamma^{K+1})\gamma}{(1-\gamma)^2} \left[\sum_{k=0}^{K-1} \alpha_k A_k |\Delta_k| + \alpha_K A_K |Q^* - \hat{Q}_0|\right].$$

Let us call $\lambda_K = \frac{2(1-\gamma^{K+1})\gamma}{(1-\gamma)^2}$.

This last inequality is pointwise over the state-action space. To compute the expected performance loss, we apply the probability distribution $\rho \in \bar{\mathcal{M}}(\mathcal{X} \times \mathcal{A})$ to its both sides. We use Jensen's inequality to move $\rho$ inside. This is possible because the sequence $(\alpha_k)$ is positive and sums to one. We get

$$\rho |Q^* - Q^{\pi_K}| \le \lambda_K \left[\sum_{k=0}^{K-1} \alpha_k \rho A_k |\Delta_k| + \alpha_K \rho A_K |Q^* - \hat{Q}_0|\right]. \tag{39}$$

Let us focus on $\rho A_k |\Delta_k|$. We first expand the $(\mathbf{I} - \gamma \mathcal{P}^{\pi_K})^{-1}$ term. Afterwards, we do a change of measure argument, as was shown in (31), and use the definition of $\bar{c}_{\mathrm{VI},\rho,\nu}$ (Definition 2) to upper bound the expected R-N. derivative:

$$\rho A_k |\Delta_k| = \frac{1-\gamma}{2} \rho \sum_{m \ge 0} \gamma^m (\mathcal{P}^{\pi_K})^m \left[(\mathcal{P}^{\pi^*})^{K-k} + (\mathcal{P}^{\pi_K} \cdots \mathcal{P}^{\pi_{k+1}})\right] |\Delta_k|$$

$$\le (1-\gamma) \sum_{m \ge 0} \gamma^m \bar{c}_{\mathrm{VI},\rho,\nu}(K - k + m) \|\Delta_k\|_{2,\nu}.$$

We plug this upper bound back into (39), and then expand $\alpha_k$. We also benefit from the assumption that $|Q^* - \hat{Q}_0|$ is $2V_{\max}$-bounded, so $\rho A_K |Q^* - \hat{Q}_0| \le 2V_{\max}$. we get

$$\|Q^* - Q^{\pi_K}\|_{1,\rho} \le \frac{2(1-\gamma^{K+1})\gamma}{(1-\gamma)^2} \Bigg[\sum_{k=0}^{K-1} \alpha_k (1-\gamma) \sum_{m \ge 0} \gamma^m \bar{c}_{\mathrm{VI},\rho,\nu}(K - k + m) \|\Delta_k\|_{2,\nu} +$$

$$2\alpha_K V_{\max}\Bigg]$$

$$\le \frac{2\gamma}{1-\gamma} \left[(1-\gamma) \sum_{k=0}^{K-1} \sum_{m \ge 0} \gamma^{K-k-1+m} \bar{c}_{\mathrm{VI},\rho,\nu}(K - k + m) \|\Delta_k\|_{2,\nu} + 2\gamma^K V_{\max}\right].$$

To simplify the expression, we take the maximum of $\|\Delta_k\|_{2,\nu}$ over $k = 0, \ldots, K-1$, and take it out of the summation. One can also see that

$$\sum_{k=0}^{K-1} \sum_{m \ge 0} \gamma^{K-k-1+m} \bar{c}_{\mathrm{VI},\rho,\nu}(K - k + m) = \sum_{m \ge 0} \sum_{k'=1}^{K-1} \gamma^{k'-1+m} \bar{c}_{\mathrm{VI},\rho,\nu}(k' + m)$$

$$= \sum_{n \ge 1} n \gamma^{n-1} \bar{c}_{\mathrm{VI},\rho,\nu}(n) = \frac{\bar{C}(\rho,\nu)}{(1-\gamma)^2}.$$

So we get

$$\|Q^* - Q^{\pi_K}\|_{1,\rho} \le \frac{2\gamma}{(1-\gamma)^2} \left[\bar{C}(\rho,\nu) \max_{0 \le k \le K-1} \|\Delta_k\|_{2,\nu} + 2\gamma^K R_{\max}\right].$$

We obtain the desired result after substituting $\|\Delta_k\|_{2,\nu}$ with $\|\varepsilon_k\|_{2,\nu} + \gamma \|e_k\|_{2,\nu}$. $\qquad \square$

We describe the similarities and differences of this result with those of Munos [2007], Farahmand et al. [2010], Farahmand [2011]. The similarity between all these results is that they analyze AVI when the errors are measured by a norm other than the supremum norm: Here the size of the errors $e_k$ and $\varepsilon_k$ is measured according to their $L_2(\nu)$-norm, and the performance loss is the difference between $Q^*$ and $Q^{\pi_K}$ according to the $L_1(\rho)$ norm. Measuring the performance loss according to the $L_1(\rho)$ is a natural choice as it is the loss in the value of following policy $\pi_K$ comparing to the optimal one, when the initial state-action distribution is $\rho$. All these results are improvement over the supremum-based norms, which can be quite conservative. Refer to the previous work for more discussion. Similar $L_p$-norm-based analysis has been done for algorithms other than AVI. For example, refer to Munos [2003], Antos et al. [2008] for API, Scherrer et al. [2012] for approximate modified policy iteration, Farahmand et al. [2016c] for finite-horizon AVI, Huang et al. [2015] for approximate Maximum Entropy Inverse Optimal Control (or RL), which is solved using a variant of finite-horizon AVI that has a softmax instead of the max operator in the definition of the Bellman operator, and Mann et al. [2015] for AVI with options.

The main difference of this result compared to the previous work is that the theorem is suitable for MBRL when the models have errors, quantified by the $(e_k)$ sequence. On the other hand, Munos [2007], Farahmand et al. [2010], Farahmand [2011] analyze AVI when there is no modelling error, i.e., either the model is exactly known or we are in a model-free RL scenario and we do not need to deal with the model error. If the model error $e_k = 0$, the form of result is in fact quite similar to Theorem 5.2 of Munos [2007], with some subtle, but important, differences.

The difference is that the concentrability coefficients $\bar{c}_{\text{VI},\rho,\nu}(k)$ used in the definition of $\bar{C}(\rho,\nu)$ in this work is w.r.t. the expected value of the R-N derivative (see Definition 2). In the work of Munos [2007], the same quantity $\bar{C}(\rho,\nu)$ is defined w.r.t. the supremum norm of the R-N derivatives, i.e., $\sup_{\pi_1,\ldots,\pi_k} \|\frac{\mathrm{d}\rho\mathcal{P}^{\pi_1}\ldots\mathcal{P}^{\pi_k}}{\mathrm{d}\nu}\|_\infty$. As argued earlier here, and in more detail by Farahmand et al. [2010], Farahmand [2011], the expected value of R-N can be a significantly smaller than its supremum. This means that the current result captures the error propagation behaviour of AVI more accurately.

Some aspects of this result is more general compared to Farahmand et al. [2010], Farahmand [2011], but it does not consider some finer aspects of the error propagation as they do. The result is more general because it considers the modelling error, but the work of Farahmand et al. [2010], Farahmand [2011] does not. On the other hand, this result does not reveal the error decaying property of AVI: Errors in the earlier iterations of AVI are less important than the errors in later ones. This is hidden in the current result as we take the maximum error over all iterations in the upper bound, i.e., $\max_{0\le k\le K-1}(\|\varepsilon_k\|_{2,\nu} + \gamma\|e_k\|_{2,\nu})$. A more careful, but slightly more involved analysis, would reveal that the errors at all iteration do not contribute equality to the final performance loss. This is apparent in the results of Farahmand et al. [2010], Farahmand [2011]. We can also observe this behaviour in (30) and Proposition 2, which are for policy evaluation, where the errors at the $k$-th iterations are multiplied by $\gamma^{K-1-k}$. We decided not to present such an analysis in order to simplify the proof and the statement of the result.

Before stating Theorem 4, which is the direct implication of Theorems 1 and 3, we state another assumption.

**Assumption A4 (Value Function Space)** The value function space $\mathcal{F}^{|\mathcal{A}|}$ is $V_{\max}$-bounded with $V_{\max} \le \frac{R_{\max}}{1-\gamma}$, and $V_{\max} \ge 1$.

This assumption requires that all the value functions $\hat{Q}_k$ and $\hat{V}_k$ generated by performing a step of AVI (12) and used in the model learning steps (11) are $V_{\max}$-bounded. This ensures that Assumption A3, which is required by Theorem 1, is satisfied in all iterations. This assumption is easy to satisfy in practice by clipping the output of the value function estimator at the level of $\pm V_{\max}$. Theoretical analysis of such a clipped value function estimator, however, is more complicated. As we do not analyze the value function estimation steps of IterVAML, which depends on the choice of $\mathcal{F}^{|\mathcal{A}|}$, we ignore this issue.

**Theorem 4.** *Consider the IterVAML procedure in which at the $k$-th iteration the model $\hat{\mathcal{P}}^{(k+1)}$ is obtained by solving (11) and $\hat{Q}_{k+1}$ is obtained by solving (12). Let $\varepsilon_k = T^*_{\hat{\mathcal{P}}^{(k+1)}}\hat{Q}_k - \hat{Q}_{k+1}$ be the regression error. Suppose that Assumptions A1, A2, and A4 hold. Consider the greedy policy $\pi_K$ w.r.t. $\hat{Q}_K$. For any $\rho \in \bar{\mathcal{M}}(\mathcal{X} \times \mathcal{A})$, there exists a finite $c(\alpha) > 0$, depending only on $\alpha$, such that for any*

$\delta > 0$, with probability at least $1 - \delta$, we have

$$\|Q^* - Q^{\pi_K}\|_{1,\rho} \leq \frac{2\gamma}{(1-\gamma)^2} \left[ \bar{C}(\rho, \nu) \left( \max_{0 \leq k \leq K-1} \|\varepsilon_k\|_{2,\nu} + \gamma e_{model}(n) \right) + 2\gamma^K R_{max} \right]$$

where

$$e_{model}(n) = \sup_{V \in \mathcal{F}^+} \inf_{\mathcal{P} \in \mathcal{M}} \|(\mathcal{P}_z - \mathcal{P}_z^*)V\|_{2,\nu} + \frac{c(\alpha)V_{max}R^{\frac{\alpha}{1+\alpha}} \sqrt[4]{\log(K/\delta)}}{n^{\frac{1}{2(1+\alpha)}}},$$

and $\mathcal{F}^+ = \left\{ \max_a Q(\cdot, a) : Q \in \mathcal{F}^{|\mathcal{A}|} \right\}$.

*Proof.* Fix $0 < \delta < 1$. For each iteration $k = 0, \ldots, K-1$ of IterVAML, invoke Theorem 1 with the confidence parameter $\delta/K$ to obtain that

$$\|e_k\|_{2,\nu}^2 = \left\|(\hat{\mathcal{P}}_z^{(k+1)} - \mathcal{P}_z^*)\hat{V}_k\right\|_{2,\nu}^2 \leq \inf_{\mathcal{P} \in \mathcal{M}} \left\|(\mathcal{P}_z - \mathcal{P}_z^*)\hat{V}_k\right\|_{2,\nu}^2 + \frac{c(\alpha)V_{max}^2 R^{\frac{2\alpha}{1+\alpha}} \sqrt{\log(K/\delta)}}{n^{\frac{1}{1+\alpha}}},$$

for some $c(\alpha) > 0$ and with probability at least $1 - \delta/K$.

Since $\hat{V}_k$ is random, we upper bound the model approximation error term by taking the supremum over all possible value functions that might be encountered during IterVAML. Since any $\hat{Q}_k$ generated by IterVAML is a member of $\mathcal{F}^{|\mathcal{A}|}$, their corresponding $\hat{V}_k$s belong to $\mathcal{F}^+ = \left\{ \max_a Q(\cdot, a) : Q \in \mathcal{F}^{|\mathcal{A}|} \right\}$. Therefore, we can upper bound the model approximation error by

$$\sup_{V \in \mathcal{F}^+} \inf_{\mathcal{P} \in \mathcal{M}} \|(\mathcal{P}_z - \mathcal{P}_z^*)V\|_{2,\nu}.$$

We apply the union bound over all $k$ to obtain that all $k = 0, \ldots, K-1$ simultaneously satisfy

$$\|e_k\|_{2,\nu}^2 \leq \sup_{V \in \mathcal{F}^+} \inf_{\mathcal{P} \in \mathcal{M}} \|(\mathcal{P}_z - \mathcal{P}_z^*)V\|_{2,\nu}^2 + \frac{c(\alpha)V_{max}^2 R^{\frac{2\alpha}{1+\alpha}} \sqrt{\log(K/\delta)}}{n^{\frac{1}{1+\alpha}}},$$

with probability at least $1 - \delta$.

We can now apply Theorem 3 with this choice of $\|e_k\|_{2,\nu}$ and $\varepsilon_k$, as defined in the statement of the theorem, to obtain the desired result. $\square$

This result provides an upper bound on the quality of the learned policy $\pi_K$, as a function of the number of samples and the properties of the model space $\mathcal{M}$ and the MDP. The estimation error due to the model learning is $n^{\frac{1}{2(1+\alpha)}}$, which is discussed in some detail after Theorem 1. The model approximation error term $\sup_{V \in \mathcal{F}^+} \inf_{\mathcal{P} \in \mathcal{M}} \|(\mathcal{P}_z - \mathcal{P}_z^*)V\|_{2,\nu}$ shows the interaction between the model $\mathcal{M}$ and the value function space $\mathcal{F}^{|\mathcal{A}|}$. This quantity is likely to be conservative and can be improved. We also note that an upper bound on $\|\varepsilon_k\|_{2,\nu}$ depends on the regression method, the choice of $\mathcal{F}^{|\mathcal{A}|}$, and the number of samples generated from $\hat{\mathcal{P}}^{(k+1)}$.

## 5 Conclusion

We have introduced IterVAML, a decision-aware model-based RL algorithm. We proved finite sample error upper bound for the model learning procedure (Theorem 1) and a generic error propagation result for an approximate value iteration algorithm that uses an inaccurate model (Theorem 3). The consequence of these two results was Theorem 4, which provides an error upper bound guarantee on the quality of the outcome policy of IterVAML.

There are several possible future research directions. One is empirical studies of IterVAML and comparing it with non-decision-aware methods. Another direction is to investigate other approaches to decision-aware model-based RL algorithms.

# A  Background Definitions and Results

We collect some background definitions and useful results in this appendix. Section A.1 defines the notations related to the norms of the functions. Section A.2 recalls the definition of the covering number and metric entropy of a function space. We define the Rademacher complexity and collect some related results in Section A.3.

## A.1  Function Space and Norms

Here we briefly define some of the notations that we use throughout the paper. Consider the domain $\mathcal{X}$ and a function space $\mathcal{F} : \mathcal{X} \to \mathbb{R}$. We do not deal with measure theoretic considerations, so we just assume that all functions involved are measurable w.r.t. an appropriate $\sigma$-algebra for that space. We use $\bar{\mathcal{M}}(\mathcal{X})$ to refer to the set of all probability distributions defined over that space. We use symbols such as $\nu, \rho, \mu \in \bar{\mathcal{M}}(\mathcal{X})$ to refer to probability distributions defined over that space. In order to emphasize that the probability distribution is defined over a specific space such as $\mathcal{X}$, we sometimes use the subscript, e.g., $\nu_{\mathcal{X}}, \rho_{\mathcal{X}}, \mu_{\mathcal{X}}$, etc.

We use $\|f\|_{p,\nu}$ to denote the $L_p(\nu)$-norm ($1 \leq p < \infty$) of a measurable function $f : \mathcal{X} \to \mathbb{R}$, i.e.,

$$\|f\|_{p,\nu}^p \triangleq \int_{\mathcal{X}} |f(x)|^p \mathrm{d}\nu(x).$$

The supremum norm is defined as

$$\|f\|_{\infty} = \sup_{x \in \mathcal{X}} |f(x)|.$$

We may define empirical norms too. Let $x_1, \ldots, x_n$ be a sequence of points in $\mathcal{X}$. We use $x_{1:n}$ to refer to this sequence. The empirical measure $P_n$ is the probability measure that puts a mass of $\frac{1}{n}$ at each $x_i$, i.e.,

$$P_n = P_{x_{1:n}} = \frac{1}{n} \sum_{i=1}^{n} \delta_{x_i},$$

where $\delta_x$ is the Dirac's delta function. Equivalently, $P_n(A) = \frac{1}{n} \sum_{i=1}^{n} \mathbb{I}\{x_i \in A\}$ for any measurable subset $A \subset \mathcal{X}$, where $\mathbb{I}\{\cdot\}$ is the indicator function. For $\mathcal{D}_n = x_{1:n}$, the empirical $L_2(P_n)$-norm of function $f : \mathcal{X} \to \mathbb{R}$ is

$$\|f\|_{\mathcal{D}_n}^2 = \|f\|_{P_n}^2 = \int f \, \mathrm{d}P_n = \frac{1}{n} \sum_{i=1}^{n} |f(x_i)|^2 \,.$$

We can define other $L_p(P_n)$-norms, sometimes denoted by $L_p(x_{1:n})$, similarly. When there is no chance of confusion about $\mathcal{D}_n$, we may denote the empirical norm simply by $\|f\|_n$.

Throughout the paper we use $\mathcal{M}$ to denote a space of transition probability kernels. Given a transition probability kernel $\mathcal{P} \in \mathcal{M}$, define the $L_2$-norm at a point $z \in \mathcal{X} \times \mathcal{A}$ as

$$\|\mathcal{P}(\cdot|z)\|^2 = \|\mathcal{P}(\cdot|z)\|_2^2 = \int |\mathcal{P}(\mathrm{d}x'|z)|^2.$$

For a sequence $z_{1:n} \triangleq z_1, \ldots, z_n \subset \mathcal{X} \times \mathcal{A}$, we define the empirical norm $L_2(z_{1:n})$ of $\mathcal{P} \in \mathcal{M}$ by

$$\|\mathcal{P}\|_{2,n}^2 = \frac{1}{n} \sum_{i=1}^{n} \|\mathcal{P}(\cdot|z_i)\|_2^2 \,.$$

We also define the following mixed supremum-total variation distance between any $\mathcal{P}_1, \mathcal{P}_2 \in \mathcal{M}$:

$$d_{\infty,\mathrm{TV}}(\mathcal{P}_1, \mathcal{P}_2) = \sup_{z \in \mathcal{Z}} \int |\mathcal{P}_1(\mathrm{d}y|z) - \mathcal{P}_2(\mathrm{d}y|z)| \,.$$

## A.2  The Covering Number and Metric Entropy

We quote the definition of the covering number from Györfi et al. [2002].

**Definition 3** (Definition 9.3 of Györfi et al. 2002)**.** *Let $\varepsilon > 0$, $\mathcal{F}$ be a set of real-valued functions defined on $\mathcal{X}$, and $\nu_{\mathcal{X}}$ be a probability measure on $\mathcal{X}$. Every finite collection of $N_{\varepsilon} = \{f_1, \ldots, f_{N_{\varepsilon}}\}$ defined on $\mathcal{X}$ with the property that for every $f \in \mathcal{F}$, there is a function $f' \in N_{\varepsilon}$ such that $\|f - f'\|_{p, \nu_{\mathcal{X}}} < \varepsilon$ is called an $\varepsilon$-cover of $\mathcal{F}$ w.r.t. $\|\cdot\|_{p, \nu_{\mathcal{X}}}$.*

*Let $\mathcal{N}(\varepsilon, \mathcal{F}, \|\cdot\|_{p, \nu_{\mathcal{X}}})$ be the size of the smallest $\varepsilon$-cover of $\mathcal{F}$ w.r.t. $\|\cdot\|_{p, \nu_{\mathcal{X}}}$. If no finite $\varepsilon$-cover exists, take $\mathcal{N}(\varepsilon, \mathcal{F}, \|\cdot\|_{p, \nu_{\mathcal{X}}}) = \infty$. Then $\mathcal{N}(\varepsilon, \mathcal{F}, \|\cdot\|_{p, \nu_{\mathcal{X}}})$ is called an $\varepsilon$-covering number of $\mathcal{F}$ and $\log \mathcal{N}(\varepsilon, \mathcal{F}, \|\cdot\|_{p, \nu_{\mathcal{X}}})$ is called the metric entropy of $\mathcal{F}$ w.r.t. the same norm.*

*Given a $x_{1:n} = (x_1, \ldots, x_n) \subset \mathcal{X}$ and its corresponding empirical measure $P_n = P_{x_{1:n}}$, we can define the empirical covering number of $\mathcal{F}$ w.r.t. the empirical norm $\|\cdot\|_{p, x_{1:n}}$ and is denoted by $\mathcal{N}_p(\varepsilon, \mathcal{F}, x_{1:n}) = \mathcal{N}(\varepsilon, \mathcal{F}, L_p(x_{1:n}))$.*

## A.3  Rademacher Complexity and Error Upper Bound

We briefly define Rademacher complexity, and its local variant, and quote a result regarding the deviation of the expectation from the empirical expectation from Bartlett et al. [2005]. For more information about Rademacher complexity, refer to Bartlett et al. [2005], Bartlett and Mendelson [2002].

Let $\sigma_1, \ldots, \sigma_n$ be independent random variables with $\mathbb{P}\{\sigma_i = 1\} = \mathbb{P}\{\sigma_i = -1\} = 1/2$. For a (measurable) function space $\mathcal{F} : \mathcal{X} \to \mathbb{R}$, define

$$R_n \mathcal{F} = \sup_{f \in \mathcal{F}} \frac{1}{n} \sum_{i=1}^{n} \sigma_i f(X_i),$$

with $X_i \sim \nu$. The Rademacher complexity (or average) of $\mathcal{F}$ is $\mathbb{E}[R_n \mathcal{G}]$, in which the expectation is w.r.t. both $\sigma$ and $X_i$. One may interpret it as a complexity measure that quantifies the extent that a function from $\mathcal{F}$ can fit a noise sequence of length $n$ [Bartlett and Mendelson, 2002].

The error behaviour of an estimator can be captured by the behaviour of the supremum of an empirical process, i.e.,

$$\sup_{f \in \mathcal{F}} (\mathbb{E}[f] - \mathbb{E}_n[f]),$$

in which $\mathbb{E}_n[f] \triangleq \frac{1}{n} \sum_{i=1}^{n} f(X_i)$. Rademacher complexity is closely related to the supremum of the empirical process, through the relation between the empirical process and its symmetrized process, so studying them leads to error upper bounds for the estimator, see e.g., Theorem 8 of Bartlett and Mendelson [2002] or Theorem 2.1 of Bartlett et al. [2005].

Nonetheless, the supremum of the empirical process may not accurately capture the behaviour of an estimator. The error behaviour of an estimator does not depends on the supremum of $\mathbb{E}[f] - \mathbb{E}_n[f]$ over the whole $\mathcal{F}$, but only around a local neighbourhood of a certain point $f_0 \in \mathcal{F}$, e.g., the minimizer of $\sup_{f \in \mathcal{F}} \mathbb{E}[f]$. This local behaviour is better captured by studying the modulus of continuity of an empirical process. This in turn is related to local Rademacher complexity, which we use in this work.

We quote the first part of Theorem 3.3 of Bartlett et al. [2005], almost verbatim, which provides an error upper bound on the difference between the expectation and empirical expectation as a function of local Rademacher complexity of the class. Before stating the result, recall the definition of a sub-root function: A non-negative and non-decreasing function $\psi : [0, \infty) \to [0, \infty)$ is called sub-root if $r \mapsto \frac{\psi(r)}{\sqrt{r}}$ is non-increasing for $r > 0$ [Bartlett et al., 2005].

**Theorem 5** (Theorem 3.3 of Bartlett et al. [2005]—First Part)**.** *Let $\mathcal{F}$ be a class of functions with ranges in $[a, b]$ and assume that there are some functional $T : \mathcal{F} \to \mathbb{R}^+$ and some constant $B$ such that for every $f \in \mathcal{F}$,*

$$\mathrm{Var}[f] \leq T(f) \leq B\mathbb{E}[f]. \tag{40}$$

*Let $\psi$ be a sub-root function and let $r^* = r^*(\mathcal{F})$ be the fixed point of $\psi$. Assume that for any $r \geq r^*$, $\psi$ satisfies*

$$\psi(r) \geq B\mathbb{E}[R_n\{f \in \mathcal{F} : T(f) \leq r\}]. \tag{41}$$

*Then with $c_1 = 704$ and $c_2 = 26$, for any $K > 1$ and every $x > 0$, with probability at least $1 - e^{-x}$, for any $f \in \mathcal{F}$, we have*

$$\mathbb{E}\left[f\right] \leq \frac{K}{K-1} \mathbb{E}_n\left[f\right] + \frac{c_1 K}{B} r^* + \frac{x(11(b-a) + c_2 BK)}{n}.$$

*Also with probability at least $1 - e^{-x}$, for any $f \in \mathcal{F}$, we have*

$$\mathbb{E}_n\left[f\right] \leq \frac{K+1}{K} \mathbb{E}\left[f\right] + \frac{c_1 K}{B} r^* + \frac{x(11(b-a) + c_2 BK)}{n}.$$

We shall use $r^*(\mathcal{F})$ to denote the fixed point of a $\psi$ that satisfies (41).

The following result, which is quoted from Bartlett et al. [2005], ensures that with a high probability a ball w.r.t. a true norm falls within a ball w.r.t. an empirical norm with a slightly larger radius whenever the Rademacher complexity is small enough. We use this result in Proposition 10 in Appendix B.3, where we use Dudley's entropy integral to upper bound the Rademacher complexity by an integral of the metric entropy. As we shall see in that proposition, we want to convert a set that is defined w.r.t. the true norm to a set that is defined w.r.t. the empirical norm.

**Proposition 6** (Corollary 2.2 of Bartlett et al. [2005]). *Let $\mathcal{F}$ be a class of $b$-bounded functions. For every $x > 0$ and $r$ that satisfies*

$$r \geq 10 b \mathbb{E}\left[R_n\left\{\, f \,:\, f \in \mathcal{F}, \mathbb{E}\left[f^2\right] \leq r \,\right\}\right] + \frac{11 b^2 x}{n},$$

*we then have*

$$\left\{\, f \in \mathcal{F} \,:\, \mathbb{E}\left[f^2\right] \leq r \,\right\} \subset \left\{\, f \in \mathcal{F} \,:\, \mathbb{E}_n\left[f^2\right] \leq 2r \,\right\},$$

*with probability at least $1 - e^{-x}$.*

# B Auxiliary Results

This appendix provides some auxiliary technical results that are required to prove Theorem 1.

## B.1 Bernstein-Type Condition

The following proposition allows us to verify the variance condition of Theorem 5 for the function space $\mathcal{G}$ (17) defined in the proof of Theorem 1. Recall that $L(\mathcal{P}) = \|(\mathcal{P}_z - \mathcal{P}_z^*)V\|_{2,\nu}^2$.

**Proposition 7.** *Let $\tilde{\mathcal{P}} \leftarrow \mathrm{argmin}_{\mathcal{P} \in \mathcal{M}} L(\mathcal{P})$. Assume that the function space $\mathcal{M}$ is convex, and compact w.r.t. $d_{\infty, TV}$. For any $\mathcal{P} \in \mathcal{M}$, we have*

$$\mathbb{E}\left[\left|(\mathcal{P}_Z - \tilde{\mathcal{P}}_Z)V\right|^2\right] \leq L(\mathcal{P}) - L(\tilde{\mathcal{P}}).$$

*Proof.* For a fixed bounded function $V$ and a transition kernel $\mathcal{P}$, define the function $h'_{\mathcal{P}}(z) = \int \mathcal{P}(\mathrm{d}y|z)V(y)$. Let $\mathcal{H}' = \{\, z \mapsto h'_{\mathcal{P}}(z) \,:\, \mathcal{P} \in \mathcal{M} \,\}$. We first show that when $\mathcal{M}$ is convex and compact, the space $\mathcal{H}'$ is convex and closed.

To see the convexity of $\mathcal{H}'$, pick $h'_1, h'_2 \in \mathcal{H}'$, with their corresponding $\mathcal{P}_1, \mathcal{P}_2 \in \mathcal{M}$. For any $0 \leq \lambda \leq 1$,

$$\lambda h'_1 + (1-\lambda)h'_2 = \int \left(\lambda \mathcal{P}_1(\mathrm{d}y|z) + (1-\lambda)\mathcal{P}_2(\mathrm{d}y|z)\right)V(y) = \int \mathcal{P}_\lambda(\mathrm{d}y|z)V(y),$$

with $\mathcal{P}_\lambda = \lambda \mathcal{P}_1 + (1-\lambda)\mathcal{P}_2$. As $\mathcal{M}$ is convex, $\mathcal{P}_\lambda \in \mathcal{M}$, and as a result $h'_{\mathcal{P}_\lambda} \in \mathcal{H}'$. So $\mathcal{H}'$ is convex.

We now show that if $\mathcal{M}$ is a compact set, $\mathcal{H}'$ is a closed set. To see this, equip $\mathcal{H}'$ with the supremum norm metric $d_\infty$, i.e., for $h'_1, h'_2$, we denote $d_\infty(h'_1, h'_2) = \|h'_1 - h'_2\|_\infty$. We use mixed TV-supremum metric $d_{\mathrm{TV},\infty}(\mathcal{P}_1, \mathcal{P}_2)$, defined in Appendix A.1, for $\mathcal{M}$.

The function $h'_{\mathcal{P}}$ is continuous from the metric space $(\mathcal{M}, d_\infty)$ to the metric space $(\mathcal{H}, d_{\mathrm{TV},\infty})$. We show this by verifying its Lipschitzness:

$$d_\infty\left(h'_{\mathcal{P}_1}, h'_{\mathcal{P}_2}\right) = \sup_z \left| \int \left(\mathcal{P}_1(\mathrm{d}y|z) - \mathcal{P}_2(\mathrm{d}y|z)\right) V(y) \right| \leq \|V\|_\infty \, d_{\mathrm{TV},\infty}(\mathcal{P}_1, \mathcal{P}_2).$$

As $\mathcal{H}'$ is the mapping of a compact $\mathcal{M}$ by the continuous function $h'$, $\mathcal{H}'$ is also compact.[7] Since compact sets are closed, we have shown that $\mathcal{H}'$ is compact too.

After establishing these facts about $\mathcal{H}$, we return to the loss function $L(\mathcal{P})$ and its minimizer $\tilde{\mathcal{P}}$. With the notation $h'_{\mathcal{P}} = \mathcal{P}V$, we have $L(\mathcal{P}) = \|(\mathcal{P}_z - \mathcal{P}_z^*)V\|_{2,\nu}^2 = \|h'_{\mathcal{P}} - h'_{\mathcal{P}^*}\|_{2,\nu}^2$. The function $h'_{\tilde{\mathcal{P}}} = \tilde{\mathcal{P}}V$ is the $L_2(\nu)$-projection of $\mathcal{P}^*V$ onto $\mathcal{H}'$. By the variational characterization of projection onto a convex and closed set, we have that for any $\mathcal{P}V = h'_{\mathcal{P}} \in \mathcal{H}'$,

$$\mathbb{E}\left[\left\langle \mathcal{P}_Z^*V - \tilde{\mathcal{P}}_Z V \,,\, \mathcal{P}_Z V - \tilde{\mathcal{P}}_Z V \right\rangle\right] \leq 0.$$

By adding and subtracting $\mathcal{P}_Z^*V$ and re-arranging, we get that

$$\mathbb{E}\left[\left\langle \mathcal{P}_Z^*V - \tilde{\mathcal{P}}_Z V \,,\, \mathcal{P}_Z V - \mathcal{P}_Z^*V + \mathcal{P}_Z^*V - \tilde{\mathcal{P}}_Z V \right\rangle\right] \leq 0$$

$$\Leftrightarrow \mathbb{E}\left[\left\langle \mathcal{P}_Z^*V - \tilde{\mathcal{P}}_Z V \,,\, \mathcal{P}_Z V - \mathcal{P}_Z^*V \right\rangle\right] \leq -\mathbb{E}\left[\left|(\mathcal{P}_Z^* - \tilde{\mathcal{P}}_Z)V\right|^2\right].$$

We now expand $\mathbb{E}\left[\left|(\mathcal{P}_Z - \tilde{\mathcal{P}}_Z)V\right|^2\right]$ and use the obtained inequality for the inner product term to get

$$\mathbb{E}\left[\left|(\mathcal{P}_Z - \tilde{\mathcal{P}}_Z)V\right|^2\right] = \mathbb{E}\left[\left|(\mathcal{P}_Z - \mathcal{P}_Z^*)V\right|^2\right] + \mathbb{E}\left[\left|(\mathcal{P}_Z^* - \tilde{\mathcal{P}}_Z)V\right|^2\right] +$$

$$2\mathbb{E}\left[\left\langle (\mathcal{P}_Z - \mathcal{P}_Z^*)V \,,\, (\mathcal{P}_Z^* - \tilde{\mathcal{P}}_Z)V \right\rangle\right]$$

$$\leq \mathbb{E}\left[\left|(\mathcal{P}_Z - \mathcal{P}_Z^*)V\right|^2\right] - \mathbb{E}\left[\left|(\mathcal{P}_Z^* - \tilde{\mathcal{P}}_Z)V\right|^2\right] = L(\mathcal{P}) - L(\tilde{\mathcal{P}}),$$

which is the desired result. $\qquad \square$

## B.2 Modulus of Continuity of Weighted Sum

Let $\mathcal{G}$ be a class of functions on $\mathcal{Z}$ and $z_{1:n} = z_1, \ldots, z_n$ be a sequence of points in $\mathcal{Z}$. Recall that $P_n$ is the empirical measure, i.e., $P_n = P_{z_{1:n}} = \frac{1}{n}\sum_{i=1}^n \delta_{z_i}$, and we define the empirical $L_2(P_n)$-norm of $g$ as $\|g\|_n^2 = \|g\|_{P_n}^2 = \frac{1}{n}\sum_{i=1}^n |g(z_i)|^2$.

Consider i.i.d. random variables $W_1, \ldots, W_n$. We denote the weighted sum of $g$ and the random vector $W$ (or their inner product) as follows:

$$\langle g \,,\, W_i \rangle_n \triangleq \frac{1}{n}\sum_{i=1}^n g(z_i)W_i. \tag{42}$$

We are interested in the behaviour of $\langle g \,,\, W_i \rangle_n$ as a function of the empirical norm $\|g\|_{P_n}$. To prove such a result, we use an intermediate result on the supremum of the weighted sums. The following lemma is Theorem 19.1 of Györfi et al. [2002], which we quote here for the ease of reference. This result, in a more general form, appears as Lemma 3.2 of van de Geer [2000].

**Lemma 8.** *Let $L > 0$ and $W_1, \ldots, W_n$ be independent random variables with expectation zero and values in $[-L, L]$. Let $z_1, \ldots, z_n \in \mathbb{R}^d$, let $R > 0$, and let $\mathcal{G}$ be a class of functions $g : \mathbb{R}^d \to \mathbb{R}$ with the property*

$$\frac{1}{n}\sum_{i=1}^n |g(z_i)|^2 \leq R^2,$$

*for all $g \in \mathcal{G}$. Then the metric entropy condition*

$$\sqrt{n}\varepsilon \geq 48\sqrt{2}L \int_{\frac{\varepsilon}{8L}}^{\frac{R}{2}} \sqrt{\log \mathcal{N}(u, \mathcal{G}, L_2(P_{z_{1:n}}))} \mathrm{d}u \tag{43}$$

*and*

$$\sqrt{n}\varepsilon \geq 36RL \tag{44}$$

*imply*

$$\mathbb{P}\left\{ \sup_{g \in \mathcal{G}} |\langle g, W_i \rangle| > \varepsilon \right\} \leq 5 \exp\left( -\frac{n\varepsilon^2}{2304L^2R^2} \right).$$

We are now ready to state and prove the main result of this section. We use the *peeling device* in the proof (cf. Section 5.3 of van de Geer 2000).

**Lemma 9.** *Consider a function space $\mathcal{G}$, points $z_{1:n} \subset \mathcal{Z}$, and random variables $W_i$. Assume that the random variables $W_i$ are i.i.d., zero mean, and bounded by $L$, i.e., $\mathbb{E}[W_i] = 0$ and $|W_i| \leq L$ for all $i = 1, \dots, n$. Suppose that the function space $\mathcal{G}$ satisfies the following metric entropy condition*

$$\log \mathcal{N}(u, \mathcal{G}, L_2(P_{z_{1:n}})) \leq c \left( \frac{B}{u} \right)^{2\alpha}$$

*for some constants $c, B > 0$ and $0 < \alpha < 1$. For any $\delta > 0$, with probability at least $1 - \delta$, we have*

$$\sup_{g \in \mathcal{G}} \frac{|\langle g, W_i \rangle_n|}{\|g\|_{P_n}^{1-\alpha}} \leq c_1(\alpha, L, B) \sqrt{\frac{\ln(1/\delta)}{n}},$$

*for some $c_1(\alpha, L, B) = c_2(\alpha)[L^{1+\alpha} + B^\alpha]$ and a finite function $c_2(\alpha) > 0$.*

*Proof.* We use the peeling device to upper bound the modulus of continuity. We write

$$\delta = \mathbb{P}\left\{ \sup_{g \in \mathcal{G}} \frac{|\langle g, W_i \rangle_n|}{\|g\|_{P_n}^{1-\alpha}} \geq t \right\}$$

$$\leq \sum_{s \geq 0} \mathbb{P}\left\{ \sup_{g \in \mathcal{G}} \frac{|\langle g, W_i \rangle_n|}{\|g\|_{P_n}^{1-\alpha}} \geq t, 2^{-(s+1)}L\,\mathbb{I}\{s \neq 0\} < \|g\|_{P_n} \leq 2^{-s}L \right\}$$

$$\leq \sum_{s \geq 0} \mathbb{P}\left\{ \sup_{g \in \mathcal{G}} |\langle g, W_i \rangle_n| \geq \left( 2^{-(s+1)}L \right)^{1-\alpha} t, \|g\|_{P_n} \leq 2^{-s}L \right\}.$$

We apply Lemma 8 to each term in the RHS summation. Upon satisfaction of the conditions of the lemma, which we shall verify, with the choice of $(2^{-(s+1)}L)^{1-\alpha}t$ as $\varepsilon$ in the lemma, setting $R = 2^{-s}L$, and keeping $L$ the same, we get

$$\delta \leq \sum_{s \geq 0} 5 \exp\left( -\frac{n[2^{-(s+1)}L]^{2(1-\alpha)}t^2}{2304L^2(2^{-s}L)^2} \right)$$

$$= 5 \sum_{s \geq 0} \exp\left( -\frac{nt^2 2^{2\alpha} 2^{2\alpha s}}{4 \times 2304L^{2(1+\alpha)}} \right)$$

$$\leq 5c_1(\alpha) \sum_{s \geq 1} \exp\left( -\frac{nt^2 2^{2\alpha} s}{4 \times 2304L^{2(1+\alpha)}} \right)$$

$$= \frac{5c_1(\alpha) \exp\left( -\frac{nt^2 2^{2\alpha}}{4 \times 2304L^{2(1+\alpha)}} \right)}{1 - \exp\left( -\frac{nt^2 2^{2\alpha}}{4 \times 2304L^{2(1+\alpha)}} \right)}$$

$$\leq 10c_1(\alpha) \exp\left( -\frac{nt^2 2^{2\alpha}}{4 \times 2304L^{2(1+\alpha)}} \right). \tag{45}$$

The multiplier $c_1(\alpha)$ compensates for all those terms when $2^{2\alpha s}$ is not larger than $s$, which is required for the substitution in the second inequality, and can conservatively be selected as $c_1(\alpha) = \frac{1}{(2\alpha \ln(2))^2}$. The last inequality is satisfied under the condition that

$$t \geq \frac{\sqrt{\ln 2} \times 2^5 \times 3 \times L^{1+\alpha}}{2^\alpha \sqrt{n}}. \tag{46}$$

We now verify conditions (43) and (44) of Lemma 8 for each $s \geq 1$.

With our choice of $R$, $\varepsilon$, and $L$, the metric entropy condition (43) is satisfied if

$$\sqrt{n} \left(2^{-(s+1)} L\right)^{1-\alpha} t \geq 48\sqrt{2} L \int_0^{2^{-(s+1)L}} \sqrt{\log \mathcal{N}(u, \mathcal{G}, L_2(P_{z_{1:n}})} du.$$

Under the assumed entropy condition, the integral on the RHS is upper bounded $\frac{\sqrt{c} B^\alpha u^{1-\alpha}}{1-\alpha}$ with $u = 2^{-(s+1)} L$, so it is sufficient to verify that

$$\sqrt{n} \left(2^{-(s+1)} L\right)^{1-\alpha} t \geq 48\sqrt{2} L \frac{\sqrt{c} B^\alpha \left(2^{-(s+1)} L\right)^{1-\alpha}}{1-\alpha},$$

which can be simplified as

$$t \geq \frac{c_2(\alpha) B^\alpha}{\sqrt{n}} \tag{47}$$

for some function $c_2(\alpha)$, which is bounded away from zero as long as $\alpha > 0$.

Likewise for the condition (44) we have

$$\sqrt{n} \left(2^{-(s+1)} L\right)^{1-\alpha} t \geq 36(2^{-s} L) L,$$

which is satisfied if

$$t \geq \frac{72 L^{1+\alpha}}{\sqrt{n}}. \tag{48}$$

Solving (45) for $\delta$ and considering the conditions (46), (47), and (48), we get that

$$\sup_{g \in \mathcal{G}} \frac{|\langle g, W_i \rangle_n|}{\|g\|_{P_n}^{1-\alpha}} \leq \frac{c_2(\alpha) B^\alpha}{\sqrt{n}} + \frac{2^5 L^{1+\alpha} \left[3\sqrt{\ln\left(\frac{10 c_1(\alpha)}{\delta}\right)} + 2\right]}{\sqrt{n}},$$

with probability at least $1 - \delta$. $\qquad\square$

## B.3 Local Rademacher Complexity of $\mathcal{F}$ and $\mathcal{G}$

We provide upper bounds for the local Rademacher complexity of function spaces $\mathcal{F}$ (25) and $\mathcal{G}$ (17), defined in the proof of Theorem 1. We follow the same notations as in that proof.

**Proposition 10.** *Under the condition that* $\log \mathcal{N}(u, \mathcal{M}, L_2(P_n)) \leq c \left(\frac{R}{u}\right)^{2\alpha}$ *with* $0 < \alpha < 1$, *we have*

$$r^*(\mathcal{F}), r^*(\mathcal{G}) \leq \frac{c_1(\alpha) V_{max}^4 R^{\frac{2\alpha}{1+\alpha}}}{n^{\frac{1}{1+\alpha}}} + \frac{336 V_{max}^4 \log n}{n},$$

*where* $c_1(\alpha) = \frac{c'}{(1-\alpha)^{\frac{2}{1+\alpha}}}$, *with possibly different finite constant* $c' > 0$ *for each case.*

*Proof.* We find sub-root functions $\psi(r)$ with the fixed points $r^*(\mathcal{G})$ and $r^*(\mathcal{F})$ such that they satisfy the conditions of Theorem 5.

**Fixed Point $r^*(\mathcal{G})$.** Recall that

$$\mathcal{G} = \{\, z \mapsto l_\mathcal{P}(z) - l_{\tilde{\mathcal{P}}}(z) \,:\, \mathcal{P} \in \mathcal{M} \,\}.$$

We want to find a sub-root $\psi(r)$ with the fixed point $r^*(\mathcal{G})$ such that

$$\psi(r) \geq B\mathbb{E}\left[R_n\{g \in \mathcal{G} : T(g) \leq r\}\right],$$

for any $r \geq r^*(\mathcal{G})$, and for $T(g)$ and $B$ that satisfy the conditions specified in the theorem. According to (21), we can choose $T(g) = 16V_{\max}^2\mathbb{E}\left[|(\mathcal{P}_z - \tilde{\mathcal{P}}_z)V|^2\right]$ and $B = 16V_{\max}^2$. So we have

$$\psi(r) \geq 16V_{\max}^2\mathbb{E}\left[R_n\left\{g \in \mathcal{G} : 16V_{\max}^2\mathbb{E}\left[|(\mathcal{P}_z - \tilde{\mathcal{P}}_z)V|^2\right] \leq r\right\}\right].$$

As $g^2(z) \leq 16V_{\max}^2|(\mathcal{P}_z - \tilde{\mathcal{P}}_z)V|^2$, the set $\left\{\, g \,:\, \mathbb{E}\left[g^2\right] \leq r \,\right\}$ is a superset of $\{\, g \,:\, T(g) \leq r \,\}$, so we verify the following relaxed condition instead:

$$\psi(r) \geq 16V_{\max}^2\mathbb{E}\left[R_n\left\{g \in \mathcal{G} : \mathbb{E}\left[g^2(Z)\right] \leq r\right\}\right]. \tag{49}$$

We may use Dudley's entropy integral to relate the covering number and Rademacher complexity, see e.g., Theorem 3.5.1 and Theorem 2.3.7 of Giné and Nickl [2015], or Theorem A.7 of Bartlett et al. [2005] (originally from Dudley).[8] But the covering number appearing in the Dudley's integral is w.r.t. the empirical norm, as opposed to the true norm we have here. We use Proposition 6, which is originally Corollary 2.2 of Bartlett et al. [2005], to relate these upper bounds. As $g$ is $4V_{\max}^2$-bounded, the condition of the proposition would be satisfied if

$$r \geq 10 \times 4V_{\max}^2\mathbb{E}\left[R_n\left\{\, g \,:\, g \in \mathcal{G}, \mathbb{E}\left[g^2\right] \leq r \,\right\}\right] + \frac{11(4V_{\max})^2\log n}{n}, \tag{50}$$

in which case we have

$$\left\{\, g \in \mathcal{G} \,:\, \mathbb{E}\left[g^2\right] \leq r \,\right\} \subset \left\{\, g \in \mathcal{G} \,:\, \mathbb{E}_n\left[g^2\right] \leq 2r \,\right\},$$

with probability at least $1 - \frac{1}{n}$. By the definition of the Rademacher complexity, if $F_1 \subset F_2$, it holds that $R_n(\mathcal{F}_1) \leq R_n(\mathcal{F}_2)$, see e.g., Theorem 12 of Bartlett and Mendelson [2002]. So whenever this subset relation is true, which holds with probability at least $1 - \frac{1}{n}$, we can compare the Rademacher complexities of both sides. The failure event of the subset relation has probability at most $1/n$, so its effect on the Rademacher complexity is at most $4V_{\max}^2/n$. Thus, we can write

$$\mathbb{E}\left[R_n\left\{\, g \in \mathcal{G} \,:\, \mathbb{E}\left[g^2\right] \leq r \,\right\}\right] \leq \mathbb{E}\left[R_n\left\{\, g \in \mathcal{G} \,:\, \mathbb{E}_n\left[g^2\right] \leq 2r \,\right\}\right] + \frac{4V_{\max}^2}{n}. \tag{51}$$

We choose the following $\psi$, which satisfies both conditions (49) and (50):

$$\psi(r) = \left(16V_{\max}^2 \vee 10 \times 4V_{\max}^2\right)\mathbb{E}\left[R_n\left\{\, g \in \mathcal{G} \,:\, \mathbb{E}\left[g^2\right] \leq r \,\right\}\right] + \frac{11(4V_{\max}^2)^2\log n}{n}. \tag{52}$$

Consider the fixed point $r^* = r^*(\mathcal{G}) = \psi(r^*(\mathcal{G}))$. By evoking (51), it can be seen that it satisfies

$$r^* \leq \psi(r^*) = 40V_{\max}^2\mathbb{E}\left[R_n\left\{\, g \in \mathcal{G} \,:\, \mathbb{E}_n\left[g^2\right] \leq 2r^* \,\right\}\right] + \frac{176V_{\max}^4\log n + 40V_{\max}^2 \times 4V_{\max}^2}{n}. \tag{53}$$

We can now use Dudley's entropy integral to upper bound the Rademacher complexity with the integral of covering numbers at different scales. For any $r > 0$, we have

$$\mathbb{E}\left[R_n\left\{\, g \in \mathcal{G} \,:\, \mathbb{E}_n\left[g^2\right] \leq r \,\right\}\right] \leq \frac{c}{\sqrt{n}}\int_0^{\sqrt{r}}\sqrt{\log\mathcal{N}(u, \{\, g \in \mathcal{G} \,:\, \mathbb{E}_n\left[g^2\right] \leq r \,\}, L_2(P_{Z_{1:n}}))}\,\mathrm{d}u$$

$$\leq \frac{c}{\sqrt{n}}\int_0^{\sqrt{r}}\sqrt{\log\mathcal{N}(u, \mathcal{G}, L_2(P_{Z_{1:n}}))}\,\mathrm{d}u.$$

The covering number assumption $\log \mathcal{N}(u, \mathcal{M}, L_2(P_n)) \leq c(\frac{R}{u})^{2\alpha}$ along Proposition 11 show that

$$\log \mathcal{N}(u, \mathcal{G}, L_2(P_n)) \leq c\left(\frac{2V_{\max}^2 R}{u}\right)^{2\alpha}.$$

With this choice of the covering number, the entropy integral is

$$\frac{c'(V_{\max}^2 R)^\alpha \, r^{\frac{1-\alpha}{2}}}{(1-\alpha)\sqrt{n}}.$$

We plug this upper bound of the Rademacher complexity into (53) to obtain the implicit inequality

$$r^* \leq \frac{c_1 V_{\max}^2 (V_{\max}^2 R)^\alpha}{(1-\alpha)\sqrt{n}} r^{* \frac{1-\alpha}{2}} + \frac{176 V_{\max}^4 \log n + 160 V_{\max}^4}{n}. \tag{54}$$

We solve this inequality to obtain that

$$r^*(\mathcal{G}) \leq \frac{c_1 V_{\max}^4 R^{\frac{2\alpha}{1+\alpha}}}{(1-\alpha)^{\frac{2}{1+\alpha}} n^{\frac{1}{1+\alpha}}} + \frac{336 V_{\max}^4 \log n}{n}.$$

**Fixed Point** $r^*(\mathcal{F})$**.** The proof is very similar to the proof for the previous part with the main difference being in the selection of $B$ and $T(f)$, which happens to be inconsequential. Recall that

$$\mathcal{F} = \left\{ z \mapsto \left|(\mathcal{P}_z - \tilde{\mathcal{P}})V\right|^2 : \mathcal{P} \in \mathcal{M} \right\}.$$

For this function space, according to (26), we can choose $B = 4V_{\max}^2$ and $T(f) = 4V_{\max}^2 \mathbb{E}[f]$. With a similar argument leading to (49), we need to verify

$$\psi(r) \geq 4V_{\max}^2 \mathbb{E}\left[R_n\left\{f \in \mathcal{F} : \mathbb{E}\left[f^2(Z)\right] \leq r\right\}\right].$$

As $f$ is $4V_{\max}^2$-bounded, which is the same as $g$, the constants in condition (50) and upper bound (51) remains unchanged, so we can choose

$$\psi(r) = \left(4V_{\max}^2 \vee 10 \times 4V_{\max}^2\right) \mathbb{E}\left[R_n\left\{g \in \mathcal{F} : \mathbb{E}\left[f^2\right] \leq r\right\}\right] + \frac{11(4V_{\max}^2)^2 \log n}{n}.$$

As the first term is $40V_{\max}^2$, this choice of $\psi$ becomes the same as (52), with the difference only in having the Rademacher complexity of subsets of $\mathcal{G}$ instead of $\mathcal{F}$. We provide an upper bound for $\mathbb{E}\left[R_n\left\{g \in \mathcal{F} : \mathbb{E}\left[f^2\right] \leq r\right\}\right]$ using Dudley's integral with the covering number of $\mathcal{F}$ instead instead of $\mathcal{G}$. Proposition 11 provides the same upper for the covering number of $\mathcal{F}$ and $\mathcal{G}$ as a function of the upper bound of $\mathcal{G}$. Therefore, the Dudley's integral would lead to the same implicit inequality as (54), and we get the same upper bound for $r^*(\mathcal{F})$ as that of $r^*(\mathcal{G})$. □

## B.4   Covering Number of $\mathcal{F}$, $\mathcal{G}$, and $\mathcal{H}$

We provide covering number results for slight generalization of function spaces $\mathcal{H}$ (23), $\mathcal{G}$ (17), and $\mathcal{F}$ (25) defined in the proof of Theorem 1 as a function of the covering number of $\mathcal{M}$. The generalization is that this result is valid for any $\mathcal{P}'$ instead of a specific $\tilde{\mathcal{P}}$.

**Proposition 11.** *For a fixed probability transition kernel $\mathcal{P}'$ and a $V_{max}$-bounded function $V$, define*

$$\mathcal{H} = \left\{ z \mapsto (\mathcal{P}_z - \mathcal{P}'_z)V : \mathcal{P} \in \mathcal{M} \right\},$$
$$\mathcal{G} = \left\{ z \mapsto l_{\mathcal{P}}(z) - l_{\mathcal{P}'}(z) : \mathcal{P} \in \mathcal{M} \right\},$$
$$\mathcal{F} = \left\{ z \mapsto |(\mathcal{P}_z - \mathcal{P}'_z)V|^2 : \mathcal{P} \in \mathcal{M} \right\}.$$

*For any sequence $z_{1:n} \subset \mathcal{Z}$, we have*

$$\mathcal{N}\left(\varepsilon, \mathcal{H}, L_2(z_{1:n})\right) \leq \mathcal{N}\left(\frac{\varepsilon}{V_{max}}, \mathcal{M}, L_2(z_{1:n})\right),$$

$$\mathcal{N}\left(\varepsilon, \mathcal{G}, L_2(z_{1:n})\right) \leq \mathcal{N}\left(\frac{\varepsilon}{2V_{max}^2}, \mathcal{M}, L_2(z_{1:n})\right),$$

$$\mathcal{N}\left(\varepsilon, \mathcal{F}, L_2(z_{1:n})\right) \leq \mathcal{N}\left(\frac{\varepsilon}{2V_{max}^2}, \mathcal{M}, L_2(z_{1:n})\right).$$

*Proof.* For any $h_1, h_2 \in \mathcal{H}$, and for any sequence $z_{1:n} \subset \mathcal{Z}$, we have

$$\frac{1}{n}\sum_{i=1}^{n}|h_1(z_i) - h_2(z_i)|^2 = \frac{1}{n}\sum_{i=1}^{n}|(\mathcal{P}_1(\cdot|z_i) - \mathcal{P}_2(\cdot|z_i))\,V|^2$$

$$= \frac{1}{n}\sum_{i=1}^{n}\left|\int(\mathcal{P}_1(\mathrm{d}y|z_i) - \mathcal{P}_2(\mathrm{d}y|z_i))\,V(y)\right|^2$$

$$\leq \frac{1}{n}\sum_{i=1}^{n}\left[V_{\max}\int|\mathcal{P}_1(\mathrm{d}y|z_i) - \mathcal{P}_2(\mathrm{d}y|z_i)|\right]^2$$

$$= \frac{V_{\max}^2}{n}\sum_{i=1}^{n}\|\mathcal{P}_1(\cdot|z_i) - \mathcal{P}_2(\cdot|z_i)\|_1^2$$

$$\leq \frac{V_{\max}^2}{n}\sum_{i=1}^{n}\|\mathcal{P}_1(\cdot|z_i) - \mathcal{P}_2(\cdot|z_i)\|_2^2.$$

So an $\varepsilon$-cover on $\{z \mapsto \mathcal{P}(\cdot|z) : \mathcal{P} \in \mathcal{M}\}$ w.r.t. $L_2(z_{1:n})$ induces an $V_{\max}\varepsilon$-cover on $\mathcal{H}$ w.r.t. the same measure.

The proof of the covering number upper bound for $\mathcal{G}$ is similar. For $g_1, g_2 \in \mathcal{G}$, we have

$$|g_1(z) - g_2(z)|^2 = |l_{\mathcal{P}_1}(z) - l_{\mathcal{P}_2}(z)|^2$$

$$\leq |(\mathcal{P}_1(\cdot|z) - \mathcal{P}_2(\cdot|z))\,V|^2 \times |(\mathcal{P}_1(\cdot|z) + \mathcal{P}_1(\cdot|z) - 2\mathcal{P}^*(\cdot|z))\,V|^2$$

$$\leq (2V_{\max})^2\,|(\mathcal{P}_1(\cdot|z) - \mathcal{P}_2(\cdot|z))\,V|^2.$$

We take summation over $z_{1:n}$ on both sides. Notice that $|(\mathcal{P}_1(\cdot|z_i) - \mathcal{P}_2(\cdot|z_i))V|^2$ terms are the same as those appearing in the previous case, so by continuing the same calculations, we can provide an upper bound that has the multiplicative constant of $4V_{\max}^4$ instead of $V_{\max}^2$ of the previous case. So an $\varepsilon$-cover on $\mathcal{M}$ induces a $2V_{\max}^2\varepsilon$-cover on $\mathcal{G}$ w.r.t. $L_2(z_{1:n})$.

Finally, for the case of $\mathcal{F}$, we see that for $f_1, f_2 \in \mathcal{F}$, we have

$$|f_1(z) - f_2(z)|^2 = |[(\mathcal{P}_1(\cdot|z) - \mathcal{P}_2(\cdot|z))\,V]\,[(\mathcal{P}_1(\cdot|z) + \mathcal{P}_2(\cdot|z) - 2\mathcal{P}'(\cdot|z))\,V]|^2$$

$$\leq (2V_{\max})^2\,|(\mathcal{P}_1(\cdot|z) - \mathcal{P}_2(\cdot|z))\,V|^2.$$

This is the same as the case of $\mathcal{G}$, so we obtain the same upper bound for the covering number of $\mathcal{F}$. $\qquad\square$

## B.5 Simplification of the Excess Error Inequality (27)

The following proposition simplifies (27). We use the same notations and definitions as in the proof of Theorem 1.

**Proposition 12.** *Consider the inequality* (27), *which is implicit in* $E \triangleq L(\hat{\mathcal{P}}) - L(\tilde{\mathcal{P}})$. *Assume that* $0 < \alpha < 1$, *and* $V_{max}, R \geq 1$. *Fix* $\delta > 0$ *and let* $\delta_1 = \delta_2 = \delta_3 = \delta/3$. *The following explicit upper bound for* $E$ *hold:*

$$E \leq c(\alpha)V_{max}^2 R^{\frac{2\alpha}{1+\alpha}}\left[\frac{1}{n^{\frac{1}{1+\alpha}}} + \frac{\log n}{n^{1-\frac{\alpha}{2}}}\right]\sqrt{\ln(1/\delta)},$$

*for a bounded function* $c(\alpha) > 0$, *which is independent of* $n$, $V_{max}$, $R$, *and* $\delta$.

*Proof.* We re-write inequality (27)

$$E \leq 4c(\alpha, V_{\max}, R)t_n\left[\underbrace{2E}_{\triangleq(\text{a1})} + \underbrace{\frac{c_1}{4V_{\max}^2}r^*(\mathcal{F}) + \frac{(44V_{\max}^2 + c_2 \times 4V_{\max}^2)\ln(1/\delta_3)}{n}}_{\triangleq(\text{a2})}\right]^{\frac{1-\alpha}{2}} +$$

$$\frac{2c_1}{16V_{\max}^2}r^*(\mathcal{G}) + \frac{(88V_{\max}^2 + 32c_2 V_{\max}^2)\ln(\frac{1}{\delta_1})}{n} \triangleq (\text{A}) + (\text{B}).$$

Here (A) refers to the first term in the RHS and (B) refers to the second and third terms; (a1) and (a2) refer to terms within the bracket.

The argument to solve the implicit inequality is based on considering several cases in which each term dominates the others, and solving the inequality for that case. The overall upper bound for $E$ is obtained by adding the upper bounds obtained in each case.

We either have (A) $\geq$ (B) or vice versa. In the former case, either (a1) $\geq$ (a2) (which we call Case (i)) or (a2) $>$ (a1) (Case (ii)). Case (iii) is when (B) $>$ (A). We analyze these cases. In the rest, we let $\delta > 0$ and set $\delta_1 = \delta_2 = \delta_3 = \delta/3$.

**Case (i):** (A) $\geq$ (B) and (a1) $\geq$ (a2). In this case, we have

$$E \leq 2 \times 4c(\alpha, V_{\max}, R)(4E)^{\frac{1-\alpha}{2}} t_0.$$

We solve for $E$, expand $c(\alpha, V_{\max}, R)$, and use the assumption that $V_{\max} \geq 1$ and $\frac{2\alpha}{1+\alpha} < 1$ for the range of $0 < \alpha < 1$ to get that

$$E \leq 2^{\frac{2(4-\alpha)}{1+\alpha}} c(\alpha, V_{\max}, R)^{\frac{2}{1+\alpha}} \left( \frac{\ln(\frac{1}{\delta_2})}{n} \right)^{\frac{1}{1+\alpha}}$$

$$\leq c(\alpha) \left[ V_{\max}^2 + V_{\max}^{\frac{2\alpha}{1+\alpha}} R^{\frac{2\alpha}{1+\alpha}} \right] \left( \frac{\ln(\frac{1}{\delta_2})}{n} \right)^{\frac{1}{1+\alpha}}$$

$$\leq c_1(\alpha) \left[ V_{\max}^2 + V_{\max} R^{\frac{2\alpha}{1+\alpha}} \right] \left( \frac{\ln(\frac{1}{\delta})}{n} \right)^{\frac{1}{1+\alpha}}. \tag{55}$$

**Case (ii):** (A) $\geq$ (B) and (a1) $<$ (a2). We have

$$E \leq 2 \times 4c(\alpha, V_{\max}, R)t_n \left[ 2 \left( \frac{c_1}{4V_{\max}^2} r^*(\mathcal{F}) + \frac{(44V_{\max}^2 + c_2 \times 4V_{\max}^2)\ln(1/\delta_3)}{n} \right) \right]^{\frac{1-\alpha}{2}}.$$

Proposition 10 shows that for our choice of function space, the local Rademacher complexity satisfies

$$r^*(\mathcal{F}) \leq \frac{c_2(\alpha)V_{\max}^4 R^{\frac{2\alpha}{1+\alpha}}}{n^{\frac{1}{1+\alpha}}} + \frac{336V_{\max}^4 \log n}{n}.$$

Therefore,

$$E \leq 8 \times 2^{\frac{1-\alpha}{2}} c(\alpha, V_{\max}, R)t_n \left[ \frac{c_3(\alpha)V_{\max}^2 R^{\frac{2\alpha}{1+\alpha}}}{n^{\frac{1}{1+\alpha}}} + \frac{c_2' V_{\max}^2 \log(n/\delta_3)}{n} \right]^{\frac{1-\alpha}{2}}$$

$$\leq c_4(\alpha) \left[ V_{\max}^{1+\alpha} + (V_{\max}R)^\alpha \right] \left[ \frac{V_{\max}^{1-\alpha} R^{\frac{\alpha(1-\alpha)}{1+\alpha}}}{n^{\frac{1}{1+\alpha}}} \sqrt{\ln(1/\delta_3)} + \frac{V_{\max}^{1-\alpha} (\log(n/\delta_3))^{\frac{1-\alpha}{2}} \sqrt{\ln(1/\delta_2)}}{n^{1-\frac{\alpha}{2}}} \right]$$

$$\leq c_5(\alpha) \left( V_{\max}^2 \vee V_{\max} \right) R^{\frac{2\alpha}{1+\alpha}} \frac{\sqrt{\ln(1/\delta)}}{n^{\frac{1}{1+\alpha}}} + c_6(\alpha) \left( V_{\max}^2 + V_{\max}R^\alpha \right) (\log n)^{\frac{1-\alpha}{2}} \left( \frac{\ln(1/\delta)}{n} \right)^{1-\frac{\alpha}{2}},$$

where we expanded $c(\alpha, V_{\max}, R)$ by its definition as $c(\alpha) \left[ V_{\max}^{1+\alpha} + (V_{\max}R)^\alpha \right]$ and used the fact that $0 < \alpha < 1$ to simplify some of the terms. After some manipulations, and using the simplifying assumptions that $V_{\max}, R \geq 1$, we obtain

$$E \leq c_7(\alpha)V_{\max}^2 R^{\frac{2\alpha}{1+\alpha}} \left[ \frac{1}{n^{\frac{1}{1+\alpha}}} + \frac{(\log n)^{\frac{1-\alpha}{2}}}{n^{1-\frac{\alpha}{2}}} \right] \sqrt{\ln(1/\delta)}. \tag{56}$$

**Case (iii):** (A) $<$ (B). We have

$$E \leq \frac{2 \times 2c_1}{16V_{\max}^2} r^*(\mathcal{G}) + \frac{2(88V_{\max}^2 + 32c_2 V_{\max}^2)\ln(\frac{1}{\delta_1})}{n}.$$

We upper bound $r^*(\mathcal{G})$ by Proposition 10 to get

$$
\begin{aligned}
E &\leq \frac{c_1}{4V_{\max}^2}\left[\frac{c_8(\alpha)V_{\max}^4 R^{\frac{2\alpha}{1+\alpha}}}{n^{\frac{1}{1+\alpha}}} + \frac{336 V_{\max}^4 \log n}{n}\right] + \frac{2(88V_{\max}^2 + 32c_2 V_{\max}^2)\ln(\frac{1}{\delta_1})}{n} \\
&\leq c_9(\alpha)\left[\frac{V_{\max}^2 R^{\frac{2\alpha}{1+\alpha}}}{n^{\frac{1}{1+\alpha}}} + \frac{V_{\max}^2 \log(n/\delta)}{n}\right].
\end{aligned}
\tag{57}
$$

By adding the upper bounds of (55), (56), and (57) we obtain

$$
\begin{aligned}
E \leq &\, c_1(\alpha)\left[V_{\max}^2 + V_{\max}R^{\frac{2\alpha}{1+\alpha}}\right]\left(\frac{\ln(\frac{1}{\delta})}{n}\right)^{\frac{1}{1+\alpha}} + \\
&\, c_7(\alpha)V_{\max}^2 R^{\frac{2\alpha}{1+\alpha}}\left[\frac{1}{n^{\frac{1}{1+\alpha}}} + \frac{(\log n)^{\frac{1-\alpha}{2}}}{n^{1-\frac{\alpha}{2}}}\right]\sqrt{\ln(1/\delta)} + c_9(\alpha)\left[\frac{V_{\max}^2 R^{\frac{2\alpha}{1+\alpha}}}{n^{\frac{1}{1+\alpha}}} + \frac{V_{\max}^2 \log(n/\delta)}{n}\right] \\
\leq &\, c_{10}(\alpha)V_{\max}^2 R^{\frac{2\alpha}{1+\alpha}}\left[\frac{1}{n^{\frac{1}{1+\alpha}}} + \frac{\log n}{n^{1-\frac{\alpha}{2}}}\right]\sqrt{\ln(1/\delta)} \\
\leq &\, c_{10}(\alpha)V_{\max}^2 R^{\frac{2\alpha}{1+\alpha}}\frac{\sqrt{\ln(1/\delta)}}{n^{\frac{1}{1+\alpha}}}.
\end{aligned}
$$

$\square$

## C   Brief Comparative Survey of Model-based RL

In this section we summarize and discuss some previous work that has studied the problem of model learning for MBRL, and compare their approach with VAML and IterVAML, whenever appropriate. We start by discussing some results from value-based MBRL (Section C.1), and afterward briefly present some policy-search-based approaches to MBRL (Section C.2). We present some examples of methods that have particularly focused on using DNN to represent the model (Section C.3). We review and compare some recent work that implicitly or explicitly incorporate the decision problem in the model learning, both in the context of MRBL (Section C.4) as well as non-RL decision problems (Section C.4.1). Finally, we briefly mention the issue of the distribution mismatch in model learning (Section C.5). We note that the division of prior work into these categories is not rigid, and a single paper might belong to several categories at the same time.

### C.1   Value-based MBRL

Model-based RL has a long history in the RL literature. The Dyna architecture is a prototypical value-based MBRL algorithm in which learning and planning are integrated [Sutton, 1990]. Based on the observed data from the environment (a real experience), a Dyna agent updates the value function using model-free RL algorithms, such as Q-learning or its variants. Meanwhile, it keeps updating the model of the environment too. The model is used to generate "hypothetical" experience (or simulated experience or "imaginary" experience), which can be fed to a value-based planning algorithm to update the value function. The Dyna architecture is quite generic and allows many choices in its instantiation. For example, the planner can be a Q-learning agent that uses the experience generated by the model as if they are coming from the environment (sample backup) or it can be a dynamic programming one that uses full backups, or anything in between.

Even though the Dyna architecture does not prescribe any particular model learning method, its instantiations often use a model learning method that ignores the underlying decision problem. For example, the work of Sutton [1990] learns a deterministic model of the environment by recording the next-state reached from each state-action pair into a lookup table. For stochastic environments, Peng and Williams [1993] use a lookup table to count the state transitions, thus calculating the empirical estimate of the transition probabilities. This estimate can be seen as a MLE with multinomial distribution. In contrast to IterVAML, these approaches do not consider the value function in learning the model.

IterVAML can be seen as a variant of Dyna where the model learning is value-aware and the planning, which is performed by a step of approximate value iteration, is interleaved with the model learning.

As already mentioned in Section 3, it is possible to use the data from real experience not only to learn the model but also to update the value function. This makes IterVAML closer to the spirit of Dyna where both real and imaginary data are used for planning, with the crucial difference that the model learning takes the value function into account.

One of the key results of MBRL is the equivalence between model-free RL and MBRL under certain conditions. Parr et al. [2008], Sutton et al. [2008] show that if the value function is represented by a linear function approximator of a set of features, the solution of Least-Squares Temporal Difference (LSTD) algorithm [Bradtke and Barto, 1996] is the same as the value function of a MBRL that uses a linear model to learn the expected next-feature vector. We believe this is an important result, so we explain it in more detail.

Let us first recall the LSTD solution. Suppose that we represent the value function $V$ by a function within the value function space $\mathcal{F} = \left\{ V(x) = \phi^\top(x)w \, : \, w \in \mathbb{R}^p \right\}$, where $\phi(x) \in \mathbb{R}^p$ is the basis (or feature) function. For simplicity, assume that the state space is finite $\mathcal{X} = \{x_1, \ldots, x_n\}$. In general, $n \gg p$. The $n \times p$ matrix $\Phi = [\phi^\top(x_1); \ldots ; \phi^\top(x_n)]^\top$ indicates the effect of applying the basis function to each of states. The LSTD solution is the fixed point of the projected Bellman operator, i.e., $V = \Pi_\mathcal{F} T^\pi V$, where $\Pi_\mathcal{F}$ is the projection onto $\mathcal{F}$. When $\mathcal{F}$ is the linear function space as defined above, the solution is $V_{\mathrm{TD}}(x) = \phi^\top(x)w_{\mathrm{TD}}$ with $w_{\mathrm{TD}} = (\Phi^\top \Phi - \gamma \Phi^\top \mathcal{P}^* \Phi)^{-1} \Phi^\top r$. This is called linear fixed-point solution and is the same solution to which a TD method will convergence, under proper conditions.

The considered MBRL is based on the linear model in the feature space. The goal of such a model is to learn a $p \times p$ matrix transformation $\hat{\mathcal{P}}_\phi$ that approximately maps the feature vector $\phi(x)$ to the expected value of the next-feature vector $\mathbb{E}[\phi(X')|X = x]$, where $X' \sim \mathcal{P}^*(\cdot|x)$. That is, the goal is to find $\hat{\mathcal{P}}_\phi$ such that $\Phi \hat{\mathcal{P}}_\phi \approx \mathcal{P}^* \Phi$. One may also learn a reward function $r_\phi$ such that $\Phi r_\phi \approx r$. The matrix $\hat{\mathcal{P}}_\phi$ is not a stochastic matrix, so it does not have a probabilistic interpretation. Nevertheless, one may treat it as a transition matrix and define a corresponding value function in the feature space $\tilde{V}(\phi(x)) = \phi^\top(x) \sum_{k \geq 0} \gamma^k \hat{\mathcal{P}}_\phi^k r_\phi$. One may call the matrix $\hat{\mathcal{P}}_\phi$ the *Linear Expected Model* (LEM) or more accurately linear expected-feature model, as it is a linear model that predicts the expected value of next features.

The surprising fact is that the value function $\tilde{V}(\phi(x))$ obtained by using LEM coincides with the value function obtained by LSTD, a model-free algorithm. The equivalence of $V_{\mathrm{TD}}(x)$ and $\tilde{V}(\phi(x))$ is shown by Theorem 3.1 of Parr et al. [2008] and Theorem 3.3 of Sutton et al. [2008]. This is an interesting result as it indicates that with certain choices of value function and model representations as well as model learning and value learning procedures, the result of a model-free and model-based RL algorithms are the same.

There are some similarities between VAML, IterVAML, and LEMs. The desire to have a model satisfying $\Phi \hat{\mathcal{P}}_\phi \approx \mathcal{P}^* \Phi$ can be expressed as minimizing the loss function (cf. Eq. (4) of Parr et al. 2008)

$$\min_{\hat{\mathcal{P}}_\phi \in \mathbb{R}^{p \times p}} \sum_x \left\| \hat{\mathcal{P}}_\phi^\top \phi(x) - \mathbb{E}_{X' \sim \mathcal{P}^*(\cdot|x)} [\phi(X')] \right\|_2^2.$$

We may compare this minimization problem with the minimization problem (9) of IterVAML and (5) of VAML. LEM finds a linear transformation that matches the expected next-state feature vector $\mathbb{E}_{X' \sim \mathcal{P}^*(\cdot|x)} [\phi(X')]$. IterVAML finds a transition probability kernel such that the next-state expectation of the value function $\hat{V}_k$ matches the expected value function $\mathbb{E}_{X' \sim \mathcal{P}^*(\cdot|x)} \left[ \hat{V}_k(X') \right]$. A crucial difference is that IterVAML searches in the space of transition probability kernels, whereas the matrix obtained by LEM does not have a probabilistic interpretation. The comparison with the loss of VAML is less obvious because of taking the supremum of $V$ over $\mathcal{F}$. When $\mathcal{F}$ is a linear function space, however, the minimization problem becomes

$$\min_{\hat{\mathcal{P}} \in \mathcal{M}} \int \mathrm{d}\nu(x) \left\| \mathbb{E}_{X' \sim \hat{\mathcal{P}}(\cdot|x)} [\phi(X')] - \mathbb{E}_{X' \sim \mathcal{P}^*(\cdot|x)} [\phi(X')] \right\|_2^2,$$

as shown by Farahmand et al. [2017a]. This is similar to the minimization problem of LEM except that it searches in the transition probability space $\mathcal{M}$ instead of in $\mathbb{R}^{p \times p}$.

There are some extensions of LEMs, which we briefly mention. One direction is to extend model learning of MDPs, which has primitive actions, to semi-MDPs and *options* [Sutton et al., 1999].

This extension allows actions to be temporally extended. Sorg and Singh [2010] introduce Linear Option Expectation Model. The model is similar to $\hat{\mathcal{P}}_\phi$ above, with the difference that it is a matrix transformation that maps the current features to the expected feature value at an option's termination, instead of the expected value of the features at the next-state. A similar equivalence exists between the value function obtained using the Linear Option Expectation Model and the solution of an extension of LSTD to the option framework.

Related to LEM is the work of Grünewälder et al. [2012] that directly estimates the conditional mean embedding operator, i.e., the mapping $V \mapsto \int \mathcal{P}(\mathrm{d}x'|x,a)V(x')$ for all $V$ in an RKHS. The computational cost of that work is improved by Lever et al. [2016]. See also Yao et al. [2014] who introduce the concept of pseudo-MDP and factored linear action models, which relaxes the constraint that $\mathcal{P}$ should be a probability kernel, and Ávila Pires and Szepesvári [2016] who analyzed and provided policy error bounds of such models.

We briefly note that work of Ormoneit and Sen [2002] for learning the transition probability kernel $\mathcal{P}$ by a particular finite approximation that is obtained by smoothing kernel. The method uses the estimated finite MDP to find the approximate value function. A computational improvement of that work is the method of Barreto et al. [2011] that finds a smaller finite approximation of the transition probability kernel using the stochastic factorization trick. These methods learn a transition model that does not explicitly benefit from the structure of the value function beyond the required condition that the value function should be Lipschitz continuous (cf. Lemma 2 of Ormoneit and Sen [2002]). This is in contrast with VAML and IterVAML that explicitly take the value function space $\mathcal{F}$ or the current estimate of the value function $\hat{V}_k$ into account.

## C.2 Policy-Search MBRL

Learning a model to be used in conjunction with a policy search algorithm has also been explored in RL as well as in control engineering. We briefly describe a few methods.

Deisenroth et al. [2015] propose the PILCO (Probabilistic Inference for Learning Control) algorithm that performs policy gradient w.r.t. the estimated Gaussian Process (GP) model of the environment. GP allows one to maintain a probability distribution over models, which in turn defines a probability distribution over future states. The probability of future states can be used to compute the gradient of the performance w.r.t. the policy parameters. PILCO is a Bayesian framework that computes the Bayesian average of the policy gradients w.r.t. the posterior distributions of the future states, which presumably leads to better estimates compared to one that is obtained using MLE or MAP. The Bayesian averaging is performed analytically, but the computations require several approximations: approximating the next-state distribution conditioned on the Gaussian distribution of the current state by a Gaussian, approximating the distribution over the action of a nonlinear policy with a Gaussian state by a Gaussian, and being limited to special types of reward functions whose expectation over a Gaussian distribution can be computed analytically. These approximations notwithstanding, they show good empirical results in some real-world physical control problems.

PILCO is quite different from IterVAML. First of all, PILCO is a policy search algorithm whereas IterVAML is a value-based approach. PILCO is a Bayesian framework and maintains an uncertainty set over the estimated model. In contrast, IterVAML, as formulated in this paper, is not, as it only returns a single estimated model. PILCO decomposes the model learning and planning part, and does not incorporate the decision problem into model learning. This is in contrast with IterVAML where the decision problem, through the estimated value function, guides the model learning.

Some models have certain structures/regularities that can be exploited to facilitate policy search. This is common in control engineering where there are relatively straightforward procedures to find a policy (i.e., controller) when the dynamics has some special structure, such as when it is a linear dynamical system or is nonlinear but feedback linearizable. With these structures, finding a policy can be formulated as an algebraic problem, which often lead to an analytical solution. For example, designing a feedback policy for a linear dynamical system described by $x(t+1) = Ax(t) + Bu(t)$ (with $x \in \mathbb{R}^d$ being the state and $u \in \mathbb{R}^p$ an action, and matrices $A \in \mathbb{R}^{d \times d}$ and $B \in \mathbb{R}^{d \times p}$ describing the dynamics and its interaction with the input action) can be formulated as finding a matrix $K \in \mathbb{R}^{p \times d}$ such that the eigenvalues of $A - BK$ have certain properties, for example all of them be within a unit circle, which guarantees stability. The (feedback) policy would then be

$u(t) = -Kx(t)$. This is merely an example; there are many methods in control engineering that can deal with more complex dynamics, including some classes of nonlinear dynamics.

Sometimes the structure of the dynamics is known, but the exact parameters describing it is not, e.g., we know that $x(t + 1) = Ax(t) + Bu(t)$, but $A$ and $B$ are unknown. We may use data to estimate the unknown parameters, and use the estimated model to design a policy. Estimation of the unknown parameters is called *system identification* in the control engineering literature, and is the same as model learning in the sense we use in this paper. The field of *adaptive control* studies how one may find a policy with desirable properties when the dynamical system is unknown. A common approach is based on estimation of the model (for example, by minimizing the squared error between prediction and the observed data, which in turn can be seen as finding a MLE under a Gaussian noise assumption), and finding a policy based on the most recent estimated model, assuming that the model is the correct one. The estimation and policy search modules are run concurrently and online, and both the model and the policy get updated as more data become available. The design principle based on the assumption that the estimated model is the actual one is called the *certainty equivalence principle*. And having separate model estimation and policy search modules is sometimes called indirect adaptive control (in contrast to direct adaptive control where a separate model is not directly estimated).

This brief description of adaptive control shows that there are clear parallels between MBRL and (indirect) adaptive control. Both of them estimate a model, and use it in order to find a good policy. Many adaptive control methods do not find a value function, but directly find a policy; hence mentioning it in this section. The main difference, however, is that one often makes more assumptions about the model in control engineering, compared to in RL, e.g., the model is assumed to be a linear dynamical system with unknown parameters. The adaptive control literature is vast, and we do not attempt to summarize it here. Some textbooks are Astrom and Wittenmark [1994], Krstic et al. [1995], and a recent survey is by Benosman [2018].

Levine and Abbeel [2014] propose learning a time-varying linear Gaussian dynamical system. A linear model is not generally accurate and expressive enough as a global model of a complex environment (e.g., a robot), but it is sufficient for finding a local policy, which is optimal w.r.t. the linear model and the quadratic approximation to the reward (or cost). This is done through an Iterative Linear Quadratic Regulator-like (iLQR) procedure [Li and Todorov, 2004]: iLQR iteratively estimates a linear model, finds a locally optimal policy using Linear Quadratic Regulator (LQR) design, and uses the local policy to rollout trajectories to collect more data, and hence improve the local linear models. The original iLQR paper [Li and Todorov, 2004] assumes dynamics is known, but Levine and Abbeel [2014] learn it. As is common in methods developed in control engineering, iLQR benefits from the linear structure of the local models to easily find locally optimal policy through the LQR procedure, which has an algebraic formulation. The limitation of iLQR is that the obtained policy is valid only locally. Levine and Abbeel [2014] formulate the problem of learning a globally valid policy as a supervised learning problem of fitting a global policy to the local policies, i.e., minimizing the KL-divergence between the global policy and the local policies. This procedure, which is called Guided Policy Search, allows one to learn a global policy that is more complex than local policies, as fitting a complex policy through a supervised learning procedure is often easier than directly solving the globally optimal policy. Levine et al. [2016] use a variant of this procedure to learn to control a robot by visual observations. The work of Levine and Abbeel [2014] can be seen as an example of benefiting from the special structure of a model in order to find a good policy, and enhance it using techniques from ML. The model learning part of that work, however, does not benefit from the underlying decision problem, which is in contrast to VAML and IterVAML.

### C.3    DNN-based MBRL

Several papers focus on MBRL methods that use DNN to represent the model. Even though in many cases the underlying learning principle is not limited to DNN, we summarize them here because of their special focus on DNN-related techniques. There are some other papers that also use DNN, for example Predictron by [Silver et al., 2017] and Value Prediction Network (VPN) by [Oh et al., 2017], but we have categorized them in a separate section (Section C.4) as we want to emphasize their decision-aware interpretation instead of their use of DNN. A recent survey on deep RL by Li [2018] overviews several model-based DRL approaches.

Oh et al. [2015] develop a model learning algorithm for environments with an image-like observation, e.g., images from Atari games. They design DNN architectures that predict the next image conditioned on the previous images and the current action. The architecture first extracts features from the images using a series of convolutional layers and a fully connected layer. Since an observed image may not be a state of the environment, one has to somehow summarize the history of observations to provide an approximation to the true state of the system. In one of the architectures, this is done by considering a fixed-window of the past observations as the input to the network, while in the other an LSTM module accumulates the output of the previous feature extraction layers, which are only given the latest observation. These define the encoder of the architecture. To incorporate the effect of the current action, a linear transformation of the action representation, which is a vector, is multiplied to the output of the encoder. The decoder of the network converts the obtained representation to the prediction of the next image through a series of deconvolutions.

The model is deterministic and may not perform well in stochastic environments with significant stochasticity. The loss function is the squared loss, and is minimized over multiple time steps in the future, by unrolling the model. Because of this choice of the loss, this model learning procedure is not decision aware.

The experiments of the paper evaluate the quality of the learned models in various ways. One is by considering how well a DQN trained on real images can control the environment defined by the learned model (with occasional resetting to the ground state). This result seems to be more an indication of the robustness of DQN in controlling the learned model, and not an indication of how the learned model can be helpful to plan a policy that performs well in the true environment. The paper also uses the learned model to guide exploration. The samples generated by the learned model are not used to train DQN, i.e., the model is not used in a Dyna-style architecture.

Watter et al. [2015] propose the *embed to control* (E2C) method, which can be viewed as a policy search-based MBRL algorithm. E2C constructs a latent representation $z$ of the agent's observation $x$, which is potentially high-dimensional, using a variational auto-encoder (VAE)-style formulation. It learns the transition model of the agent in the latent representation, i.e., it learns $\mathcal{P}(z_{t+1}|z_t, a_t)$. E2C uses locally optimal planners, such as iLQR [Li and Todorov, 2004], that are based on linearization of the dynamics. A key aspect of E2C is that instead of linearizing the estimated transition model in the latent space, it enforces the transition model to be a (time-varying) linear model. That is, it learns a model in the form of $\hat{z}_{t+1} = A_t z_t + B_t a_t + \eta_t$ with $A_t$ and $B_t$ being time-varying matrices that depends on $z_t$. The predicted latent variable $\hat{z}_{t+1}$ is compared with the latent representation $z_{t+1}$ of the real observed data $x_{t+1}$, which is obtained by the decoder of the VAE. This time-varying linear form of the learned model makes it compatible with methods such as LQR that work with linear dynamical systems.

Ha and Schmidhuber [2018] study the problem of learning a representation and model of the environment, focusing on problems where the observations $o$ is high-dimensional (specifically a visual input) but may not be the true state $x$ of the agent in the environment, i.e., input alone does not contain all information about the past. The suggested approach encodes the (visual) observation $o_t$ onto a lower-dimensional representation $\bar{o}_t$ by using a VAE. It then uses an RNN to learn the dynamics of the environment, which is a function of $\bar{o}_t$ as well as the hidden state $h_t$ of the RNN. The paper uses Mixture Density Network combined with an RNN to model the *distribution* of the next approximate state, i.e., it learns $\mathcal{P}(\bar{o}_{t+1}|\bar{o}_t, h_t, a_t)$. This probabilistic model is a notable aspect of this work that distinguishes it from the use of LSTM by Oh et al. [2015], which only learns a deterministic model. The learned dynamics is used to "dream" rollouts (or generate rollouts based on virtual samples) of following a policy $\pi$, which can then be optimized, for example by an evolutionary strategy algorithm as is done in that paper.

The focus of Ha and Schmidhuber [2018] is different from ours. Their goal is to find a good and compact predictive model of the environment, but their model learning is not decision aware. Therefore, their method may learn a representation and a model that captures aspects of the observation that are not relevant to the decision task. This is in fact noted in their Discussion. It is suggested that by learning a model that predicts the reward, one may alleviate this shortcoming. This is along our idea of learning a model that is tied to some aspects of the decision problem. But note that learning a reward of the current time step (i.e., $\hat{r}(\bar{o}_t)$ or $\hat{r}(\bar{o}_t, h_t)$) alone or even a predictive model of the reward for the next time step (i.e., $\hat{r}(\bar{o}_{t+1})$) is not enough for learning a value-aware model of the environment; it ignores the fact that the value function depends not only on the reward of the current

step and the next step, but also on all the rewards in the future. Of course this idea can be extended by trying to predict all future rewards (i.e., $\hat{r}(\bar{o}_{t+k})$ for $k = 0, 1, \dots$). This would be somehow similar to IterVAML.

## C.4 Decision-Aware MBRL

Incorporating the decision problem into model learning is one of the key aspect of this work. Iter-VAML is closely related to the VAML framework of Farahmand et al. [2017a, 2016a], which argues that learning a model by conventional probabilistic losses, such as minimizing the KL divergence, might be an overkill. Farahmand et al. [2017a] show that when we have a model approximation error (i.e., the true model of the environment does not belong to the model class from which the estimator is selected), the difference between the solution of the maximum likelihood estimate and VAML's can be significant. Beside VAML, which is reviewed in Section 2, and IterVAML, which is introduced and discussed in detail in Section 3, there are some other papers that find a model that is somehow related to the decision problem that should be solved. We summarize them in this section.

Joseph et al. [2013] note that a model with minimum prediction error does not necessarily lead to a policy with the best performance. They define a model gradient procedure that maximizes the performance. The method is similar to a policy gradient algorithm, but instead tunes the parameters of the model.

Predictron [Silver et al., 2017] learns an abstract model of the environment by enforcing that the predicted value by the model is close to the observed values. This is done by ensuring that the rollout-based estimates of multi-step returns, according to the learned model, matches the returns observed from interacting with the true environment. Concretely, it learns a transition model $\hat{\mathcal{P}}$, a reward model $\hat{r}$, a value function model $\hat{V}$, and a discount model $\hat{\gamma}$ such that its $k$-step returns

$$\hat{G}_k = \sum_{i=0}^{k-1} \left[ \bigotimes \hat{\gamma}_{1:i} \right] \hat{\mathcal{P}}^{(i)} \hat{r} + \left[ \bigotimes \hat{\gamma}_{1:k} \right] \hat{V}_k,$$

where $\bigotimes \hat{\gamma}_{1:i} = \prod_{j=1}^{i} \hat{\gamma}_j$ with the convention $\bigotimes \hat{\gamma}_{1:0} = 1$, and $\hat{V}_k$ denoting the value function estimated at the $k$-th step, be close to the return $G = R_1 + \gamma_1 R_2 + \dots$ observed from the true environment. Predictron also uses a consistency loss that ensures that various $k$-step returns $\hat{G}_k$ are close to a weighted sum of them. Note that the model learned by predictron is abstract in the sense that the discount factor $\hat{\gamma}$ can be varying, therefore, the number of steps predicted by the model does not correspond to the time steps in the true model. Predictron is presented only for the value prediction task, and does not incorporate any policy optimization/improvement aspect.

Predictron, VAML, and IterVAML are similar as they all learn models that somehow minimize the error in the value prediction. The difference is in the way the value prediction error is defined, as well as the semantic of the considered model. In predictron, the value prediction error is the difference between the prediction of the returns according to the learned model $\hat{G}_k$ and the true return $G$, which is an unbiased estimate of the value function. As such, to obtain these noisy samples from the value function according to the true model of the environment, predictron requires generating rollout trajectories within the true environment. On the other hand, VAML and IterVAML do not assume having access to the environment for the purpose of estimating the true value of a state. The value prediction error in IterVAML is based on the most recent estimate of the value function obtained through a model-based AVI procedure. The value estimation part of IterVAML is therefore not based on rollouts of the true model, but based on performing AVI. This is similar to the difference between Monte Carlo estimate of the value function vs. the bootstrapped estimate used by VI, AVI, and TD learning.

The difference of predictron and VAML is more significant. VAML does not consider any single value function in defining the value prediction error, but considers a class of value functions. As indicated in (5), VAML minimizes the worst case error over a value function class $\mathcal{F}$.

Predictron allows learning the discount factor, as opposed to the more conventional fixed discount factor at all time steps. The models learned by VAML and IterVAML have the same temporal structure as the true MDP model of the environment, but the abstract model learned by predictron does not necessarily correspond to the environment's time steps.

Value Prediction Network (VPN) [Oh et al., 2017] also incorporates some aspects of the decision problem in model learning, in a way similar to predictron and IterVAML. It learns a model and a value function such that the value of the predicted $k$-step future states matches the estimated value of the real observed state.

We describe VPN in more detail, while omitting some aspects of VPN in order to simplify the discussion. VPN learns a deterministic transition model $\hat{\mathcal{P}}$ and the value function estimator $\hat{V}$ such that certain consistencies are satisfied. A backup tree-based planning module uses $\hat{\mathcal{P}}$ in order to estimate the state-action pairs. This planning module uses $\hat{V}$ in order to bootstrap the estimate. Let us denote the result by $\hat{Q}_d$, where $d$ refers to the tree depth using in planning. The estimate $\hat{Q}_d$ depends on both $\hat{\mathcal{P}}$ and $\hat{V}$. This planning module is used for action selection as well as providing the target values for learning the value function $\hat{V}$, as shall be described shortly.

The procedure for model and value function learning works as follows. Suppose that we have a trajectory $X_1, A_1, R_1, \ldots, X_n, A_n, R_n, X_{n+1}$, with actions selected by the tree-based planning module (e.g., by selecting the greedy action w.r.t. $\hat{Q}_d$ at any state) and the state transitions coming from the true dynamics of the environment, i.e., $\mathcal{P}^*$. VPN unrolls the estimated model $\hat{\mathcal{P}}$ for $k$-steps from the initial state $X_1$ to obtain $\hat{X}_{k+1}$ (for values of $k$ between 1 to $n-1$). The predicted state $\hat{X}_{k+1}$ is different from the true state $X_{k+1}$. Instead of making sure that the predicted and the true states are close, for example by minimizing the $\ell_2$-distance between them as is done by Oh et al. [2015], VPN compares the estimates of their values. Specifically, it aims to enforce $\hat{V}(\hat{X}_{k+1})$ to be close to the true value $V^*(X_{k+1})$ at state $X_{k+1}$. As the true value $V^*(X_{k+1})$ is unknown, VPN uses a bootstrapped MC estimate of the return instead.[9] It computes the bootstrapped $(n-k)$-step return

$$\hat{G}_{k+1} = R_{k+1} + \gamma R_{k+2} + \ldots + \gamma^{n-k} \max_{a'} \hat{Q}'_d(X_{n+1}, a').$$

This is an estimate of the true value at state $X_{k+1}$ that is based on the true reward obtained from interacting with the environment, and the backup tree-based estimate of the value function $\hat{Q}'_d$ obtained from the previous estimate of the value function $\hat{V}'$ (this $\hat{V}'$ is the same as the target network in DQN, and is occasionally replaced by the most recent estimate $\hat{V}$). The loss function of VPN is then $|\hat{V}(\hat{X}_{k+1}) - \hat{G}_{k+1}|^2$, for various $k$-step ahead values, as well as initial state $X_1$. An omitted detail is that VPN learns the reward and discount models too, as the model allows varying discount factor. This is similar to predictron. Moreover, VPN learns a mapping from observations $o$ to features/abstract state $x$, and the planning and value/model estimation are performed in the latter space.

A closely related method to VPN is TreeQN of Farquhar et al. [2018]. The tree-based planning of TreeQN is an integrated part of a DNN that predicts the backup tree-based value function estimate $\hat{Q}_d$. The architecture learns $\hat{\mathcal{P}}$, which is used to expand the tree, as well as $\hat{V}$, which is used to evaluate the value of a state. This integration of planning and value estimation within DNN is in contrast with VPN where $\hat{V}$ is represented by a DNN, but $\hat{Q}_d$ is computed outside of the DNN architecture. As a result of this integration, $\hat{Q}_d$ is end-to-end differentiable. The paper benefits from this differentiability by defining a loss function in the form of $|\hat{Q}_d(\hat{X}_{k+1}) - \hat{G}_{k+1}|^2$ instead of $|\hat{V}(\hat{X}_{k+1}) - \hat{G}_{k+1}|^2$ of VPN. In other words, it finds a $\hat{\mathcal{P}}$ and $\hat{V}$ such that the backup tree-based estimate (according to $\hat{\mathcal{P}}$ and $\hat{V}$) of the value function $\hat{Q}_d$ is close to a bootstrapped MC estimate of the value function, which is obtained by following the true dynamics $\mathcal{P}^*$ and using the previous value function estimate $\hat{V}'$ to bootstrap. Farquhar et al. [2018] also present an actor-critic variant, called ATreeC, which we do not discuss.

VPN, TreeQC, predictron, and IterVAML have similarities. First, we compare VPN and predictron. Since TreeQN is quite similar to VPN and our comparison is at a higher level than the difference between TreeQN and VPN, we do not discuss TreeQN any further. Both VPN and predictron predict values using the learned model, and train their network to ensure that the predicted values are close to the real values, or an estimate thereof. VPN uses the learned model to plan using a specific planning module, while predictron, as presented, is only for policy evaluation, and not control. A difference in

their model learning procedure is that VPN uses a bootstrapped MC estimate of the value instead of the return obtained from the true environment.

VPN and IterVAML are also similar. IterVAML, as presented and theoretically analyzed, considers looking only one step ahead ($k = 1$) instead of multiple steps ahead of VPN, i.e., only the predicted state $\hat{X}_2$, starting from $X_1$, is used for comparison. Moreover, instead of computing multiple-step ahead bootstrapped return $R_2 + \gamma R_3 + \cdots + \gamma^{n-1} \max_{a'} \hat{Q}'_d(X_{n+1}, a')$, IterVAML uses $\hat{V}'(X_2)$, which corresponds to the value function estimated at the previous iteration of IterVAML. From this perspective, the model learning aspect of IterVAML as presented can be seen as a simplified version of VPN that is more clear and amenable to analysis.[10] Another important difference between them is that the model learned by VPN is deterministic, whereas the model of IterVAML is a stochastic model.

These algorithmic differences are beside the empirical focus of Silver et al. [2017], Oh et al. [2017], Farquhar et al. [2018] and the theoretical focus of this paper.

### C.4.1 Other Decision-Aware Model Learning Methods

There are some other work outside reinforcement learning and sequential decision making that either argue in favour of or explicitly consider incorporating the decision problem in learning a model. Many of them follow the same theme: The decision making problem is formulated as an optimization problem, for example a linear program or a stochastic program. Some aspect of the optimization problem, however, is not known and should be estimated from data. For example, it might be the coefficients of the cost function [Elmachtoub and Grigas, 2017, Kao and Van Roy, 2014] or the probability distribution in a stochastic program [Donti et al., 2017]. The conventional approach estimates the unknown parameters independently of the optimization problem. This approach is sometimes called "Predict, then Optimize" [Elmachtoub and Grigas, 2017]. The aforementioned papers, on the other hand, incorporate some aspects of the optimization in estimating the model. They are examples of the concept of decision-aware model learning, which is formulated and developed in the context of RL in this work.

Bengio [1997] is one of the earliest work arguing that when the ultimate goal is decision making, and not prediction, it might be beneficial to directly optimize the desired criteria instead of training a good predictor first and then training another module that takes the predictor's output as its input and optimizes the desired decision loss. This is the same argument as the decision-aware model learning, of which VAML and IterVAML are examples.

Tulabandhula and Rudin [2013] explicitly incorporate the decision cost of using a predictive model into the model estimation procedure. They consider tasks such as scheduling medical staff or deciding what house to purchase. If the parameters of the tasks are known (e.g., the duration to finish a job, or the price of a house), these tasks can be formulated as a linear programm or $\{0, 1\}$-knapsack problem, or other well-studied decision theoretic optimization-based formulations. Since the parameters of the task (e.g., the time to finish a job, or the price of a house based on its location) are not known, we can use a predictive model for the parameters based on historic data. The conventional approach is to first learn a predictive model, using loss functions that only pays attention to the quality of predictions, and then use the predictive model to estimate the (operational) cost of the task based on the predicted parameters, e.g., the predicted total time of finishing jobs, or the predicted total price of purchased houses among a selection of available ones. In the context of MBRL, this is similar to first learning a good model of the environment using MLE (or other probabilistic approaches), and then using the learned model for planning. The suggested approach, however, incorporates the operational cost in addition to the estimation loss while learning the predictive model $f$. This involves solving an optimization problem within the optimization problem corresponding to the model estimation part, i.e.,

$$\hat{f} \leftarrow \operatorname*{argmin}_{P \in \mathcal{M}} L_n(f) + \lambda_1 J(f) + \lambda_2 \min_{\pi} L_{\text{operation}}(f, \pi),$$

where $L_n(f)$ is an estimation cost (e.g., the squared error on the data points), $J(f)$ is the regularizer, and $L_{\text{operation}}$ is the cost associated with the decision problem. Here we use $\pi$ to denote the solution of the decision problem, for example the purchased houses.

Kao and Van Roy [2014] propose a decision-aware estimator of an unknown covariance matrix. They consider this problem: Given samples from a Gaussian distribution with an unknown covariance matrix $\Sigma^*$, optimize an objective function that depends on $\Sigma^*$. More concretely, the goal is to find the action (decision) vector $a \in \mathbb{R}^p$ such that $c^\top a - a^\top \Sigma^* a$ is maximized. As $\Sigma^*$ is unknown, we have to estimate it from data. When certain structure is known about the covariance matrix, for example if it has a low-rank factor, one can use Principal Component Analysis (PCA) to estimate a low-rank approximation $\hat{\Sigma}$ of $\Sigma^*$ based on the empirical covariance matrix. After estimating $\hat{\Sigma}$, one can then optimize the objective function using $\hat{\Sigma}$ instead of $\Sigma^*$. This two-step approach (i.e., estimation and optimization), which is reminiscent of conventional model learning and planning in MBRL, does not take into account the objective function that should be maximized in the estimation process. The paper formulates the covariance estimation problem as maximizing $f(\Sigma) = \max_a c^\top a - a^\top \Sigma a$ over $\Sigma$ such that $\mathbb{P}\{\Sigma|\text{data}\}$ is larger than a certain threshold and $\Sigma$ satisfies certain structural properties. They empirically and theoretically show that this approach can be more efficient than the covariance estimation procedure that does not incorporate the decision problem.

Elmachtoub and Grigas [2017] consider an optimization problem with unknown linear objective and known convex constraint. The unknown part of the objective depends on an observed covariate $x$ in a way that can be estimated from data. More concretely, we have a reward function $r(x, a) = r(x)^\top a$ with $r : \mathcal{X} \to \mathbb{R}^p$ being an unknown function of $x$, and $a \in \mathbb{R}^p$ being the action (or decision) to be found. We are given a dataset in the form of $\{(X_i, r(X_i))\}_{i=1}^n$. The goal is to solve $\max_{a \in C} r(x, a)$ for a given $x$, with $C$ being a convex set.

The conventional "Predict, then Optimize" approach first estimates $\hat{r} : \mathcal{X} \to \mathbb{R}^p$ based on data using a prediction-only loss function (e.g., squared loss), and then solves $\max_{a \in C} \hat{r}(x)^\top a$. Their approach, on the other hand, incorporates the optimization problem in the loss function. If we denote the solution of the optimization for a given $r \in \mathbb{R}^p$ by $a^*(r) = \text{argmax}_{a \in C} r^\top a$, their loss is $\ell(r, \hat{r}) = r^\top [a^*(\hat{r}) - a^*(r)]$. This can be interpreted as the regret of choosing the optimal action according to $\hat{r}$ compared to choosing the optimal action according to the true reward $r$. The estimation problem is formulated as finding the empirical risk minimizer of

$$\min_{\hat{r}} \frac{1}{n} \sum_{i=1}^n \ell(r(X_i), \hat{r}(X_i)),$$

or its regularized variant, within some function space. The challenge is that the loss function $\ell(r, \hat{r})$ is not continuous in $\hat{r}$. Their solution is to provide a convex surrogate upper bound for this loss function, by benefitting from the convexity of $C$ and linearity of the objective (their loss, however, is not necessarily differentiable).

Donti et al. [2017] consider the stochastic programming problem where the objective (and potentially constraints) are known, but the distribution of the random variable is not. As before, the conventional approach is to ignore the optimization problem and use data along with a probabilistic loss function to estimate the distribution, e.g., by finding the MLE solution within a parametric model. The decision-aware model learning approach, which they call task-based model learning, is to adapt the model such that the solution obtained by optimizing the stochastic programming w.r.t. the estimated model performs well on the true model.

### C.5  Distribution Mismatch in MBRL

An important issue that we have not addressed is the question of how to choose the data distribution $\nu$. One particular concern is that the error of a single-step model may accumulate in such a way that the multi-step predictions become useless. Talvitie [2014], Venkatraman et al. [2015] suggest hallucinated replay to mitigate this issue. This is further analyzed by Talvitie [2017], who shows that, among other things, the hallucinated replay might not be a good solution for stochastic dynamics. The selection of data distribution to learn a good model is an issue that deserves further research.

## D  Data Collection in IterVAML vs. MLE

We make a few remarks regarding how IterVAML and MLE might be compared in terms of data collection and storage.

Converting the model learning step of Algorithm 1 to an online update might be less straightforward compared to MLE. The reason is that the loss function of IterVAML depends on $\hat{V}_k$ and changes at each iteration. To solve

$$\hat{\mathcal{P}}^{(k+1)} \leftarrow \operatorname*{argmin}_{\mathcal{P} \in \mathcal{M}} \|\hat{V}_k(X_i') - \int \mathcal{P}(\mathrm{d}x'|X_i, A_i)\hat{V}_k(x')\|^2_{\cup_{i=0}^k \mathcal{D}_n^{(i)}},$$

we have to store all datasets $\cup_{i=0}^k \mathcal{D}_n^{(i)}$. In contrast, MLE does not depend on $\hat{V}_k$, so it might be easier to design an online update without keeping all the datasets. For example, if the state and action spaces are finite, MLE only requires counters to keep track of state transitions, which can easily be updated online. This is less straightforward in IterVAML. We would like to make a few remarks about this.

It is conceivable that as IterVAML proceeds, $\hat{V}_k$ converges to $V^*$ (which should happen in the ideal case of exact VI and no modelling error $\hat{\mathcal{P}}V_k = \mathcal{P}^*V_k$ at each step of IterVAML), so the model learning loss functions of IterVAML converge too. If this is the case, at least approximately, throwing out the data from previous iterations and gradual updating of $\hat{\mathcal{P}}^{(k+1)}$ from its previous value (e.g., by performing stochastic gradient descent starting from $\hat{\mathcal{P}}^{(k)}$) may be reasonable. Studying this possibility requires further research.

In real-world applications where samples are expensive, keeping all datasets may not be the main challenge, and keeping the whole batch of data can be reasonable. Note that our theoretical analysis, however, assumes that the datasets of two different iterations are independent (Assumption A1). This is only to simplify the analysis, and in practice we might reuse the same dataset in all iterations. Theoretical results by Munos and Szepesvári [2008] suggest that the dependence between iterations may not lead to significant performance degradation.

That being said, suppose that we want to strictly follow the prescription of the current theoretical result, which requires a fresh data of size $n$ in each iteration. The total number of data points used by each iteration of IterVAML will be $n$, but for MLE, under the same fresh sample collection strategy, gradually grows from $n$ to $Kn$, with $K$ being the total number of iterations. Surprisingly, the effect of using fresh data after each iteration or keep adding them does not have a significant influence on the convergence rate. The reason, briefly speaking, is that in AVI errors decrease as $\gamma^K$ but the statistical difference between having $K$ times more samples is only sub-linear in $K$. To optimally balance them we can pick a small $K$, hence a small rate-wise benefit in having $K$ more times data.

To be more concrete, consider a simpler setup of policy evaluation with samples from the stationary distribution of the policy. Inequality (30) in Section 4.2 provides an error propagation result for this case. We consider two scenarios:

In Scenario (A), we divide a total budget of $N$ samples to $K$ independent batches of $\frac{N}{K}$ samples each, each of them is used to learn a model $\hat{\mathcal{P}}^{(k)}$ for $k = 1, \ldots, K$. This is the setting of this paper's analysis that satisfies Assumption A1.

In Scenario (B), which is closer to the setup of a conventional model learning that does not require throwing out data, we keep adding $\frac{N}{K}$ data points at each iteration, so that by the $K$-th iteration we have $N$ samples in total. To be generous to this scenario, we assume that there are $N$ samples throughout all iterations, which leads to extra bias in favour of Scenario (B), especially in earlier iterations. We would like to compare the error upper bounds of these two scenarios from the model learning perspective.

Assume that the only source of error is due to the model error, i.e., $\|\varepsilon_k\| = 0$ and $\|e_k\|_{2,\rho_X^\pi} = e_{\text{model}}$. Theorem 1 shows that given $n$ samples in estimating the model, the statistical part of the error for IterVAML (i.e., $e_{\text{model}}$) behaves as $n^{-\frac{1}{2(1+\alpha)}}$. We ignore the model approximation error part of the theorem. We make a simplifying assumption that a similar theorem exists for the conventional MLE-based model learning, that is, learning an MLE model has the same sample complexity as learning an IterVAML model. This is not proven in this paper.

Under these assumptions, we conclude that the first term in (30) is $O(n^{-\frac{1}{2(1+\alpha)}})$, and its second term is $O(Q_{\max}\gamma^K)$. We can find a $K$ to optimally balance these two terms in both scenarios. Instead, we simply pick $K = \frac{c \ln N}{\ln(1/\gamma)}$ with $c > 1$. With this choice, the second term is $\frac{1}{N^c}$. Since $\frac{1}{N^c}$ for $c > 1$ is faster than any statistical rate, the second term is negligible. So we focus on the first term and compare it in both scenarios. In scenario (A), we get $\left(\frac{\ln N}{N}\right)^{\frac{1}{2(1+\alpha)}}$ and in Scenario (B), we get

$\left(\frac{1}{N}\right)^{\frac{1}{2(1+\alpha)}}$. The difference in the rate is only logarithmic. This shows that even if we throw out data after each iteration, the effect in the rate is not significant for some good, but not necessarily optimal, choice of $K$.

We emphasize that this analysis is based on the assumption that we do not reuse datasets in IterVAML, which is likely not needed and can be relaxed.

### Acknowledgments

I would like to thank the anonymous reviewers for their helpful feedback, and Mehdi Ghasemi and Murat A. Erdogdu for discussions.

## Footnotes

[2] Given a set $\Omega$ and its $\sigma$-algebra $\sigma_\Omega$, $\bar{\mathcal{M}}(\Omega)$ refers to the set of all probability distributions defined over $\sigma_\Omega$. As we do not get involved in the measure theoretic issues in this paper, we do not explicitly define the $\sigma$-algebra, and simply use a well-defined and "standard" one, e.g., Borel sets defined for metric spaces.

[3] Learning the expected reward $r$ is also a part of model learning, which can be formulated as a regression problem. For simplicity of presentation, we assume that $r$ is known.

[4]There are at least two possible differences with the standard supervised learning results. The first is that the data source may not be i.i.d., but be dependent as it comes from a trajectory of interaction of the agent with the environment. In such a case, one should use results that are available for regression with dependent input data, e.g., see Steinwart and Christmann [2009], Mohri and Rostamizadeh [2010], Farahmand and Szepesvári [2012]. The second difference is that because the target function $T^*_{\hat{\mathcal{P}}(k)} \hat{Q}_k$ is not fixed, extra care must be considered in order to analyze the behaviour of the algorithm. Such analysis, however, has been done before, e.g., see the concept of *inherent Bellman error* by Munos and Szepesvári [2008] for analyzing the function approximation error with a moving target and the refined analysis in Chapter 5 of Farahmand [2011].

[5]For policy evaluation we use the distribution $\nu_{\mathcal{X}} \in \bar{\mathcal{M}}(\mathcal{X})$, which is defined over the state space, instead of $\nu \in \bar{\mathcal{M}}(\mathcal{X} \times \mathcal{A})$, in order to define the norms. Theorem 1, even though stated for $\nu \in \bar{\mathcal{M}}(\mathcal{X} \times \mathcal{A})$, holds for $\nu_{\mathcal{X}}$ too without any change.

[6]In IterVAML, $\hat{V}_0$ is the function that approximates the reward function $r$, i.e., $\hat{V}_0 \approx r$.

[7]For example, see Theorem 1.66 of Hunter and Nachtergaele [2001], which we quote here: Let $f : K \to Y$ be continuous on $K$, where $K$ is a compact metric space and $Y$ is any metric space. Then $f(K)$ is compact.

[8]The upper bound of the integral in the latter paper does not need to go up to infinity when the function space is bounded in the norm.

[9]The paper does not present the method in this way, and only provides a loss function to be minimized. This interpretation is in light of VAML and IterVAML.

[10]Note that the algorithmic idea of IterVAML had been briefly discussed by Farahmand et al. [2016a] prior to the publication of VPN [Oh et al., 2017] and predictron [Silver et al., 2017]. The detailed description and analysis, however, are new to this work.