[Reviews · NeurIPS 2018]

Reviewer 1



The paper proposes a modification of a reinforcement learning (RL) framework, called Value-Aware Model Learning (VAML), that makes the associated optimization problem more tractable. VAML is a model-based approach that takes into account the value function while learning the model. In its original formulation, VAML poses the problem as a “min max” optimization in which one seeks a model considering the worst-case scenario over the space of representable value functions. This paper proposes to replace the problem above with a sequence of optimizations whose objective functions include the actual value-function approximations that arise in value iteration (that is, one replaces the “max” above with a sequence of concrete approximations). The paper presents a theoretical analysis of the proposed method, first providing finite sample guarantees for the model-based approximation, then providing a general error propagation analysis, and finally combining the two. The main point of the paper, to replace a worst-case minimization with a sequence of easier problems, is insightful, and can potentially lead to practical algorithms. The paper is also didactic, with clear explanations throughout that help the reader understand each step of the derivations. I have a few observations regarding the proposed approach itself and the way it is presented in the current version of the paper. It seems to me that the fact that the model P computed by IterVAML depends on the current value function approximation Q restricts the use of the data, and potentially the applicability of the method itself. The way Algorithm 1 is presented suggests that it can be used in the online scenario, where the agent alternates between collecting sample transitions and refining its value function approximation Q. When the model is independent of the current approximation Q, like in conventional model-based approaches, the data collected in successive iterations can be slowly incorporated into the model (this is the case of Dyna, for example). However, as far as I can see, the only way to incorporate all the data into a model that depends on the current Q, like in IterVAML, is to store all the sample transitions collected along the way. Although the point raised above might come across as an algorithmic limitation of lesser importance, it may in fact touch upon one of the core arguments in favor of model-based RL. Model-based approaches similar to IterVAML learn a set of conditional distributions P(.|X,A) that are then used to compute an approximation of Q(X,A). As the former object is potentially much more complex than the latter, when is it reasonable to expect that this indirect approach will be more effective than simply learning Q? One possible case is when P is in fact easier to learn than Q. Since in general there is no reason to expect this to be true, another argument in favor of model-based approaches is that P can use all the data, even if the collection of samples is influenced by the current policy computed by value iteration. This is only possible because there is a single model; one cannot do the same with value functions without storing all the data because approximation Q_{t+1} depends on Q_{t}. Thus, unless one is willing to store all samples collected, P can indeed become a better approximation than Q_t, for a large enough t, despite the fact that it is a more complex object. However, by tying each of its models to a value function approximation, IterVAML loses the ability to incorporate all the data into the model without storing all the samples. In summary: when IterVAML is used online, it seems less likely that the strategy of computing an approximation P to then compute Q will be as effective as in other model-based approaches (including the original formulation of VAML). Note that this issue does not arise in batch RL, where one has a single, fixed, set of sample transitions. The current version of the paper also lacks a more intuitive discussion of the potential benefits of taking the value function into account while learning the model. In line 82 the authors argue that the objective function optimized by the original VAML is tighter than that optimized under other criteria, such as maximum likelihood estimation. It would be nice to see a more intuitive explanation of this point, perhaps specialized to IterVAML, especially if such an explanation were accompanied by some simple experiment illustrating the point. It would also be interesting to see simple experiments illustrating how IterVAML compare against other model-based approaches, such as the original VAML or some standard algorithm that minimizes the KL-divergence. Such a comparison would also provide some intuition on the issue raised on the previous paragraph. Other points: -- The discussion on the policy evaluation version of IterVAML, starting in line 141, seems a bit lengthy to me. Why not simply say that the evaluation case can be obtained if we define a new MDP with a single action per state corresponding to the action that the policy to be evaluated would pick? -- Assumption A2 is difficult to understand; having a more intuitive explanation here would help a lot. -- The presentation seems a bit rushed at times. For example, in line 269 the authors promise an explanation that is never given (probably a result of editing the full version of the paper). The paper also ends quite abruptly; it would be nice to have a summary in the end reminding the reader of all the contributions. Minor corrections: Line 88: provide *an* expression 101: MDP -> MDPs 199: It is probably best to use another notation for the radius, because ‘R’ is too much associated with rewards in the context of RL 201: I think mathcal{N} is not defined 229: Closing parenthesis is wrongly superscripted 255: function -> functions 257: *an* integer k \ge 0 258: “denote” shows up twice Post-rebuttal update: -------------------------- Thank you for the very detailed explanation addressing my concern! I suggest the authors mention this issue in the main paper and perhaps add the discussion in the rebuttal to the appendix. It would also be nice to see simple experiments illustrating the behavior of the proposed approach in practice, as mentioned in some of the reviews and also in the rebuttal.

Reviewer 2



This paper deals with model-based reinforcement learning. The objective is to learn a model of the transition dynamics of the MDP at hand. This paper uses the Value aware Model learning [Farahmand et al., 2017a] where the problem is formulated as finding a good model of the MDP with respect to the task to solve therefore ignoring details of the MDP that could be not important to solve the task. For that purpose, this paper proposes to learn the model while solving the task and with respect to the current learned value function while other previous approaches proposed to be robust to a potentially large set of value functions. The paper is clearly written. The new approach is convincing and seems to be the main contribution of the paper. It seems to give a natural way to guide the model learning with respect to the task at hand without being too conservative. The theoretical analysis seems to be following the classical tools in the literature. An experimental evaluation of the idea would have been of interest.

Reviewer 3



This paper proposes IterVAML, an iterative version of value-function aware model learning for RL where a dynamics model is being fitted to minimize the bellman error during every iteration of an approximate batch value-iteration step. I think the idea of fitting a model in order to minimize bellman error is a neat idea, and this iterative version makes it a much easier algorithm to implement in practice. I'm not very familiar with RKHS and metric entropy so I only skimmed the proofs. It would've been nice to see some toy experiments, perhaps in simple discrete MDPs with linear function approximation, comparing this approach with just learning an MLE model. On a broader note of discussion, if ultimately you will be using the learned, approximate value function to get a policy, I'm not sure if there's benefit to learning a model and performing model-based RL as opposed to something model-free like fitted Q-iteration. While fitted Q-iteration could be seen in some ways as using the empirical MLE model, it is also more directly fitting a value function. This approach ends up using your data twice, once to fit an approximate model, and then again to fit a value function. Again, perhaps some empirical experiments in toy domains could illustrate the difference. Overall I liked the idea, and this could potentially result in improved value iteration especially when scaled up in complex domains where complex function approximators are used. === After Author Rebuttal === I do like the theoretical contribution of the tighter bound. If the authors can provide simple toy experiments, basically to show that their method is not completely impractical, I would be fully convinced. Still, I do think just based on the theory I would still accept this paper.